# Orexin signaling modulates synchronized excitation in the sublaterodorsal tegmental nucleus to stabilize REM sleep

Hui Feng[1,2], Si-Yi Wen[1,2], Qi-Cheng Qiao[1,2], Yu-Jie Pang[1], Sheng-Yun Wang[1], Hao-Yi Li[1], Jiao Cai[1], Kai-Xuan Zhang[1], Jing Chen[1], Zhi-An Hu[1], Fen-Lan Luo[1], Guan-Zhong Wang[1], Nian Yang[1✉] & Jun Zhang[1✉]

The relationship between orexin/hypocretin and rapid eye movement (REM) sleep remains elusive. Here, we find that a proportion of orexin neurons project to the sublaterodorsal tegmental nucleus (SLD) and exhibit REM sleep-related activation. In SLD, orexin directly excites orexin receptor-positive neurons (occupying ~3/4 of total-population) and increases gap junction conductance among neurons. Their interaction spreads the orexin-elicited partial-excitation to activate SLD network globally. Besides, the activated SLD network exhibits increased probability of synchronized firings. This synchronized excitation promotes the correspondence between SLD and its downstream target to enhance SLD output. Using optogenetics and fiber-photometry, we consequently find that orexin-enhanced SLD output prolongs REM sleep episodes through consolidating brain state activation/muscle tone inhibition. After chemogenetic silencing of SLD orexin signaling, a ~17% reduction of REM sleep amounts and disruptions of REM sleep muscle atonia are observed. These findings reveal a stabilization role of orexin in REM sleep.

[1] Department of Physiology, Third Military Medical University, 400038 Chongqing, P.R. China. [2] These authors contributed equally: Hui Feng, Si-Yi Wen, Qi-Cheng Qiao. ✉email: yangnian@tmmu.edu.cn; zhangjun@tmmu.edu.cn

S table vigilance states, which fundamentally serve vital brain functions, depend on the regulation of diverse neuro-chemical signals[1]. Among them, the hypothalamic neuro-peptide orexin/hypocretin is indispensably involved in this process[2–5]. Orexin signaling is provided by diffusely distributed fibers, and separately assigned to different brain circuitries that orchestrate sleep/wakefulness states[6,7]. Intriguingly, in addition to the prominent deficits in maintaining wakefulness state after orexin deficiency[7], impaired quality of rapid eye movement (REM) sleep has been observed in increasing clinical observations[8–10]. The sublaterodorsal tegmental nucleus (SLD), which is regarded as both necessary and sufficient for the generation and maintenance of REM sleep[11–13], has been reported to integrate a variety of neurochemical signals in sleep/wakefulness cycles[14]. Orexin fibers are also detected in the SLD[15]. Moreover, a previous report has found that microinjection of orexin into the SLD produces a REM sleep-like pattern of muscle tone decrease in decerebrated rats[16]. Therefore, the REM sleep relevance and the exact function of orexin signaling in the SLD, especially the underlying mechanisms need to be clarified.

Orexin signaling is generally excitatory in nearly all target brain regions, including the SLD[17]. The SLD mainly contains gluta-matergic neurons that constitute elementary units of its REM sleep-related output[12,18]. During REM sleep, the firing activities of the SLD glutamatergic neurons are carriers of output infor-mation for brain state activation, such as fast electro-encephalogram (EEG) activities in the corticohippocampal region, and muscle atonia[11,19,20]. In general, the elevated firing activities in individual SLD glutamatergic neurons have been found to contribute to the SLD output[19,21,22]. Intriguingly, in addition to the readily observed membrane potential level of the SLD neurons[17,23], several forms of sub-threshold fluctuations, such as gamma band and spikelet activities, are also involved in shaping their firings[24,25]. Furthermore, correct orchestration of the elevated firings are also thought to be required for the output of a given neuronal network, including the SLD[26–28]. It has been found that the pontine wave, which is thought to mirror a coordinated firing pattern of the SLD[17,29], is highly correlated with the strength of the hippocampal theta activities during REM sleep[27]. While loss of orexin signaling has been reported to cor-relate with impaired brain states reflected by abnormal fast-EEG activities and muscle tone regulation in either wakefulness or REM sleep[9,10,30], the specific influences of orexin on the neuronal membrane and network dynamics in the SLD, and their con-tributions to the functional brain state during REM sleep have not been investigated.

Here, the REM sleep relevance of SLD orexin signaling is first uncovered through retrograde tracing and c-Fos screening of the SLD-projecting orexin (OX$^{SLD}$) neurons. These orexin neurons are non-overlapping from those projecting to the wake-promoting locus coeruleus (LC). A global excitation of orexin in the SLD is next identified. Through in vitro/vivo electro-physiological assays, we identified an interaction between orexin's effects on neuronal membrane dynamics and electrical connec-tions, which supports global and synchronized excitation in the SLD network. The consequent enhancement of the SLD output and its contribution to consolidating brain state activation and muscle tone inhibition are revealed through optogenetics. Com-bining fiber photometry, we further find that all these substrates underlying SLD orexin signaling are employed during REM sleep to prolong individual episodes. A reduced REM sleep amount caused by disrupted REM sleep performances, including insuffi-cient levels of muscle atonia and EEG theta oscillation power, is finally observed after chemogenetic inhibition of SLD orexin signaling. Thus, an essential role for SLD orexin signaling in REM sleep stabilization is presented.

## Results

**OX$^{SLD}$ neurons show REM sleep-related activation.** We first identified the OX$^{SLD}$ neurons and examined whether their activities were related to REM sleep. Cholera toxin subunit B (CTB)-488 and CTB-555 were injected into the SLD and its neighboring LC, respectively, to label the origins of orexin sig-naling (Fig. 1a, b). CTB-488 and CTB-555 labeled orexin-A$^+$ neurons accounted for $7.1 \pm 0.8\%$ and $15.2 \pm 2.1\%$, respectively, of the total orexin-A$^+$ neurons ($n = 6$ rats). Both of them were sporadically distributed across the lateral hypothalamus (LH) (Supplementary Fig. 1a–d), indicating that the injected region (Supplementary Fig. 2a) of the SLD was innervated by orexin neurons. Intriguingly, the OX$^{SLD}$ neurons were largely non-overlapping with the LC-projecting orexin (OX$^{LC}$) neurons (overlap: $1.9 \pm 0.3\%$; $n = 6$ rats) (Fig. 1d). The orexin receptors (OXRs) and bead-like orexin-A$^+$ varicosities were also detected in the rat SLD (Fig. 1e), suggesting actions of orexin through volume transmission in this area. In the SLD, the OXRs were abundantly expressed in vesicular glutamate transporter-1 (VGLUT1)-positive glutamatergic neurons (Fig. 1f), and few OXRs/glutamate decarboxylase-67 (GAD-67)-positive GABAer-gic somas were detected (Fig. 1g). The abundance of glutama-tergic, but not GABAergic, neurons in this SLD region was also observed by using glutamate and GABA antibodies (Supple-mentary Fig. 2b, c).

We then assessed the REM sleep relevance of SLD orexin signaling. Rats with CTB-488 injection in the SLD were subjected to a 72-h REM sleep deprivation (RSD), and c-Fos screening was performed after the following REM sleep rebound (Fig. 1h). The REM sleep amount of these rats ($n = 7$) increased to $39.0 \pm 2.8\%$ in this period, compared to that of the control rats ($9.1 \pm 0.8\%$; $n = 4$) with normal sleep/wakefulness cycles (Fig. 1i). When analyzed as an entirety, the orexin neurons (OX$^{entirety}$) exhibited no difference in c-Fos expression between the control and RSD groups (control: $7.1 \pm 1.3\%$, $n = 4$ rats; RSD: $6.0 \pm 0.6\%$; $n = 7$ rats; $P = 0.435$) (Fig. 1j, k). Nevertheless, c-Fos expression was dramatically increased within the OX$^{SLD}$ neurons (control: $6.8 \pm 2.8\%$, $n = 4$ rats; RSD: $28.2 \pm 2.9\%$, $n = 7$ rats; $P = 9.221 \times 10^{-4}$) (Fig. 1j, k). These data revealed that the OX$^{SLD}$ neurons exhibited increased activity in the presence of enriched REM sleep, implying the REM sleep relevance of SLD orexin signaling. In contrast, a proportion of the LH melanin-concentrating hormone (MCH) neurons also innervated the SLD, and they exhibited similar c-Fos expression level to that of the total MCH population after the REM sleep rebound (Supplementary Fig. 3a–d). More-over, the abundant expression of OXRs in the SLD REM-on (c-Fos$^+$) neurons was also observed after the REM sleep rebound (Supplementary Fig. 4a).

**Orexin globally excites the electrically coupled SLD network.** Whole-cell patch-clamp recordings were employed to examine the electrophysiological effects of orexin in this glutamatergic neuron-enriched SLD region in rats (Supplementary Fig. 5a). The membrane potential of recorded neurons was adjusted to silence their spontaneous firing, and a cocktail of blockers (picrotoxin/ MK-801/DNQX) was applied to block chemical synaptic trans-mission. In this condition, orexin-A (100 nM) strongly depolar-ized all tested SLD neurons (baseline: $-63.3 \pm 1.3$ mV, orexin-A: $-55.4 \pm 1.4$ mV; $n = 32$; $P = 7.940 \times 10^{-7}$) (Fig. 2a, c), revealing a global excitation effect. In addition, before orexin-A application, carbenoxolone (CBX)-sensitive spikelet activities were detected in 21 of 32 (65.6%) tested neurons (Fig. 2b, d), suggesting that the SLD glutamatergic neurons were electrically connected. This connection was further confirmed by the expression of the neu-ronal gap junction (GJ) protein Connexin-36 (Cx-36)[31] in the

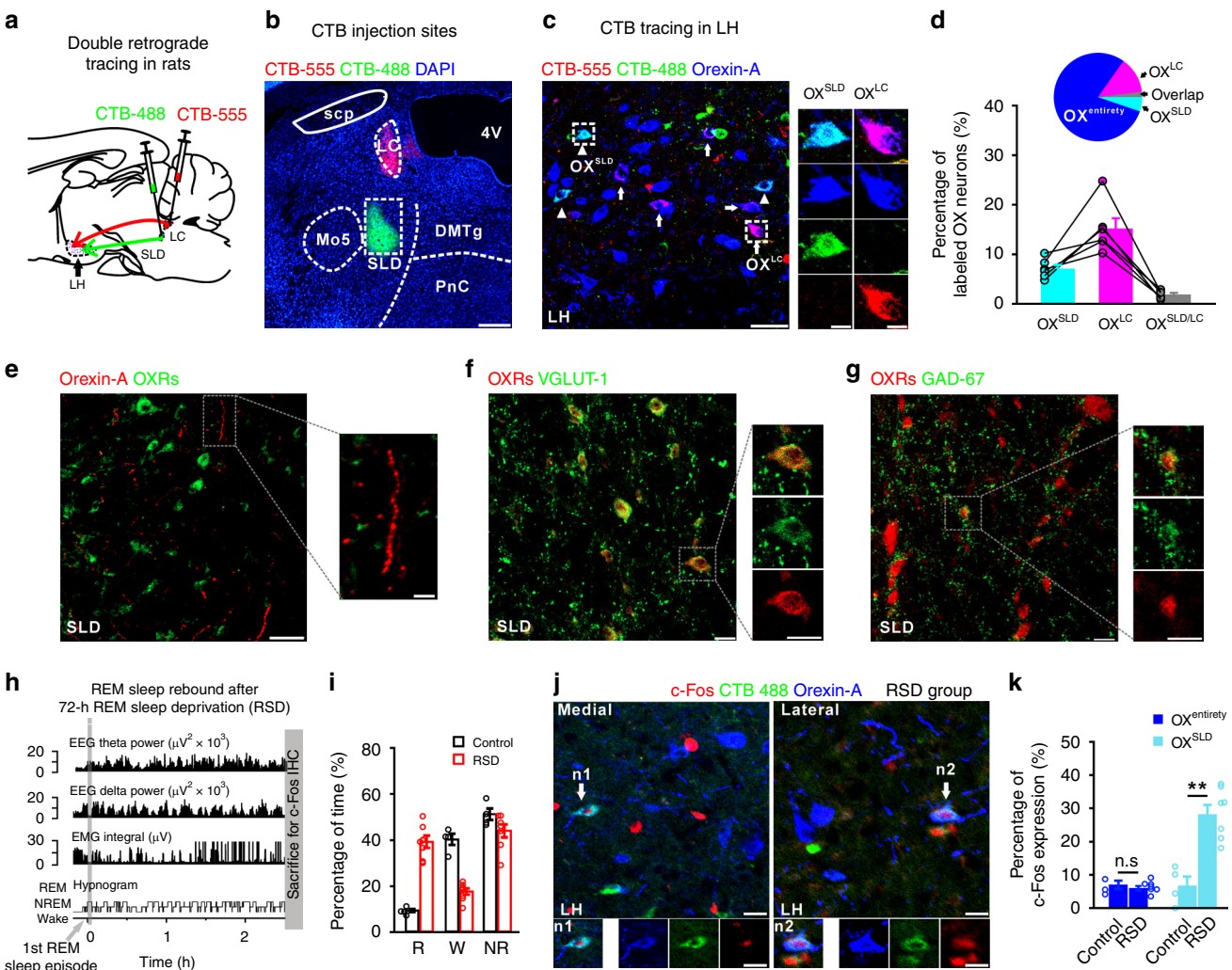

**Fig. 1 LH OX^SLD neurons show REM sleep-related activation. a, b** Schematic drawing (**a**) and representative CTB-555 (red)/CTB-488 (green) injections (**b**) for retrograde tracing. LH lateral hypothalamus, scp superior cerebellar peduncle. Mo5 trigeminal motor nucleus, 4V 4th ventricle, DMTg dorsomedial tegmental area, PnC caudal pontine reticular nucleus. Scale bar: 500 μm. **c** Representative SLD-projecting (OX^SLD, arrow heads, CTB-488^+/orexin-A^+, cyan) and LC-projecting (OX^LC, arrows, CTB-555^+/orexin-A^+, purple) orexin-A^+ (blue) neurons in the rat LH. Scale bar: 50 μm. Examples (square boxes) were magnified on the right. Scale bars: 10 μm. **d** Percentage of orexin (OX) neurons projecting to the SLD, LC, and both (OX^SLD/LC), respectively (*n* = 6 rats; analysis of 3902 orexin neurons). **e** Expression of OXRs (green) and orexin-A^+ varicosities (red) in the rat SLD. Scale bars: left, 50 μm; right, 10 μm. **f, g** Expression of OXRs (red) with VGLUT1 (green, **f**) and GAD 67 (green, **g**) in the rat SLD. All scale bars: 20 μm. **h** Representative sleep/wakefulness architecture following the 72-h RSD. The rat was sacrificed for c-Fos immunostaining (IHC) after 2.5 h from the first REM sleep episode (gray arrow). Control rats were set with normal sleep/wakefulness cycle and underwent same processes. **i** Percentage of time spent in REM sleep (R), wakefulness (W), and NREM sleep (NR) during the 2.5 h preceding sacrifice in the control (black, *n* = 4) and RSD (red, *n* = 7) rats. **j** Representative c-Fos (red) expression in orexin-A^+ (blue) and OX^SLD (cyan) neurons (RSD group). C-Fos-positive OX^SLD neurons (arrows) from the medial (n1) and lateral (n2) part of the LH were presented below. All scale bars, 20 μm. **k** Percentage of c-Fos expression in the OX^entirety (blue, two-sided unpaired *t*-test; $t_9 = 0.817$, $P = 0.435$) and OX^SLD neurons (cyan, two-sided unpaired *t*-test; $t_9 = 4.839$, $P = 9.221 \times 10^{-4}$) in the control (*n* = 4, analysis of 2767 OX^entirety and 178 OX^SLD neurons) and RSD (*n* = 7, analysis of 6012 OX^entirety and 370 OX^SLD neurons) rats, respectively. Data represent mean ± SEM. **P < 0.01. Source data are provided as a Source Data file.

SLD glutamatergic neurons (Supplementary Fig. 6a) and the spread of biocytin among them in patch-clamp recordings (Supplementary Fig. 6b–d). We also noticed that orexin-A (100 nM) elicited spikelet activities in six neurons without spikelet activities at baseline, increasing the percentage of SLD neurons exhibiting spikelet activities to 84.4% (27 of 32) (Fig. 2d).

The current basis of the orexin-elicited global excitation was further analyzed. At −70 mV holding voltage, orexin-A (30–300 nM) elicited inward currents in all 58 tested SLD neurons in normal artificial cerebrospinal fluid (ACSF). The orexin-elicited inward current was dose-dependent (Fig. 2e–g), and was

independent of chemical synaptic transmission (control: −25.3 ± 7.9 pA, blockers: −24.6 ± 7.6 pA; *n* = 7; *P* = 0.528) (Fig. 2h, i). Furthermore, the whole-cell current were usually noisy with rhythmic fluctuations (Fig. 2h), reflecting the GJ currents underlying spikelet activities[32]. With the onset of the orexin-elicited inward currents, the current noise apparently increased, and the increase persisted after blocking chemical synaptic transmission (control: 59.9 ± 26.2%, blockers: 57.4 ± 13.6%; *n* = 7; *P* = 0.854) (Fig. 2j). We further found that the orexin-elicited noisy inward currents were mediated by the activation of orexin-1 and orexin-2 receptors (Fig. 2k–m).

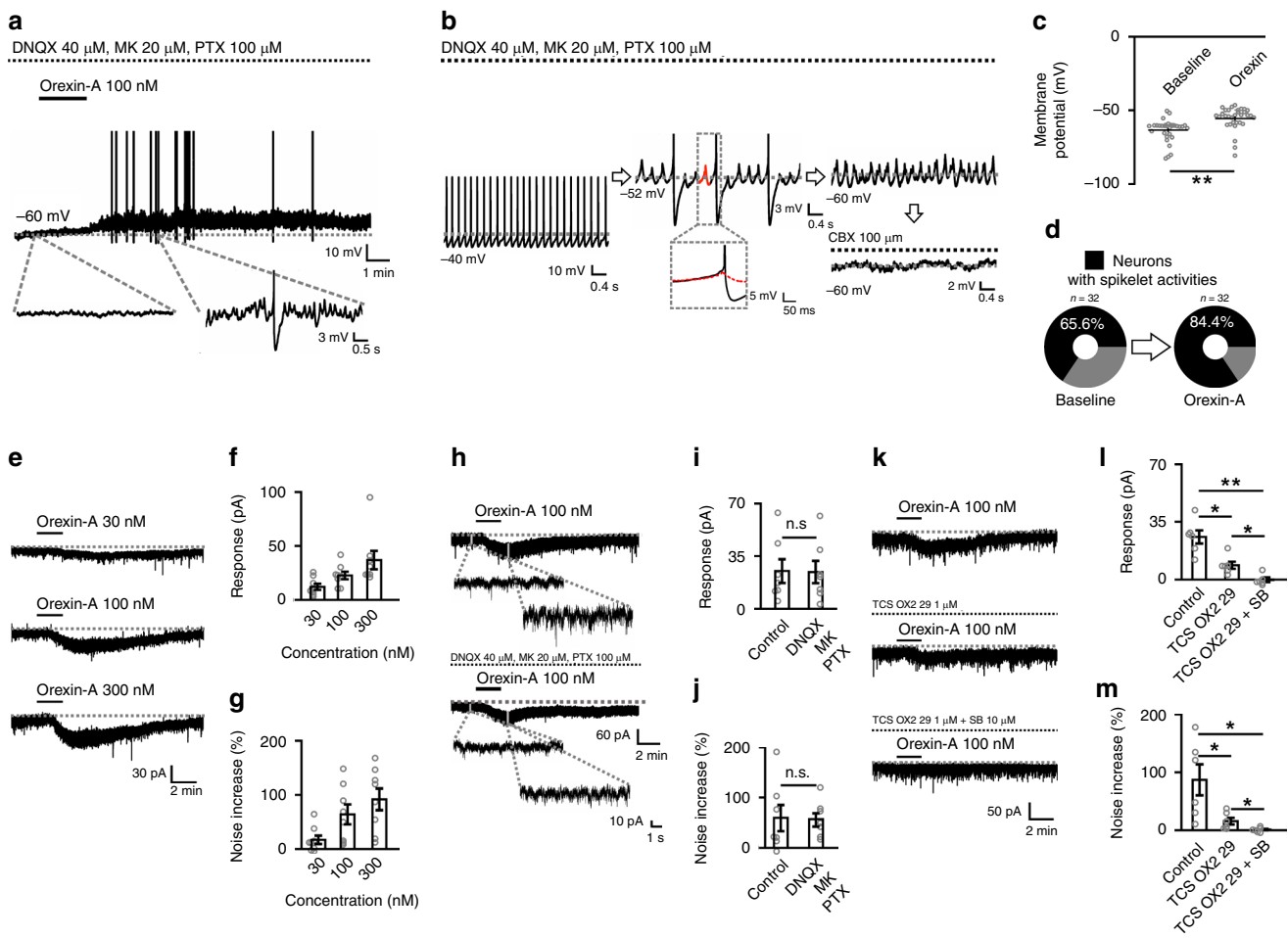

**Fig. 2 Orexin elicits depolarization and increases spikelets in the SLD neurons. a** Orexin-A (100 nM) depolarizes a rat SLD neuron in DNQX/MK-801 (MK)/picrotoxin (PTX). Expanded traces showing that orexin-A also elicited consecutive spikelets. **b** The spikelets emerged after holding down the membrane potential in a rat SLD neuron and were blocked by CBX (100 μM). The spikelet had similar rise kinetics to action potential (box labeled). **c** Membrane potential of the SLD neurons before and after orexin-A applications ($n = 32$, two-sided Wilcoxon signed-rank test; $z = 4.937$, $P = 7.940 \times 10^{-7}$). **d** Orexin-A increased the percentage of the SLD neurons exhibiting spikelets ($n = 32$). **e-g** Representative traces (**e**) and statistics showing that 30–300 nM orexin-A concentration-dependently elicited inward current (**f**) and noise increase of the current (**g**) in rat SLD neurons ($n = 8$). **h-j** Representative traces (**h**) and statistics showing that the amplitude (**i**, two-sided paired t-test; $t_6 = 0.670$, $P = 0.528$) and the noise increase (**j**, two-sided paired t-test; $t_6 = 0.192$, $P = 0.854$) of the orexin-elicited inward current remained unaffected by DNQX/MK/PTX in rat SLD neurons ($n = 7$). **k-m** Representative traces (**k**) and statistics showing that the amplitude (**l**; one-way repeated-measures ANOVA, $F_{(2, 10)} = 22.137$, $P = 2.124 \times 10^{-4}$; post hoc LSD comparison test; control vs. TCS OX2 29, $P = 0.0122$; control vs. TCS OX2 29/SB, $P = 2.218 \times 10^{-3}$; TCS OX2 29 vs. TCS OX2 29/SB, $P = 0.0204$) and the noise increase (**m**; one-way repeated-measures ANOVA, $F_{(2, 10)} = 8.721$, $P = 6.425 \times 10^{-3}$; post hoc LSD comparison test; control vs. TCS OX2 29, $P = 0.0422$; control vs. TCS OX2 29/SB, $P = 0.0249$; TCS OX2 29 vs. TCS OX2 29/SB, $P = 0.0387$) of the orexin-elicited inward current were partially blocked by TCS OX2 29, and totally abolished by TCS OX2 29/SB 334867 (SB) in rat SLD neurons ($n = 6$). Data represent mean ± SEM. *$P < 0.05$; **$P < 0.01$. Source data are provided as a Source Data file.

**The global excitation of orexin in the SLD depends on GJs.** Surprisingly, after blockage of GJ by 100 μM CBX, the orexin-elicited inward current was dramatically decreased by 42.9% (control: 26.6 ± 6.2 pA, CBX: 15.2 ± 4.3 pA; $n = 15$; $P = 3.143 \times 10^{-3}$) (Fig. 3a–c). Furthermore, in 4 of 15 (26.7%) neurons, the inward current was completely abolished by CBX (Fig. 3b, c). These data indicate an interaction between the orexin-elicited excitation and SLD GJ activities. We further employed another GJ blocker, mefloquine (MEF), for confirmation. In addition, biocytin-based post hoc immunostaining was employed to determine the expression of OXRs in each electrophysiologically tested neuron. Pretreatment (2 h) with MEF (10 μM) also reduced the amplitude of the orexin-elicited inward current by 36.2% (control: 24.3 ± 2.1 pA, $n = 58$; MEF: 15.5 ± 3.4 pA, $n = 25$; $P = 1.018 \times 10^{-3}$) (Fig. 3d, e). In addition, 6 of 25 (24%) neurons showed no

response to orexin-A after MEF pretreatment (Fig. 3d, e). Post hoc immunostaining further revealed that the orexin-A non-responsive neurons did not express OXRs (Fig. 3d). After blocking GJ activities, ~1/4 of the SLD neurons were found to be orexin-A non-responsive, suggesting that SLD orexin signaling directly targets ~3/4 of the SLD neurons. We then determined the substrates of the orexin-elicited effects that may spread through GJs to generate the global excitation. After addition of TTX to block spike generation, the spikelet currents were all abolished (Fig. 3f, g). Nevertheless, orexin-A still elicited inward currents in all tested neurons and the amplitude remained unaffected (control: 20.7 ± 2.6 pA, blockers: 22.6 ± 2.4 pA; $n = 8$; $P = 0.0589$) (Fig. 3h). Together, these results indicate that the orexin-elicited sustained depolarization by this current on the SLD OXRs[+] neurons spreads through GJs to form the orexin-elicited global

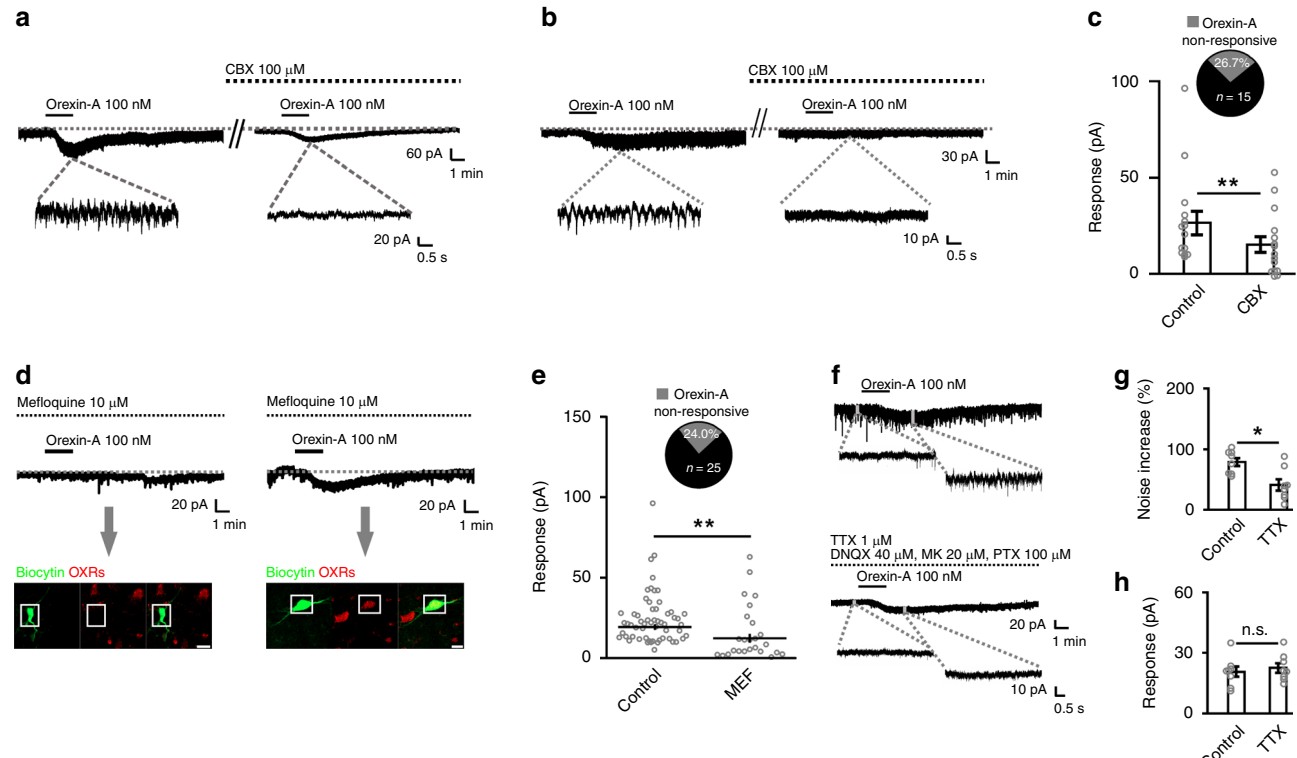

**Fig. 3 The excitation of orexin in ~3/4 of SLD neurons spreads via GJs. a**, **b** Representative traces showing that the orexin-A (100 nM) elicited inward currents were partially blocked by CBX (100 μM) in a rat SLD neuron (**a**) and completely abolished by CBX in another (**b**). Note that the spikelet currents meditated by GJs were all blocked by CBX. **c** The orexin-elicited inward current was reduced in the presence of CBX in rat SLD neurons ($n = 15$, two-sided Wilcoxon signed-rank test; $z = 2.953$, $P = 3.143 \times 10^{-3}$). Pie charts showing that the orexin-elicited inward current was completely abolished in 4 of 15 (26.7%) tested neurons. **d** Biocytin (green)-based post hoc immunostaining showed that an orexin-A non-responsive neuron from the rat SLD (left) in the presence of mefloquine (MEF) did not express OXRs (red), and another orexin-A-responsive neuron (right) from the rat SLD in the presence of MEF expressed OXRs. Scale bars, 20 μm. **e** The orexin-elicited inward current was also reduced in the presence of MEF, compared to the normal ACSF condition in rat SLD neurons (control: $n = 58$, MEF: $n = 25$; two-sided Mann–Whitney rank-sum test; $z = 3.286$, $P = 1.018 \times 10^{-3}$). Pie charts showing that the orexin-elicited inward current was completely abolished in 6 of 25 tested neurons. **f** Representative traces showing that orexin-A (100 nM) elicited inward current in a rat SLD neuron before and after the application of TTX (1 μM) plus DNQX/MK/PTX. Note that spikelet currents were observed after orexin-A application in normal ACSF. In the presence of TTX, rhythmic spikelet currents cannot be elicited by orexin-A. **g**, **h** The orexin-elicited noise increase of the inward current was attenuated in the presence of TTX (**g**, two-sided paired $t$-test; $t = 3.430$, $P = 0.0110$), but the amplitude of the orexin-elicited inward current remained unaffected in rat SLD neurons (**h**, two-sided paired $t$-test; $t = 2.253$, $P = 0.0589$) ($n = 8$). Data represent mean ± SEM. *$P < 0.05$; **$P < 0.01$. Source data are provided as a Source Data file.

excitatory effects. In addition, the depolarization may eventually evoke firings in the SLD neurons and thus contribute to the increase in spikelet activities. We also found that the orexin-elicited excitation originates from the activation of the non-selective cationic conductance (NSCC) (Supplementary Fig. 7a–g).

**Orexin increases the GJ conductance in the SLD network.** Although the orexin-elicited excitatory effects on individual SLD neurons contribute to the depolarization and the increased spikelet activities, the orexin's effect on GJs may also be involved. We thus recorded pairs of electrically connected SLD neurons in rats. In electrically connected SLD neurons, current injections in either side caused voltage responses in both neurons (Fig. 4a). After orexin-A (100 nM) application, the coupling coefficient was bilaterally increased ($n = 7$ pairs; control$_{cell1 \to cell2}$: $0.046 \pm 0.015$, orexin-A$_{cell1 \to cell2}$: $0.059 \pm 0.017$, $P = 0.0180$; control$_{cell2 \to cell1}$: $0.042 \pm 0.016$, orexin-A$_{cell2 \to cell1}$: $0.054 \pm 0.019$, $P = 0.0280$) (Fig. 4b). While the coupling coefficient describes functional coupling relationship, it does not identify the electrophysiological mechanisms, e.g. the underlying GJ conductance or input resistance of the connected cells[33]. Therefore, we further calculated

the GJ conductance in these pairs. Orexin-A (100 nM) bilaterally increased the conductance ($n = 7$ pairs; control$_{cell1 \to cell2}$: $0.35 \pm 0.09$ ns, orexin-A$_{cell1 \to cell2}$: $0.46 \pm 0.13$ ns, $P = 0.0180$; control$_{cell2 \to cell1}$: $0.31 \pm 0.08$ ns, orexin-A$_{cell2 \to cell1}$: $0.43 \pm 0.12$ ns, $P = 0.0180$), while the input resistances were not changed ($n = 7$ pairs; control$_{cell1}$: $143.0 \pm 43.7$ MΩ, orexin-A$_{cell1}$: $139.8 \pm 38.9$ MΩ, $P = 0.735$; control$_{cell2}$: $147.3 \pm 44.8$ MΩ, orexin-A$_{cell2}$: $153.5 \pm 45.8$ MΩ, $P = 0.0910$) (Fig. 4c, d). Thus, in addition to providing the original power that drives the depolarization and the increase in spikelet activities, orexin also promotes the spread of these effects in the SLD network by increasing the GJ conductance.

**Orexin modulates synchronized excitation in the SLD in vivo.** We next examined the modulations of orexin on the network dynamics of the SLD in vivo through multiple channel recordings in urethane-anesthetized rats (Fig. 5a, b). Microinjections of orexin-A (30 μM, 0.3 μl) in the SLD increased the firing frequency in 67 of 97 (69.1%) SLD units (baseline: $8.0 \pm 0.9$ Hz, orexin-A: $14.7 \pm 1.6$ Hz; $P = 1.120 \times 10^{-12}$) (Fig. 5c, d). In addition, the GJ activities were reflected in vivo by coincidental spiking within a sharp time window of ±1 ms between unit pairs. In this condition,

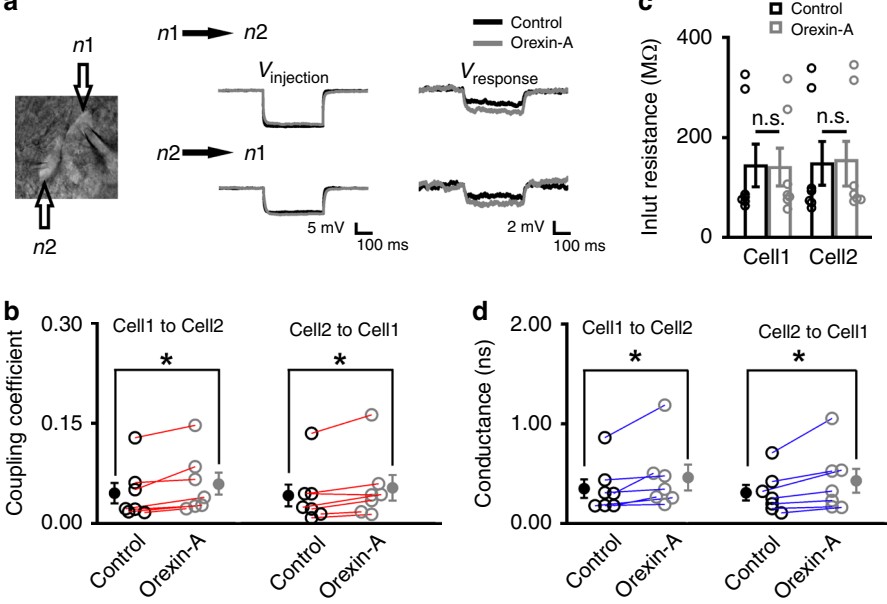

**Fig. 4 Orexin increases the GJ conductance in coupled SLD neurons. a** (Left) Double patch-clamp recordings in a pair of electrically connected SLD neurons in rat. (Right) Representative traces from this electrically connected pair of neurons showing that similar amplitude of voltage change in n1 ($V_{injection}$) induced larger voltage response ($V_{response}$) in n2 in the presence of orexin-A (100 nM), and vice versa. **b** Group data showing that orexin-A bilaterally increased the coupling coefficient from cell 1 to cell 2 (left, two-sided Wilcoxon signed-rank test; $z = 2.366$, $P = 0.0180$) and cell 2 to cell 1 (right, two-sided Wilcoxon signed-rank test; $z = 2.197$, $P = 0.0280$) ($n = 7$ pairs of neurons). **c** Group data showing that the input resistances of cell 1 (left, two-sided Wilcoxon signed-rank test; $z = 0.338$, $P = 0.735$) and cell 2 (right, two-sided Wilcoxon signed-rank test; $z = 1.690$, $P = 0.0910$) were not affected by orexin-A ($n = 7$ pairs of neurons). **d** Group data showing that orexin-A bilaterally increased the GJ conductance from cell 1 to cell 2 (left, two-sided Wilcoxon signed-rank test; $z = 2.366$, $P = 0.0180$) and cell 2 to cell 1 (right, two-sided Wilcoxon signed-rank test; $z = 2.366$, $P = 0.0180$) ($n = 7$ pairs of neurons). Data represent mean ± SEM. *$P < 0.05$. Source data are provided as a Source Data file.

18 of 215 (8.4%) SLD unit pairs showed significant coincident spiking activities. After orexin-A microinjection, this percentage increased to 28 of 215 (13%). Moreover, the coincident probability from the 18 SLD unit pairs with significant interactions in both baseline and orexin-A conditions increased from 1.5 ± 0.2% to 2.4 ± 0.3% ($P = 1.076 \times 10^{-4}$) (Fig. 5e, f). These data suggest that the orexin-elicited excitation and increase in GJ conductance drive synchronized excitation in the SLD network in vivo.

To further assess the contribution of GJ activities in the orexin-elicited synchronized excitation, we tried to use CBX to interfere with the SLD GJs. Although CBX microinjection alone (100 mM, 0.3 µl) only partially reduced the coincident spiking probability of the SLD unit pairs in vivo (Supplementary Fig. 8a–c), we found that CBX pre-injection was able to abolish the orexin-elicited elevation of synchronized spiking and orexin-A still increased the spiking rate in 39 of 72 (54.2%) SLD units (CBX: 5.2 ± 0.9 Hz, CBX/orexin-A: 8.9 ± 1.5 Hz; $P = 5.255 \times 10^{-8}$) (Fig. 5g–i). In this circumstances, orexin-A microinjection after CBX did not change the number of SLD unit pairs with significant interactions (CBX: 7.7%, 14 of 182 pairs; CBX/orexin-A: 7.7%, 14 of 182 pairs) and the coincident probability from 13 pairs with significant interactions in both CBX and CBX/orexin-A conditions (CBX: 2.5 ± 0.3%; CBX/orexin-A: 2.4 ± 0.3%; $P = 0.542$). These data further suggest that the elevated GJ activities are necessarily involved in the orexin-elicited synchronized excitation of the SLD network.

To evaluate changes of the SLD output induced by drug injections, we also simultaneously recorded the hippocampal local field potential (LFP) activities, which largely depend on the SLD output during REM sleep[11]. In urethane-anesthesia, the hippocampal activities were dominated by slow oscillations at ~1 Hz (Supplementary Fig. 9a). We found that microinjections of orexin-A into the SLD increased the power of slow oscillations

(0.3–2.5 Hz) and the phase-locking strength between the SLD spiking activities and these oscillations (Supplementary Fig. 9b–d), suggesting that orexin facilitates SLD output and the enhanced SLD-hippocampus correspondence underlies. Moreover, these effects were all blunted by pre-injection of CBX into the SLD (Supplementary Fig. 9e–l). Collectively, these data suggest that SLD orexin signaling promotes the correspondence between the SLD and its downstream target to enhance the SLD output, and the orexin-elicited synchronized excitation may contribute to this process.

**Activation of SLD orexin signaling prolongs REM sleep episode.** Optogenetics was then employed to examine the contributions of the orexin-enhanced SLD output in brain state regulation. AAV-Ef1α-DIO-ChR2-mCherry was bilaterally injected into the LH of orexin-Cre mice (orexin[ChR2] mice), and optical fibers were implanted in the bilateral SLD (Fig. 6a, b). Through patch-clamp recordings, we found that in orexin[ChR2] mice, optogenetic activation of the SLD orexin terminals excited SLD neurons through the release of orexin[34] (Supplementary Fig. 10a–f). In addition, the functional expression of light-sensitive opsins in the LH orexin neurons after virus infections was also confirmed by immunostaining and patch-clamp methods (Supplementary Fig. 11a–e).

During NREM sleep, 20-Hz light stimulation (473 nm, 20 pulses every 3 s) of the SLD orexin terminals caused an immediate (latency: 2.4 ± 0.3 s) decrease in the total EEG power (baseline: 963.2 ± 196.6 µV$^2$, light: 574.8 ± 186.8 µV$^2$; $P = 5.667 \times 10^{-3}$), followed by a late-onset (latency: 9.4 ± 1.7 s) decrease in EMG amplitude (baseline: 6.6 ± 1.7 µV, light: 5.6 ± 1.6 µV; $P = 3.912 \times 10^{-3}$) unless disrupted by transitions to wakefulness before the light-off ($n = 6$ orexin[ChR2] mice) (Fig. 6c–f). The EEG

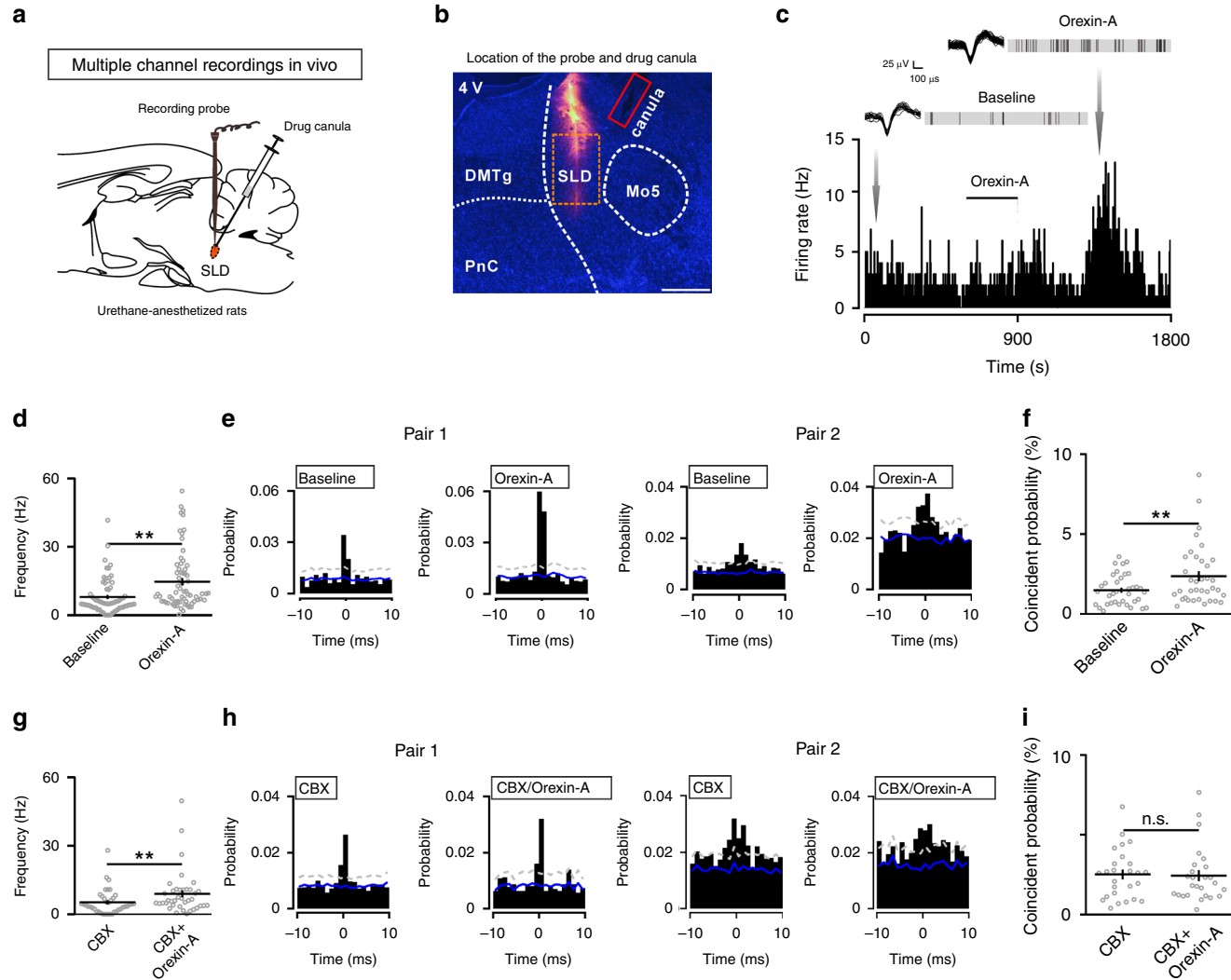

**Fig. 5 Orexin increases synchronized spiking activities in the SLD in vivo. a**, **b** Schematic drawing (**a**) and representative locations (**b**) of electrode arrays (pre-covered by DiI, red)/drug canulas (red rectangle) for the SLD multiple channel recordings in urethane-anesthetized rats in vivo. Scale bar: 500 μm. **c** Orexin-A (30 μM, 0.3 μl) microinjection increased the firing frequency of a SLD unit in vivo. Overlaid spike waveforms and raster plot of spikes in 10-s epochs before and after orexin-A application were expanded. **d** Orexin-A significantly increased the firing frequency of the SLD units ($n = 67/97$ units, two-sided Wilcoxon signed-rank test; $z = 7.115$, $P = 1.120 \times 10^{-12}$). **e** Two example cross-correlograms with significant coincidental spiking within ±1 ms before and after orexin-A microinjections from the rat in **c**. Note that GJ-mediated interactions occurred either within ±1 ms (pair 1) or ±5 ms (pair 2). We focused on the coincident activities within ±1 ms. A significant interaction was counted if a pairwise 1% threshold (dotted gray lines) was crossed. The mean of the chance coincident (solid blue line) was computed from 30 shuffled cross-correlograms. **f** Orexin-A significantly increased the coincident spiking probability of the SLD unit pairs with significant interactions in both baseline and orexin-A conditions ($n = 18/215$ pairs, two-sided Wilcoxon signed-rank test; $z = 3.881$, $P = 1.076 \times 10^{-4}$). Only unit pairs between the adjacent recording sites were analyzed. **g** Orexin-A (30 μM, 0.3 μl) microinjection still increased the firing frequency of SLD neurons after the pre-injection of CBX (100 mM) ($n = 39/72$ units, two-sided Wilcoxon signed-rank test; $z = 5.442$, $P = 5.255 \times 10^{-8}$). **h** Two example cross-correlograms with significant coincidental spiking within ±1 ms before and after orexin-A microinjection in the presence of CBX. **i** Orexin-A failed to increase the coincident spiking probability of the SLD unit pairs with significant interactions in both baseline and orexin-A conditions in the presence of CBX ($n = 13/182$ pairs, two-sided Wilcoxon signed-rank test; $z = 0.622$, $P = 0.542$). Data represent mean ± SEM. **$P < 0.01$. Source data are provided as a Source Data file.

theta component (theta/total power) increased before the disruption of wakefulness (baseline: 0.19 ± 0.02, light: 0.23 ± 0.02; $n = 6$ orexin$^{ChR2}$ mice; $P = 0.0277$) (Fig. 6g), suggesting an activation of brain state in this period. In the control group expressing only mCherry (orexin$^{mCherry}$ mice), light (20 Hz) stimulation did not affect the total EEG power (baseline: 836.9 ± 126.2 μV$^2$, light: 870.2 ± 142.9 μV$^2$; $P = 0.448$) or EMG amplitude (baseline: 5.6 ± 1.5 μV, light: 5.5 ± 1.5 μV; $P = 0.640$) ($n = 5$ orexin$^{mCherry}$ mice) (Fig. 6e, f). The activation of brain state followed by a decrease of EMG amplitude also occurred during physiological NREM to REM sleep transitions. We thus analyzed

the EEG power spectrogram during the light stimulation (20-Hz) and found that consecutive theta oscillations characterizing REM sleep were not induced (Supplementary Fig. 12a). Moreover, the latency to REM sleep was not changed by light (Supplementary Fig. 13a). Together, these data demonstrate that the activation of SLD orexin signaling does not induce REM sleep transitions, but indeed causes the activation of brain state with increased EEG theta component and muscle tone decrease. In NREM sleep, this activation may cause a short awakening effect. Although the latency from NREM sleep to wakefulness was also not significantly affected, the NREM sleep to wakefulness transition

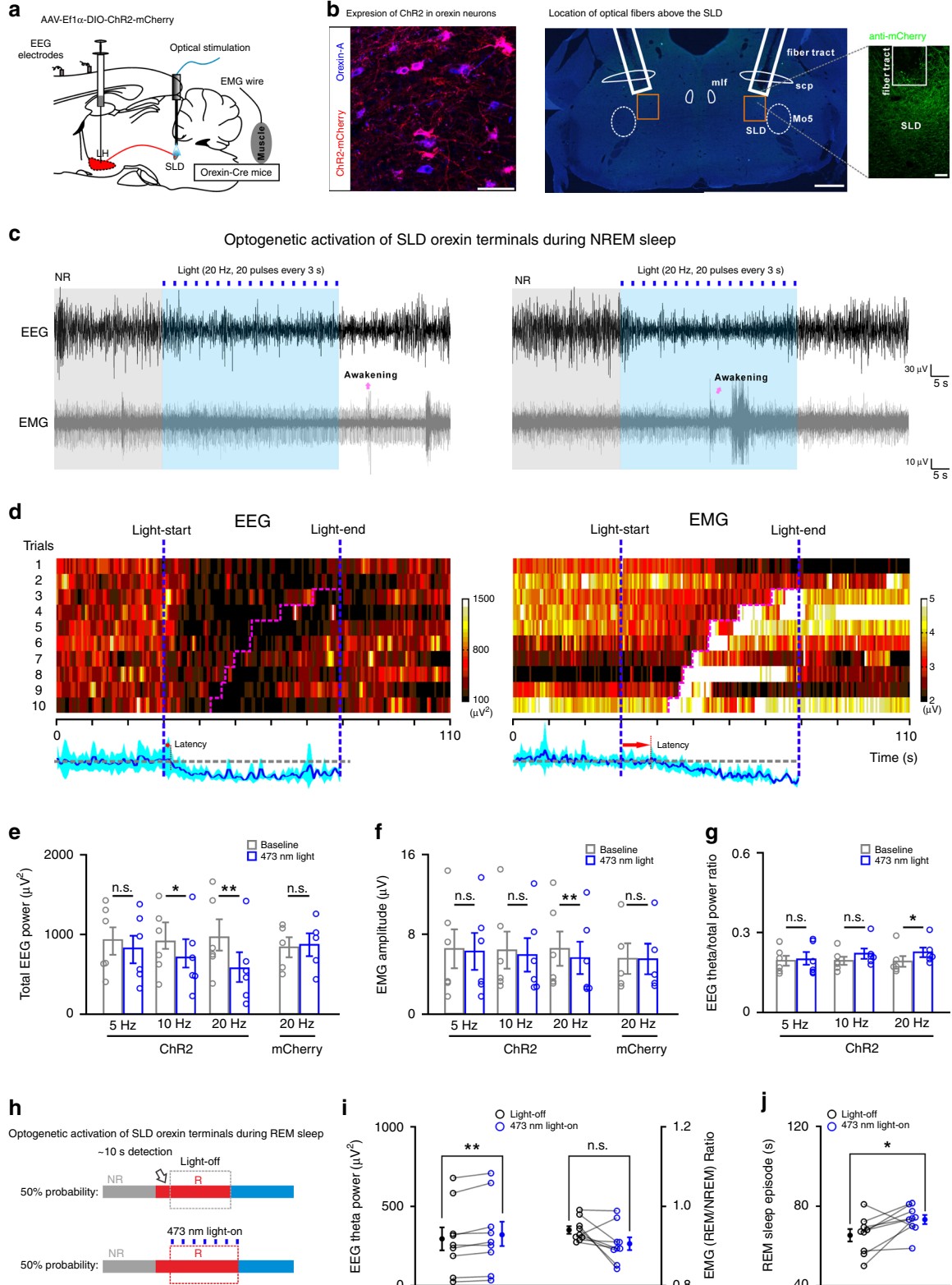

probability slightly increased between ~20 and ~30 s after light delivery (Supplementary Fig. 13b–e). Increasing the stimulation intensity (20 Hz for 20 s) still only caused the brain state activation during NREM sleep, but failed to induce REM sleep transitions. In this condition, the transition probability from NREM sleep to wakefulness was high (Supplementary Fig. 14a–d).

We next examined whether the brain state activation and muscle tone decrease induced by SLD orexin signaling contribute to REM sleep regulation. Optogenetic activation of the SLD orexin terminals was applied during individual REM sleep episodes with 50% probability, and the REM sleep performances and length were compared between the light-on and light-off trials (Fig. 6h). In REM sleep, light stimulation (473 nm, 20 Hz,

**Fig. 6 Optogenetic activation of SLD orexin signaling prolongs REM sleep bouts. a** Schematic drawing for optogenetic activation of the SLD orexin terminals. **b** Representative viral infections of ChR2-mCherry$^+$ (red) in the LH orexin-A$^+$ (blue) neurons (left, scale bar: 50 μm) and optical-fiber location above the SLD (right, scale bars: 500 and 50 μm in magnification). **c** Representative trials of raw EEG/EMG recordings during 20-Hz optical activation (20 pulses every 3 s) of SLD orexin signaling during NREM sleep. Purple arrows indicated wakefulness. **d** Heatmap of EEG (left)/EMG (right) recordings from all tested trials in the mouse in (**c**). EEG/EMG recordings before wakefulness (purple line) were normalized and averaged in the bottom. Blue shadow represented SEM. **e**, **f** Effects of optical activation during NREM sleep on the total EEG power (**e**, 5-Hz: $t_5 = 2.442$, $P = 0.0585$; 10 Hz: $t_5 = 3.939$, $P = 0.0110$; 20 Hz: $t_5 = 4.633$, $P = 5.667 \times 10^{-3}$) and EMG amplitude (**f**, 5 Hz: $t_5 = 1.610$, $P = 0.168$; 10 Hz: $t_5 = 2.180$, $P = 0.0811$; 20 Hz: $t_5 = 5.056$, $P = 3.912 \times 10^{-3}$) ($n = 6$ orexin$^{ChR2}$ mice). Orexin$^{mCherry}$ mice was set as control ($n = 5$ mice; EEG (20 Hz): $t_4 = 0.840$, $P = 0.448$; EMG (20 Hz): $t_4 = 0.506$, $P = 0.640$). All analyzed by two-sided paired $t$-test. **g** Effects of optical activation during NREM sleep on the EEG theta/total power ratio ($n = 6$ orexin$^{ChR2}$ mice, two-sided Wilcoxon signed-rank test; 5 Hz: $z = 0.105$, $P = 0.917$; 10 Hz: $z = 1.572$, $P = 0.116$; 20 Hz: $z = 2.201$, $P = 0.0277$). **h** Procedures of closed-loop optogenetics during REM sleep. **i**, **j** Effects of optical activation (20 Hz, 20 pulses every 3 s) during REM sleep on the EEG theta power (**i**, $t_8 = 3.381$, $P = 9.624 \times 10^{-3}$), EMG$_{REM/NREM}$ ratio (**i**, $t_8 = 2.126$, $P = 0.0662$), and episode duration (**j**, $t_8 = 2.368$, $P = 0.0454$) ($n = 9$ orexin$^{ChR2}$ mice). All analyzed by two-sided paired $t$-test. Data represent mean ± SEM. *$P < 0.05$; **$P < 0.01$. Source data are provided as a Source Data file.

20 pulses every 3 s) further consolidated the EEG theta oscillations (theta power; light-off: $295.3 \pm 72.6$ μV$^2$, light-on: $321.1 \pm 77.0$ μV$^2$; $n = 9$ orexin$^{ChR2}$ mice); $P = 9.624 \times 10^{-3}$) (Fig. 6i). In addition, to eliminate biased EMG modifications induced by posture changes of different sleep epochs[35], the EMG$_{REM/NREM}$ ratio was used to examine changes of muscle atonia. In this condition, a decreasing trend in the EMG$_{REM/NREM}$ ratio was also observed (EMG$_{REM/NREM}$ ratio; light-off: $0.94 \pm 0.01$, light-on: $0.91 \pm 0.02$; $n = 9$ orexin$^{ChR2}$ mice; $P = 0.0662$) (Fig. 6i). Consequently, optical activation significantly prolonged the REM sleep episode duration by 12.1% (light-off: $65.4 \pm 3.1$ s, light-on: $73.3 \pm 2.3$ s; $n = 9$ orexin$^{ChR2}$ mice; $P = 0.0454$) (Fig. 6j). All these effects were not observed in the mice expressing only mCherry (Supplementary Fig. 15a–c). These data suggest that SLD orexin signaling prolonged REM sleep episodes through consolidating the brain state activation and muscle tone decrease, and may thereby, contribute to REM sleep stabilization.

**Silencing of SLD orexin signaling destabilizes REM sleep.** The contributions of SLD orexin signaling in the maintenance of REM sleep episodes were then examined. To observe the activities of SLD orexin signaling during REM sleep, AAV-CAG-FLEX-jGCaMP7b was injected into the LH of the orexin-Cre mice, followed by implantation of optical fibers above the SLD (Fig. 7a, b). Intriguingly, jGCaMP fluorescence changes ($\Delta F/F$) of the SLD orexin terminals were observed during the REM sleep episodes ($0.26 \pm 0.03\%$), which was significantly higher than that ($0.02 \pm 0.01\%$) of the preceding NREM sleep episodes and lower than that ($0.89 \pm 0.20\%$) of the following wakefulness episodes ($n = 7$ mice; one-way repeated-measures ANOVA; $F_{(2, 12)} = 16.758$; $P = 3.358 \times 10^{-4}$; post hoc LSD comparison test; REM vs. NREM: $P = 2.481 \times 10^{-4}$; REM vs. wakefulness: $P = 0.0127$; NREM vs. wakefulness: $P = 4.563 \times 10^{-3}$) (Fig. 7c, d), suggesting that the activity of SLD orexin signaling increased during natural REM sleep episodes. In addition, the jGCaMP signals elevated throughout the entire period, including both phasic and tonic REM sleep (Supplementary Fig. 16a). We thus performed optogenetic inhibition of SLD orexin signaling during REM sleep. AAV-CAG-FLEX-ArchT-GFP was bilaterally injected into the LH of orexin-Cre mice (orexin$^{ArchT}$ mice) followed by implantation of optical fibers bilaterally in the SLD (Fig. 7e). The effectiveness of optogenetic inhibition was validated by immunostaining and patch-clamp methods (Supplementary Fig. 11a–d). Optogenetic inhibition consistently decreased the theta oscillation power (light-off: $405.0 \pm 61.0$ μV$^2$, light-on: $375.1 \pm 58.3$ μV$^2$; $P = 0.0325$), and the REM sleep episode duration (light-off: $76.2 \pm 4.4$ s; light-on: $62.3 \pm 4.1$ s; $P = 0.0363$) ($n = 9$ orexin$^{ArchT}$ mice) (Fig. 7f, g). Changes in the EMG$_{REM/NREM}$ ratio (light-off: $0.91 \pm 0.02$, light-on: $0.90 \pm 0.01$; $n = 9$ orexin$^{ArchT}$ mice; $P = 0.320$; Fig. 7f) were not detected by this inhibition. These data thus suggest

that SLD orexin signaling is involved in the maintenance of REM sleep episodes.

To silence SLD orexin signaling on an hourly time scale, AAV-retro-hSyn-DIO-hM4D (Gi)-mCherry (mixed with AAV-hSyn-EGFP for histological analysis) was bilaterally injected into the SLD of the orexin-Cre mice (orexin$^{SLD-hM4D}$ mice) (Fig. 8a–c). The effectiveness of chemogenetic manipulation was confirmed by immunostaining and patch-clamp methods (Supplementary Fig. 17a–d). Clozapine N-oxide (CNO, 3 mg/kg) or vehicle (0.9% NaCl) was injected intraperitoneally (i.p.) in the orexin$^{SLD-hM4D}$ mice at ZT 4 (12:00 a.m.), and the brain states were monitored for the next 4 h. Compared to vehicle, CNO caused a 17.0% reduction in the total REM sleep amount (vehicle: $8.8 \pm 1.2\%$, CNO: $7.3 \pm 1.2\%$; $P = 6.170 \times 10^{-3}$), and the amount of wakefulness (vehicle: $37.3 \pm 2.7\%$, CNO: $39.7 \pm 2.7\%$; $P = 0.217$) or NREM sleep (vehicle: $53.9 \pm 2.7\%$, CNO: $53.1 \pm 2.9\%$; $P = 0.615$) was not affected ($n = 8$ orexin$^{SLD-hM4D}$ mice) (Fig. 8d, e). The reduced REM sleep amount was due to a decrease in the episode duration (vehicle: $51.3 \pm 5.1$ s, CNO: $40.0 \pm 2.7$ s; $P = 0.0269$), whereas the episode number was not altered (vehicle: $25.3 \pm 3.5$, CNO: $25.3 \pm 3.3$; $P = 1.000$) ($n = 8$ orexin$^{SLD-hM4D}$ mice) (Fig. 8f). We also assessed the regulation of SLD orexin signaling in REM sleep through pharmacological manipulation in rats (Supplementary Fig. 18a, b). Similarly, through microinjection of orexin-A or TCS 1102 in the SLD, we found that SLD orexin signaling increased the REM sleep amount via prolonging the episode duration (Supplementary Fig. 18a, b). These data demonstrate a routinely involved role of SLD orexin signaling in the stabilization of REM sleep.

Intriguingly, through EEG/EMG analysis in REM sleep episodes of the entire 4-h recordings, a general failure of muscle atonia was obviously observed after CNO injections in all tested orexin$^{SLD-hM4D}$ mice. The physiological muscle tone decrease from NREM to REM sleep (EMG$_{REM/NREM}$ ratio) was abolished (vehicle: $93.4 \pm 1.6\%$; CNO: $99.0 \pm 1.5\%$; $n = 8$ orexin$^{SLD-hM4D}$ mice; $P = 8.518 \times 10^{-3}$) (Fig. 8g–i). In this case, an abnormal behavior phenotype characterized by severe disruption of muscle atonia was observed after constructing the EMG activity map (Fig. 8h). These observations may explain that more than half of narcoleptic patients suffer from failure of REM sleep paralysis[36]. Moreover, the theta oscillation power (vehicle: $409.3 \pm 52.3$ μV$^2$; CNO: $372.9 \pm 55.6$ μV$^2$; $n = 8$ orexin$^{SLD-hM4D}$ mice; $P = 0.0484$) (Fig. 8j) in REM sleep were also decreased after silencing of SLD orexin signaling, and the power distribution remained unchanged (Supplementary Fig. 19a, b). All these effects were not observed in the control mice expressing only mCherry (Supplementary Fig. 20a, b). Together, the disruption of these core features in REM sleep behavior after the loss of SLD orexin signaling may eventually destabilize individual REM sleep episodes, and thus reduce

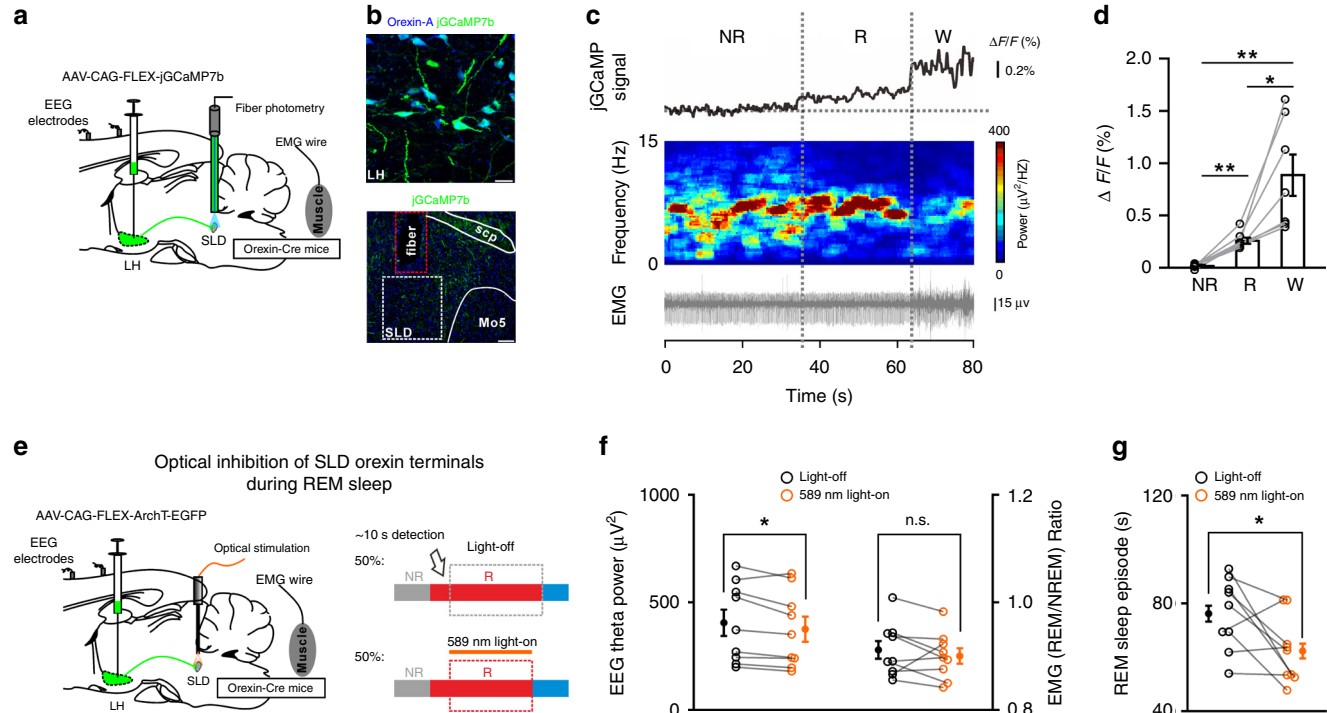

**Fig. 7 Optogenetic inhibition of SLD orexin signaling shortens REM sleep bouts. a** Schematic drawing of fiber photometry recordings in the SLD orexin terminals. **b** Representative viral infections of jGCaMP7b (green) in the LH orexin-A+ neurons (blue) (top, scale bar: 25 μm), and optical fiber location above the SLD orexin terminals (bottom, scale bar: 100 μm). **c** Representative jGCaMP fluorescence traces of the SLD orexin terminals, spectrogram of EEG recordings, and raw EMG recorded simultaneously in a episode of REM sleep (R), the preceding NREM sleep (NR), and the following wakefulness (W). **d** Averaged changes of jGCaMP fluorescence in REM sleep, the preceding NREM sleep, and the following wakefulness ($n = 7$ mice; one-way repeated-measures ANOVA; $F_{(2, 12)} = 16.758$, $P = 3.358 \times 10^{-4}$; post hoc LSD comparison test; REM vs. NREM: $P = 2.481 \times 10^{-4}$; wake vs. NREM: $P = 4.563 \times 10^{-3}$; REM vs. wake: $P = 0.0127$). **e** Schematic drawing for optogenetic inhibition of the SLD orexin terminals (left) and procedures for closed-loop optogenetic inhibition during REM sleep (right). The EEG/EMG signals were visually inspected on-line. Lasers were manually turned on with 50% probability after the detection of REM sleep (~10 s), and turned off when the REM sleep episode ended. **f** Effects of optical inhibition of the SLD orexin terminals during REM sleep on the EEG theta power (left, two-sided paired *t*-test; $t_8 = 2.582$; $P = 0.0325$) and EMG$_{REM/NREM}$ ratio (right, two-sided paired *t*-test; $t_8 = 1.060$; $P = 0.320$) ($n = 9$ orexin$^{ArchT}$ mice). **g** Effects of optical inhibition of the SLD orexin terminals during REM sleep on the episode duration of REM sleep ($n = 9$ orexin$^{ArchT}$ mice, two-sided paired *t*-test; $t_8 = 2.512$; $P = 0.0363$). Data represent mean ± SEM. *$P < 0.05$; **$P < 0.01$. Source data are provided as a Source Data file.

REM sleep amount, further suggesting the essential role of SLD orexin signaling in REM sleep stabilization.

## Discussion

The present study demonstrated a direct role of SLD orexin signaling in REM sleep stabilization. The REM sleep relevance was firstly identified in a sub-cluster of orexin neurons, which constituted a specific efferent node of orexin signaling to the glutamatergic neuron-enriched SLD region. We next observed the electrophysiological modulations of orexin on the SLD neurons. Intriguingly, in addition to the reported excitatory effects in the SLD neurons[14,17], orexin also actively increased the GJ conductance among the SLD network. These parallel actions interacted to restrain the SLD neuronal activities towards a synchronized excitation pattern. Moreover, the SLD output was consequently enhanced to induce brain state activation and muscle tone decrease in vivo. During REM sleep, this mechanism contributed to consolidating the EEG theta-band activities and muscle atonia. In this way, the stabilization of REM sleep was achieved under physiological conditions.

Orexin neurons innervate multiple neuronal circuitries controlling sleep/wakefulness states[7]. In general, many wakefulness-promoting structures receive abundant orexin innervations[6]. It thus readily leads to assumptions of secondary effects on sleep by

orexin signaling[1]. However, accumulating evidence has shown orexin innervations in sleep-promoting regions[7,15,37]. The orexin innervations have been observed in the SLD by previous reports and the present study[6,15]. We further found that the OXRs were abundantly expressed on the SLD glutamatergic neurons or REM-on neurons. These findings rationally suggest an alternatively direct effect of SLD orexin signaling in REM sleep regulation. Moreover, the observed characteristics of the OX$^{SLD}$ neurons were also consistent with direct REM sleep regulation. These neurons were sporadically distributed and distinct from the OX$^{LC}$ neurons, even though the LC was adjacent to the SLD. These findings support notions that orexin neurons can be classified into different groups based on collateral downstream projection patterns[38]. Actually, it has long been suspected that orexin neurons are divided into distinct groups, according to their roles in seemingly disparate behaviors, such as arousal, reward seeking, and motor control[39,40]. A group of REM sleep-related orexin neurons may also exist, as micro-dialysis has reported that orexin release increased during REM sleep[41]. Intriguingly, with projection-specific tagging, REM sleep-related activation emerged in the relatively minor (~7%) OX$^{SLD}$ class. Furthermore, by using fiber photometry on regional SLD orexin terminals, their increased activities during REM sleep was also observed in the present study. Another interesting finding of the c-Fos screening was that the OX$^{SLD}$ neurons were activated against the generally

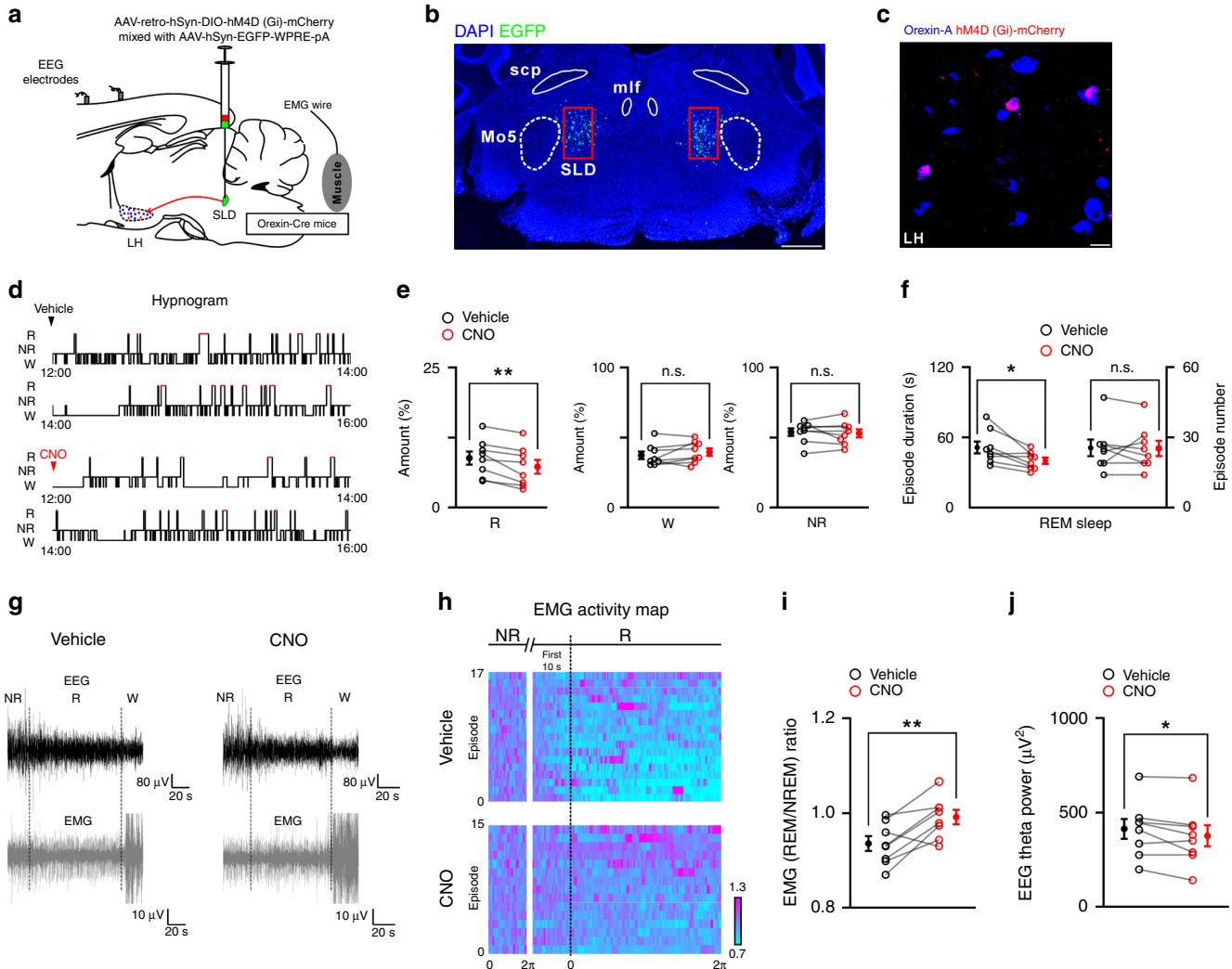

**Fig. 8 Chemogenetic silencing of SLD orexin signaling destabilizes REM sleep. a** Schematic drawing for chemogenetic silencing of SLD orexin signaling. **b** Representative virus injection sites in the SLD (scale bar: 500 μm). mlf medial longitudinal fasciculus. **c** Representative expression of hM4D (Gi)-mCherry (red) in the orexin-A+ (blue) neurons. Scale bar: 20 μm. **d** Hypnogram showing the sleep/wakefulness states in the following 4 h after vehicle and CNO injections in a tested orexin$^{SLD-hM4D}$ mouse. **e** The amount of REM sleep (two-sided paired t-test; $t_7 = 3.865$, $P = 6.170 \times 10^{-3}$), wakefulness (two-sided paired t-test; $t_7 = 1.355$, $P = 0.217$), and NREM sleep (two-sided paired t-test; $t_7 = 0.527$, $P = 0.615$) during the 4-h recording period after CNO and vehicle injections ($n = 8$ orexin$^{SLD-hM4D}$ mice). **f** The episode duration (left, two-sided paired t-test; $t_7 = 2.789$, $P = 0.0269$) and number (right, two-sided paired t-test; $t_7 = 0$, $P = 1.000$) of REM sleep during the 4-h recording period after CNO and vehicle injections ($n = 8$ orexin$^{SLD-hM4D}$ mice). **g** Representative raw EEG/EMG activities in REM sleep, the preceding NREM sleep, and the following wakefulness after vehicle and CNO injections in a tested orexin$^{SLD-hM4D}$ mouse. Note that a prominent failure of muscle atonia in REM sleep occurred after CNO injection. **h** EMG$_{REM/NREM}$ activity map after vehicle (top) and CNO (bottom) injections from all REM sleep episodes and the preceding NREM sleep in a orexin$^{SLD-hM4D}$ mice. These episodes are sorted vertically from the highest (purple) to the lowest (blue) mean EMG$_{REM/NREM}$ ratio. The x-axis represents the episode duration normalized between 0 and $2\pi$. Note that CNO injection disrupted the muscle tone decrease from NREM to REM sleep, and failure of muscle atonia occurred in REM sleep. **i, j** Changes of EMG$_{REM/NREM}$ ratio (**i**, two-sided paired t-test; $t_7 = 3.619$, $P = 8.518 \times 10^{-3}$) and EEG theta power (**j**, two-sided paired t-test; $t_7 = 2.387$, $P = 0.0484$) after CNO and vehicle injections ($n = 8$ orexin$^{SLD-hM4D}$ mice). Data represent mean ± SEM. *$P < 0.05$; **$P < 0.01$. Source data are provided as a Source Data file.

silent background of the OX$^{entirety}$ during REM sleep. This phenomenon may also account for the conflicting results from previous reports[37]. Based on extracellular recordings from indiscriminate orexin neurons, the firing activities of them in REM sleep have been described as intermediate, low, or even silent[42–44].

The elevated activities of the OX$^{SLD}$ neurons in REM sleep indicate their essential roles on the SLD neurons. Intriguingly, following the activation of orexin neurons, complex modulations have been reported in different sleep/wakefulness-related structures, such as the suprachiasmatic nuclei (SCN) and the tuberomammillary nucleus (TMN)[45–48]. Importantly, we found that

the SLD GJs provided specific conditions for orexin to elicit a more dedicated modulation pattern, rather than just excitation[17]. CBX-sensitive spikelet activities and functional Cx-36 expression were both abundantly observed in the SLD neurons, suggesting possibly extensive electrical connections in the SLD network. With the aid of GJs, the orexin-elicited direct and partial excitation could spread from the OXRs+ SLD neurons (occupying ~3/4 of the total population) to affect the large-scale SLD network. The substrates exchanged in this process are still unclear, as GJs are reciprocal pathways for either ionic current or small molecules[49]. Alternatively, we showed that the electrophysiological influences generated by exchange may largely affect

the SLD neuronal activity pattern. In detail, the propagation of orexin-elicited direct excitation, including the sub-threshold depolarization and depolarization-evoked spikes, caused two forms of consequences. On the one hand, the spread of sustained depolarization via GJs alone may increase the effect range[50]. The OXRs⁻ SLD neurons can thus be recruited in the excitation of orexin. This phenomenon was reflected by the reduced number of orexin-excited neurons after blocking GJs by CBX/MEF and the unchanged excitation amplitude after blocking spike generation by TTX. On the other hand, the activation-evoked spikes would form another discrete spread source, increasing the spike to spikelet transmission[32]. We indeed observed increased spikelet activities after orexin applications in the SLD. Intriguingly, spikelet activities are thought to be the basis for synchronized firings[32,49,50]. Therefore, orexin elicited widespread activation in the electrically connected SLD network, and the activation-evoked firings may tend to be synchronized.

The significance of orexin-elicited synchronized excitation on the SLD output is also demonstrated. The increased SLD firings promote several types of REM sleep physiology, such as ponto-geniculo-occipital (PGO) waves, hippocampal theta activities and muscle atonia[1,11]. The global excitation of orexin may thus enhance the SLD output to regulate REM sleep. In fact, the regulation of firing frequency in the SLD neurons appears to be a common way to influence REM sleep[19,21,22]. Intriguingly, it has been assumed that the synchrony of SLD firings constitutes another essential force for enhancing the output[26–28]. This phenomenon is directly evidenced here, as the orexin-elevated synchronized SLD firings were more effectively phase-locked to the hippocampal oscillations (0.3–2.5 Hz) in anesthesia, and consequently, increased its power. Furthermore, disruption of the orexin-induced synchronized excitation in the SLD by CBX blunted all these effects. Therefore, both the orexin-elicited excitation and firing synchrony are essentially involved in the SLD output.

Considering that the SLD output with synchronized information may be fundamental for recruiting coherence activities in REM sleep-related brain regions[28], the orexin-elicited synchronized excitation may contribute to all processes involved in REM sleep, such as the core features, e.g., corticohippocampal activation and muscle atonia. Actually, for the corticohippocampal theta oscillations, several reports have indeed provided the anatomical basis for the orexin-enhanced SLD output to influence them. The neural pathways may include ascending projections from the SLD to the intralaminar thalamus[12], the precoeruleus region[11], or the hippocampus[51]. In addition, muscle atonia can also be influenced through descending projections from the SLD to the gigantocellular reticular nucleus or the spinal cord[19,35]. Consistently, we observed brain state activation with an increased theta component and muscle tone decrease after optogenetic activation of SLD orexin signaling in free-moving animals. Moreover, insufficiencies of EEG theta oscillation and muscle atonia were observed after chemogenetic inhibition of SLD orexin signaling. Notably, alterations in EEG theta oscillations and failure of muscle atonia are frequently reported during REM sleep in narcoleptic patients[8–10]. Given the tight relationship between EEG theta oscillation/muscle atonia and REM sleep stability[1], the orexin-enhanced SLD output seems to be essentially involved in REM sleep stabilization. We consequently found that the REM episode duration was increased by optogenetic activation and decreased by optogenetic/chemogenetic inhibition of SLD orexin signaling. Thus, a decrease in REM sleep amount was observed after prolonged pharmacological/chemogenetic silencing of this signaling. These findings provide evidence that the orexin-enhanced SLD output and the consequent brain state

activation/muscle tone decrease contribute to REM sleep stabilization under physiological conditions.

In summary, the maintenance of REM sleep, another active brain state in addition to wakefulness, also requires excitatory orexin signaling. In fact, it has been reported that multiple projection sites of orexin signaling help to fulfill its diverse physiological roles[6,34,40]. In this condition, the present study rationally suggests that the orexin projections to the SLD stabilizes REM sleep, expanding previously reported findings that its projections to wakefulness-promoting nuclei mediate arousal[52–56]. Our findings provide additional insights to understand that both REM sleep symptoms and wakefulness-maintaining deficits exist after loss of central orexin signaling[9,10,36]. Intriguingly, based on the demands of internal and external states, the orexin neuronal entirety is thought to operate as an integrator in the homeostatic regulation of diverse physiological functions[10,40,57]. For integral sleep/wakefulness cycling behavior, loss of orexin signaling indeed causes severely disrupted arrangements of vigilance states, while the total time of the individual state remains unchanged[58]. Therefore, the fine division of the orexin neuronal entirety in wakefulness and REM sleep stabilization may work together to orchestrate this basic brain function.

## Methods

**Animals.** Sprague-Dawley rats (Laboratory Animal Center, Third Military Medical University) were used for experiments in Figs. 1–5 and Supplementary Figs. 1–9 and 18. To selectively manipulate SLD orexin signaling in behavioral experiments, orexin-Cre mice (gift from L. de Lecea, Stanford University, USA)[59] and Ai27D mice (012567, Jackson Laboratory, USA) were used for experiments in Figs. 6–8 and Supplementary Figs. 10–17 and 19, 20. All animal care and experimental procedures were approved by the Guide for the Care and Use of Laboratory Animals of the Third Military Medical University. Animals were housed in 12-h light/dark cycle, with lights-on at 8:00 a.m. (ZT 0) and lights-off at 8:00 p.m. (ZT 12). The environment temperature was kept constant at 22 ± 1 °C and the relative humidity was kept between 40% and 60%. Food and water were available ad libitum.

**CTB retrograde tracing.** Male rats (250–300 g) were anesthetized with sodium pentobarbital (75 mg/kg, i.p.) and then fixed on a stereotaxic apparatus (RWD Life Science, China). For this and the following experiments in rats, we focused on the SLD region from AP: −9.20 to −9.80 mm according to the rat brain atlas[60], as previous studies reported that this region was enriched of glutamatergic neurons that were responsible for REM sleep generation[18]. To target this region, CTB-488 (69 nl, 5 µg/µl) was injected into the SLD unilaterally (bregma: AP: −9.50 mm, ML: −1.30 mm, DV: −8.20 mm) (Supplementary Fig. 2a). Besides, CTB-555 (69 nl, 5 µg/µl) was injected into the LC (AP: −9.60 mm, ML: −1.40 mm, DV: −7.20 mm), ipsilaterally to the SLD CTB-488 injection. CTB tracers were injected using Nanoject II (Drummond Scientific, USA) via a silicate-glass micro-pipette (tip diameter: ~20 µm). Multiple 2.3 nl injections were made at 10-s intervals and the pipette was left in place for additional 10 min after the injections. At least 2 weeks after injections, rats were sacrificed and coronal brain slices containing the LH and the SLD/LC were collected for following immunohistological processes.

**REM sleep deprivation.** Rats with CTB-488 injection in the unilateral SLD were subjected to RSD through employing the well-accepted inverted flowerpot technique. Briefly, rats were placed on a small-round platform (diameter: 6.5 cm) surrounded by water (1 cm under the platform) at 10:00 a.m. to prevent REM sleep, but not NREM sleep[18]. Rats were available to food and water ad libitum and kept in the 12-h light/dark cycles. After 72-h RSD, rats were removed from the platform at 10:00 a.m. and were placed to their original cages to allow REM sleep recovery. EEG and EMG recordings were performed to monitor the sleep/wakefulness states. Another group of rats that remained in their home cages throughout the experiment were set as control. All rats were anesthetized and sacrificed after 2.5 h from the first REM sleep episode in recording sessions. The coronal brain sections containing LH and SLD were collected for further c-Fos immunostaining and histological processes, respectively.

**Immunohistochemistry.** Rats or mice were deeply anesthetized with pentobarbital sodium and transcardially perfused with saline, followed by 4% paraformaldehyde. Brains were fixed in 4% paraformaldehyde and kept in 30% sucrose/PBS at 4 °C. Coronal slices of 10–20 µm were made for immunohistochemical staining, using a vibratome (CM 3050S, Leica, Germany). Slices for immunostaining were sequentially incubated in the primary and secondary antibodies following the instructions. The primary antibodies used in the present study were as follows, goat anti-orexin-

A (1:500, sc-8070, Santa Cruz, USA), rabbit anti-c-Fos (1:1000, ABE-457, Millipore, USA), mouse anti-c-Fos (1:1000, AB208942, Abcam, USA), rabbit anti-orexin 1/2R (1:500, bs-1095R, Bioss, China), mouse anti-Cx-36 (1:1000, sc-398063, Santa Cruz, USA), mouse-anti-NeuN (1:1000, MAB377, Millipore, USA), guinea-pig-anti-VGLUT1 (1:1000, AB 5905, Millipore, USA), mouse anti-GAD 67 (1:500, MAB5406, Millipore, USA), rabbit anti-glutamate (1:500, G6642, Sigma, USA), rabbit anti-GABA (1:1000, A2052, Sigma, USA), and rabbit anti-MCH (1:2000, H-070-47, Phoenix Pharmaceuticals, USA). Images were acquired with a LSM 800 (Carl Zeiss, Germany) and analyzed by Zen software 2012 (Carl Zeiss, Germany). Histological processes were also performed to locate the sites of CTB/virus injections and drug canula/electrode/optical fiber implantations. Data were excluded if the locations were not correct.

**Cell counting**. To quantify the SLD-projecting or LC-projecting orexin neurons, six coronal sections of each CTB-injected rat containing the LH with a distance of 200 μm (between AP: −2.30 and −3.80 mm) were collected. After orexin-A immunostaining, we counted the number of total orexin neurons, CTB-488-labeled orexin neurons (SLD-projecting), CTB-555-labeled orexin neurons (LC-projecting), and orexin neurons labeled by both CTB tracers (SLD/LC-projecting). The ratio of each group was then reported. To examine the REM sleep-related activities of OX$^{SLD}$ neurons, c-Fos immunostaining was further applied in rats with CTB-488 injection after recovery from RSD and the control rats. The counting method was similar.

**EEG/EMG recordings and drug injection in free-moving rats**. Rats which underwent EEG/EMG recording were chronically implanted with EEG/EMG electrodes. Two EEG electrodes were implanted at the frontal and entorhinal cortex region. Two EMG electrodes were placed between the neck musculature. Two drug canulas were implanted bilaterally above the SLD (bregma: AP: −9.50 mm, ML: −1.70 mm, DV: −7.80 mm) for drug injections in the pharmacological tests. All electrodes were soldered to a micro-pin connector, and then affixed to the skull with dental cement. Before tests, rats were allowed to recover within their home cages for at least 7 days and acclimated in the recording cage for 2 days. The EEG/EMG signals were recorded and stored for further analysis using omniplex neural data acquisition system (PlexControl 1.10, Plexon, USA). The EEG/EMG signals were digitized at 1000 Hz and band-pass filtered (EEG: 1–30 Hz, EMG: 20–100 Hz). EEG/EMG signals were on-line monitored and off-line analyzed by two investigators who are blind to treatments based on spectral signatures of EEG-EMG waveforms in 5-s epochs. Wakefulness was defined as desynchronized, low-amplitude EEG rhythms and elevated EMG activity with phasic bursts. NREM sleep was defined as synchronized, high amplitude and low frequency (1–4 Hz, delta) EEG activity and lower EMG activity compared with wakefulness. REM sleep was defined as containing consecutive theta (6–9 Hz) oscillations with further decreased EMG activity compared with the preceding NREM sleep. In pharmacological tests, drugs were injected into SLD by a micro-syringe (filled with drugs) connected to a micro-drive (KD310, KD Scientific, USA). Drugs were delivered in a random pattern with 2 days interval at a speed of 0.06 μl/min between ZT4 (11:00 a.m.) and ZT5 (12:00 a.m.). The percentage of time spent in each state in the following hour after drug injection was reported, as drug effects returned to baseline level after the first post-injection hour.

**Whole-cell patch-clamp recording**. Coronal brainstem slices containing the SLD (300–400 μm) were prepared with a vibroslicer (VT 1200S, Leica, Germany) from rats aged 9–14 days. During recordings, the slices were continuously superfused with 95% $O_2$ and 5% $CO_2$ oxygenated ACSF (composition in mM: 125 NaCl, 2.5 KCl, 1.25 NaH$_2$PO$_4$, 1.3 MgSO$_4$, 26 NaHCO$_3$, 2 CaCl$_2$ and 20 D-glucose) at 2 ml/min in room temperature. Whole-cell recordings were performed on SLD neurons with borosilicate glass pipettes (3–5 MΩ) filled with an internal solution (composition in mM: 130 K-methylsulfate, 5 KCl, 2 MgCl$_2$, 10 HEPES, 0.1 EGTA, 2 Na$_2$-ATP, 0.2 Na$_2$-GTP, adjusted to pH 7.25 with 1 M KOH). Several previous reports[61–63] and our immunostaing observations (Supplementary Fig. 2b, c) found that this SLD region mainly contained glutamatergic neurons and low density of small (5–15 μm) GABAergic neurons. In order to investigate the effects of orexin on SLD glutamatergic neurons, SLD neurons had a soma diameter larger than 15 μm (membrane capacitance > 80 pF) were recorded. We also used biocytin (0.5%) to label a group of recorded neurons. Post hoc immunostaining revealed that all pre-tested neurons were co-labeled by biocytin and VGLUT1 in this condition (Supplementary Fig. 5a). Recordings were performed in current/voltage clamp mode by Clampex 10.3 (Molecular Devices, USA) using a Multiclamp-700B amplifier (Molecular Devices, USA). Data were analyzed by Clampfit 10.3 (Molecular Devices, USA). In all experiments, neurons were excluded from the study if the series resistance exceeded 20 MΩ or changed by 20%.

Double patch-clamp recordings were applied in pairs of electrically connected SLD neurons. Recorded neurons were first adjusted to around −60 mV by injecting constant current in current-clamp. During orexin application, the orexin-elicited depolarization in membrane potential at the steady state was adjusted to the baseline by constant current injection. Coupling coefficient and input resistance of both cells were determined by applying a series of hyperpolarizing current steps in either cell before and during the steady state of the orexin's effects. Voltage changes

from each step were averaged from at least three sweeps. Input resistance was calculated from the slope of the relationship between injected current and the corresponding voltage changes. Coupling coefficient was calculated from the slope of the relationship between $\Delta V_{Coupled\ cell}/\Delta V_{Injected\ cell}$. All linear fits has a $R$-square value higher than 0.97. To examine whether orexin directly modulates GJs, we calculated the GJ conductance by the following equation[64]:

$$R_{c,cell1} = R_{in,cell1} \times cc_{12} / \left[ \left( R_{in,cell1} \times R_{in,cell2} \right) - \left( R_{in,cell1} \times cc_{12} \right)^2 \right]$$

where $R_{c,cell1}$ represents the GJ resistance from cell1 to cell2, $R_{in,cell1}$ represents the input resistance of cell1, $R_{in,cell2}$ represents the input resistance of cell2, $cc_{12}$ represents the coupling coefficient from cell1 to cell2. GJ conductance from cell1 to cell2 is the inverse of $R_{c,cell1}$.

**Multiple channel recording**. Rats were anesthetized with urethane (1.5 g/kg, i.p.), and supplemental urethane doses of 0.3 g/kg were administered as needed. The skull surface was then exposed. The LFP was obtained from the dorsal hippocampal CA1, using a steel teflon-coated electrode (76.2 μm diameter, 777000, A-M System, USA). Multiple channel recordings were performed by a single-shank silicon probe with eight channels (site diameter: 20 μm, 300–500 kΩ, site interval: 200 μm; Plexon, USA) in the SLD (AP: −9.50 mm, ML: −1.30 mm, DV: −7.60 to −8.60 mm). The probes were pre-covered by DiI for histological analysis, and data were excluded if the recording channels exceeded the SLD. Besides, a micro-syringe (filled with drugs) connecting to a micro-drive (KD310, KD Scientific, USA) was implanted above the SLD (AP = −9.50 mm, ML = −1.50 mm, DV = −7.60 mm) for drug injection. A small screw was fixed above reference electrode. Electrodes were connected to a head-stage with a preamplifier and an omniplex neural data acquisition system (PlexControl 1.10, Plexon, USA) for data collection. Offline Sorter 3.3 (Plexon, USA), Neuroexplorer 4.1 (Nex Technologies, USA), and MATLAB 2014a (MathWorks, USA) were used for data analysis. Wide-band and field potential signals were digitally sampled at 40 and 1 kHz, respectively. After stable recordings in the hippocampus and SLD for at least 15 min, drugs were delivered at a speed of 0.06 μl/min. The baseline of the SLD neuronal activities and hippocampal LFP was calculated from a 600-s epoch before drug applications. The drug responses were evaluated from an equal-epoch when maximal effects were observed. After experiments, the brains were extracted for histological analysis.

**Spike sorting**. To exact spikes in the SLD, the recorded wide-band signal from each channel within the SLD was high-pass filtered at 250 Hz. A threshold that was −4 times the standard deviation of the channel noise was set to detect spikes. A 1 ms refractory period was used to avoid detection of a subsequent spike during this window. Detected spike waveforms were then stored from −0.4 to 1.0 ms around the threshold crossing. Through using off-line sorter 3.3 (Plexon, USA), principal component (PC) analysis of these detected waveforms was performed and the first three linearly uncorrelated PCs were extracted. Then a clustering algorithm with standard expectation-maximation measures operating on the first three PCs was used to distinguish different single units. After the automated sorting, we manually checked the clustering through the three-dimensional plot of PCs. We verified spike times with auto-correlograms to assess the refractory period violations and used cross-correlograms to eliminate duplicates. To assess the response of single units to drugs, the histograms of firing frequency before and after drug application were generated in 1-s bins. A drug response was considered to be present when the change in the firing frequency was larger than twice the SD of the baseline[65].

**Coincidental spiking analysis**. To assess the involvement of GJ activities, we examined coincident spiking within the time scale of ±1 ms by cross-correlogram analysis between spike times of single units (bin size: 1 ms). The mean of the chance coincident was computed from 30 shuffled cross-correlograms. The probability of coincidental spiking was calculated by subtracting the probability of the mean chance from that of the raw cross-correlogram. This method controlled for chance effects in firing rate between pairs[32]. An interaction was then counted if a pairwise 1% threshold was crossed after the subtraction. Note that only pairs of single units between the adjacent recording sites were analyzed, as GJ-mediated interactions occurred between spatially confined recordings[66]. And because prevented by the 1 ms refractory period in spike-detecting process, pairs within the same recording sites were not included. The changes in the probability of coincident spiking within ±1 ms from these data were still sufficient to reflect the drug effects on GJs.

**LFP and phase-locking analysis**. To extract hippocampal LFP, the field potential signals were filtered between 0.3 and 30 Hz. To determine the changes of LFP activities before and after drug applications, the power spectrogram (10 s sliding windows, sequentially shifted by 5 s increments) of the hippocampal LFP were generated using multi-taper methods (five Slepian taper functions, time bandwidth product of 3) with the Chronux data analysis toolbox for MATLAB (http://chronux.org/)[67]. The phase-locking analyses were next performed between SLD single-unit spikes and hippocampal oscillations (0.3–2.5 Hz). Briefly, the hippocampal LFP was band-pass filtered (0.3–2.5 Hz) with a zero-phase filter, and then the instantaneous 0.3–2.5 Hz phase was extracted with a Hilbert transform. Every

spike was assigned to its corresponding phase. Rayleigh's test for circular uniformity was applied to test the significance of phase-locking ($P < 0.01$). The locking strength was defined as the modulus of the average vector of all spike events corresponding to the 0.3–2.5 Hz phase.

**Optogenetics and chemogenetics.** Orexin-Cre mice (male, 8–12 weeks old) were anesthetized with sodium pentobarbital (50 mg/kg, i.p.) and then fixed on a stereotaxic apparatus (RWD Life Science, China). For optogenetics, AAV-Ef1α-DIO-ChR2(H134R)-mCherry, AAV-CAG-FLEX-ArchT-GFP, and AAV-Ef1α-DIO-mCherry (150 nl, OBiO, China) were bilaterally injected into the LH (bregma, AP: −1.55 mm, ML: ±1.05 mm, DV: −5.25 mm) in different groups of orexin-Cre mice. Optical fibers (200 μm in diameter, NA of 0.37, Newdoon, China) were bilaterally implanted above the mice SLD (AP: −5.10 mm, ML: ±1.20 mm, DV: −3.75 mm) for light delivery[68]. For chemogenetics, AAV-retro-hSyn-DIO-hM4D (Gi)-mCherry or AAV-retro-hSyn-DIO-mCherry mixed with AAV-hSyn-EGFP-WPRE-pA (1:1, 23 nl, BrainVTA, China) were bilaterally injected into the SLD (bregma, AP: −5.10 mm, ML: ±0.95 mm, DV: −4.25 mm). The infected region by AAV-hSyn-EGFP-WPRE-pA was used to evaluate the injection site. The viruses were stereotaxically injected using Nanoject II (Drummond Scientific, USA) via a silicate-glass micro-pipette (tip diameter: ~20 μm). Multiple 23 nl (13 nl/s) injections were made at 30-s intervals. After injections, the pipette was left in place for additional 10 min and then slowly retracted, to avoid potential damage to brain tissue.

We crossed orexin-Cre and Ai27D mice to generate the Orexin-Cre;Ai27D offspring (10–14 days) for the validation of activating orexin signaling in SLD by light stimulation in slice physiology. Ai27D mice express the ChR2(H134R)/tdTomato fusion protein in a Cre-dependent manner[69]. The myelination in SLD prevented the identification of SLD neurons in adult mice in the brain slice[17]. But at this age (10–14 days), we found that ChR2 was already abundantly expressed in orexin neurons (Supplementary Fig. 6a). The patch-clamp recording procedure was similar to that of rats. After stale recordings of the baseline, an optical fiber (diameter: 200 μm) was used for light delivery, with the fiber tip positioned above the brain slices. The laser power was gradually increased to induce maximal responses of recorded neurons, and the final power used is estimated to be 5–20 mW. Light was delivered in pulse trains (473 nm, 5 ms, 20-Hz) for 1 s.

After at least 3 weeks of virus injections, mice were implanted with EEG/EMG electrodes. The surgery, recording, and analysis procedures were similar to that of rats. To generate the power spectrum for the EEG signals (5 s sliding windows, sequentially shifted by 0.5 s increments), the multi-taper methods (five Slepian taper functions, time bandwidth product of 3) were used. Hypnograms were built in chemogenetics to calculate the total amount, episode duration, and number of each brain state.

Light lasers (473 or 589 nm, Viashow, China) were controlled by a waveform generator (Master-8, AMPI, Israel). The laser intensity was calibrated to 10–15 mW at the tip by an optical power meter (PM100D, Thorlabs). After the EEG/EMG signals indicating NREM sleep appeared for 30 s, pulses (5 ms) of 473 nm light at 1, 5, and 20-Hz were delivered in the SLD for 1 in every 3 s, and the changes of EMG/EEG signals were recorded in orexin[ChR2] mice. At least 10 trials were conducted for each mice. The inter-trial interval was randomly chosen from a uniform distribution from 15 to 25 min. Optical activation during NREM sleep induced fast decrease in EEG power and a late-onset decrease in EMG amplitude, which can be disrupted by behavioral bouts of wakefulness before the light-off. We thus averaged EEG/EMG data from the trials till the light-off or disrupted by wakefulness. Control experiments were conducted in orexin-Cre mice that received AAV-Ef1α-DIO-mCherry injection in the LH.

To test the role of SLD orexin signaling in REM sleep, we applied a closed-loop stimulation protocol[70] in orexin[ArchT] mice. The EEG signals were visually inspected on-line by an experienced experimenter. After the EEG/EMG signals indicating REM sleep appeared, the laser (473 nm: 20-Hz, 20 pulses every 3 s; 589 nm: a single pulse) was turned on with 50% probability and turned off only when the REM episode ended. An ~10 s delay existed for manual detection to ensure that the mice were in REM sleep. This allowed comparisons of the REM episode durations with and without laser stimulation within the same recording session. The experiments were conducted for 20 trials in each mice. Data were excluded if the duration of the optical stimulation was <10 s due to transitions out of REM sleep. The same criterion was also applied to the control group without laser stimulation. All optogenetics were conducted between ZT 2 and ZT 10.

After recovery from surgery, orexin[SLD-hM4D] mice were allowed to habituate to the recording environment for at least 5 days. On the test day, the EEG/EMG recordings started at ZT 3 (11:00 a.m.). After 1-h stable recordings, CNO (3 mg/kg) or vehicle (0.9% NaCl) were injected intraperitoneally (i.p.) at ZT 4 (12:00 a.m.). CNO and vehicle injections were performed on 2 consecutive days in a random arrangement. After drug injections, brain states were monitored for the next 4 h through EEG/EMG recordings.

To objectively evaluate effects on EEG/EMG performances during REM sleep, EEG/EMG signals during REM sleep (>20 s) and in 10-s epoch of the preceding NREM sleep were computed to extract EMG/EEG values for each REM sleep episodes. Note that the first 10 s of REM sleep was not included to make sure that signals from REM sleep were sampled from a stable state. The EEG theta power during REM sleep was reported to reflect the quality of EEG

theta oscillation. Besides, the EMG$_{REM/NREM}$ ratio was reported to reflect the changes in EMG tone. This may eliminate the posture changes of the animal between NREM sleep and REM sleep that can induce biased EMG modifications[35]. After analyzing all epodes, the hot-map of EMG activities in each mouse were constructed.

**Fiber photometry.** We injected AAV-CAG-FLEX-jGCaMP7b (150 nl, BrainVTA, China) bilaterally into the LH (bregma, AP: −1.55 mm, ML: ±1.05 mm, DV: −5.25 mm) of orexin-Cre mice, to selectively express jGCaMP7b in the orexin neurons. Six weeks after virus injections, mice were implanted with EEG–EMG electrodes and ceramic ferrules above the SLD (AP: −5.10 mm, ML: ±1.20 mm, DV: −3.80 mm). The surgery procedure was similar as above. To observe the REM sleep-related activity of the SLD orexin terminals, a fiber photometry system (Inperstudio Alpha 8.2, Inper, China) was used for recording jGCaMP signals. Data were analyzed by MATLAB 2014a (MathWorks, USA). During recording, an optical fiber (200 μm in diameter, NA of 0.37, Newdoon, China) was inserted into the ferrule. A 488 nm and a 405 nm laser beam were used for jGCaMP7b excitation and isosbestic wavelength, respectively. The power of 488-nm imaging light was set at 30-40 μW, and the 405-nm light power was adjusted to approximately match the jGCaMP fluorescence signals. The emitted signals were captured at 30 Hz with alternating pulses of 488 and 405-nm light, resulting in frame rates of 15 Hz for jGCaMP and the control signals. To synchronize fiber photometry and EEG/EMG recordings, a BNC cable carrying TTL pulses from the Inper system was connected to a digital input channel of the EEG/EMG recording system (Plexon, USA). The sampled signals were low-pass filtered at 2 Hz with a zero-phase filter from the two excitation wavelengths, 488 and 405 nm. The filtered 405 nm signal was aligned to the 488 nm signal through using a least-squares linear fit. $\Delta F/F$ was then calculated according to: (488 nm signal−fitted 405 nm signal)/(fitted 405 nm signal). We recorded signals from 10 REM sleep episodes (>20 s) from each tested animal. The preceding NREM sleep and the following wakefulness of the 10 REM sleep episodes were analyzed for comparison. The $\Delta F/F$ of 10 s from the steady state of each brain state were then averaged.

**Statistics and reproducibility.** All data were plotted and reported as mean ± SEM. Statistical analyses were performed in SPSS Statistics 22.0 (IBM, USA). Shapiro–Wilk test was first used to test the normality on each dataset. If the dataset passed the normality test, parametric tests (two-sided paired or unpaired $t$-test) were used. Otherwise, non-parametric tests (Mann–Whitney rank-sum test or Wilcoxon signed-rank test) were used. One-way repeated-measures ANOVA followed by post hoc LSD comparison tests was used for tests in three or more groups. A threshold of $P < 0.05$ was accepted as statistically different. Significance levels of data are denoted as $*P < 0.05$ and $**P < 0.01$. $P > 0.05$ was considered non-significant and was denoted as n.s. Statistical methods used were all reported in the figure legends.

Experiments were repeated independently in 6 rats for Fig. 1b, c, three times from 3 rats for Fig. 1e–g, in 7 rats for Fig. 1j, in 14 rats for Fig. 5b, in 9 orexin[ChR2] mice for Fig. 6b (left panel), in 27 orexin-cre mice for Fig. 6b (right panel), in 7 orexin[SLD-jGCaMP7b] mice for Figs. 7b, 8, and in orexin[SLD-hM4D] mice for Fig. 8b, c. In the Supplementary figures, experiments were repeated independently in 14 rats for Fig. 2a, three times from 3 rats for Fig. 2b, c, in 3 rats for Fig. 3b, in 16 SLD neurons for Fig. 5a, five times from 5 rats for Fig. 6a, three times from 3 orexin-Cre; Ai27D offsprings for Fig. 10b, in 9 orexin[ArchT-GFP] mice for Fig. 11b, in 8 orexin[SLD-hM4D] mice for Fig 17b, and in 6 rats for Fig. 18a.

**Reporting summary.** Further information on research design is available in the Nature Research Reporting Summary linked to this article.

## Data availability
Source data are provided with this paper. The source data underlying Figs. 1d, i, k, 2c, f, g, i, j, l, m, 3c, e, g, h, 4b, d, 5d, f, g, i, 6e, f, g, i, j, 7d, f, g, 8e, f, i, j and Supplementary Figs. 1b–d, 3c, d, 7c, d, f, g, 8b, c, 9b, d, f, h, j, l, 10f, 11c, 13a–d, 14a–d, 15a–c, 17c, 18b–g, 20a–c are provided as a Source Data file. Source data are provided with this paper.

## Code availability
All the codes used in this study are available upon reasonable request. Source data are provided with this paper.

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

## Acknowledgements

We thank Luis de Lecea (Stanford University) for providing the orexin-Cre mice. This work was supported by grants from the National Natural Science Foundation of China (Nos. 81471346 and 31400956), the National Basic Research Program of China ("973 Program": 2015CB759500), the Advanced Project of College of Basic Medical Sciences, Third Military Medical University (2018JCQY05), and the Foundation For Innovative Research Groups of the Natural Science Foundation of China (31921003).

## Author contributions

N.Y. and J.Z. designed the experiments. S.-Y. Wen, Q.-C.Q., H.F., and N.Y. conducted tracing experiments, S.-Y. Wen., Y.-J.P., and N.Y. conducted in vitro electrophysiology. H.F. and J.Z. conducted the in vivo electrophysiology. H.F., Q.-C.Q., and J.Z. conducted optogenetics and fiber photometry. S.-Y. Wang, H.-Y.L., J.C., and J.Z. conducted pharmacological tests. N.Y. and J.Z. analyzed the data. K.-X.Z., J.C., Z.-A.H., F.-L.L., and G.-Z. W. contributed to discussion of the study. N.Y. and J.Z. wrote the paper with the help of all authors. J.Z. supervised all aspects of the project.

## Competing interests

The authors declare no competing interests.

## Additional information

**Peer review information** *Nature Communications* thanks Antoine Adamantidis and other, anonymous, reviewers for their contributions. Peer review reports are available.

