## [Peer Review File · Nature Communications]

Reviewers' comments:

Reviewer #1 (Remarks to the Author):

The study by Feng and collaborators investigate the role of a subset of orexin neurons in stabilizing REM sleep in rodents. Although the findings are quite surprising - Orexins and their receptors are classically identified as an arousal/wake-promoting system-, the authors show that orexin activate SLD neurons, a brainstem REM sleep center where they increase the synchronous activity of SLD neurons through gap junctions. They further use optogenetic or DREADD approaches to activate either orexin neurons terminals in SLD, or silence SLD-projecting orexin neurons and claim that it increases, and decreases, REM sleep episode duration, respectively. The authors claim that " These findings reveal a stabilization role of orexin in REM sleep".

This manuscript reports original findings on the cellular mechanism of orexin activation of SLD neurons through gap junction. The behavioral experiments supporting the main claim of the study clearly contrast with previous studies in the field (several of which are missing from the reference list), and the current working model. If true, it would suggest that subpopulation of orexin neurons have opposite action (on REM sleep in particular), which is interesting per se, even if the behavioral changes observed here remains quite small (< 20%).

Some parts of the manuscript needs clarifications and re-organization. The authors used a mix of rats and mice in their experiments which should be clearly indicated throughout the manuscript (main text and figure legends). Some figure also include in vitro and in vivo data, which makes it difficult to understand the experimental findings and often bias the interpretation of the results. Similarly, the results on the SLD-Hippocampus connection is awkward and not supported by proper anatomical data. Importantly, the authors often relies on unusual quantification parameter (theta/delta amplitude ratio instead of theta amplitude alone) or different normalization of EEG amplitude (percentage vs mV2). This must be homogenized throughout the manuscript.

Conceptually, I have two major concerns. First, the SLD encompasses both REM-ON and REM-OFF neurons (see work from Luppi & Saper group). Thus, to support the main authors's claim in this manuscript (Orexin synchronize neuronal activity in SLD to stabilize REM sleep), the authors should provide evidence that orexin increase the activity of REM-ON, rather than REM-OFF neurons. That would provide a specific mechanism for the finding and strengthen the study. Second, orexin neurons release both orexin peptide and glutamate (see work from Burdakov group). Thus, in optogenetic experiments described in this study, it is unclear whether orexin, or glutamate (or both), is responsible for the observed effect on REM sleep. Without clear supportive data, the strong claim from the authors is unsubstantiated and the behavioral effect may result from an indirect synaptic pathways.

These concerns are described further below in the in vitro and in vivo section of the study.

In vitro study

-Fig. 1 reports the anatomical projections of orexin neurons to SLD including both excitatory and inhibitory SLD neurons. Could the authors identify the REM-ON and REM-OFF cells to further support their in vivo findings ? according to the current model, those should be c-fos positive after the REM sleep rebound conducted in Fig 1h.

-Do MCH cells are retrogradely labelled ? a comparison of orexin vs MCH population would be interesting to report, in particular due to their involvement in REM sleep (in particular MCH neurons).

-Why the authors use Metfloquine (Fig 3 d-e) or CBX (Fig 3 a-c) to block pannexin (and therefore gap junction) in similar experiments? Same compound should be used for direct comparisons (Fig 3).

-Legend of fig 3c: it should read :“showing that The orexin-elicited”

In vivo studies

- Figures 5 & 6:

Fig 5d: if I understand well this results, it relates to paired cell recording in vitro. If so, it is confusing to have it side-by-side to in vivo data. These panels (Fig 5d, e) must be moved to the in vitro figure to avoid confusion.

Perhaps they could be merge with Fig 6a which also mix in vitro and in vivo datasets. This is very confusing. Could Figure 6 and figure 5 be consolidated into a single figure ?

Fig 5h: Phase locking analysis between spike recordings in (b) and hippocampal slow oscillations in (f) before and after SLD orexin injection does not reveal any significant locking. There is perhaps an increase of amplitude of urethane-induced low frequency-high amplitude oscillatory activities in the hippocampus but the phase locking of SLD neurons to hippocampal LFP remains minimal. Are those findings relevant for the main message of the paper ? If so, the authors should provide anatomical evidence as a rationale to record LFP/single unit in those two areas.

On the heat map (Fig 5f), there is an interruption of the main oscillatory activity around 1000s (after Orexin-A application) – What does this correspond to ? Could it be that the animal emerges from anesthesia ?

-Panels from Fig 6e, f must be represented on the same figure since they compared similar experimental conditions. For consistency, the authors must represent their data using the same Y-axis units for power/amplitude quantification (either percentage or mV²).

I would encourage the authors to show raw EEG traces (with magnification) in addition to power spectrum heatmaps (Fig. 5 & 6).

- Figure 7 & 8: My interpretation of these results is quite different than the authors's claims. From Fig 7c(d and also g), the optogenetic activation of orexin terminals in SLD region during NREM sleep induced a cortical activation (left panel) that is likely results in some cases in a (short) awakening (as clearly shown by the EMG signals on the left panel). This cortical 're-activation' reflect a decrease of slow wave activities (0-5-4.5 Hz) and a concomitant increase of fast oscillations including theta and gamma that are typical of wake states. Therefore, my interpretation of these data is that the optogenetic stimulation wake up the animal but certainly do not prolong REM sleep. This latency to wake should be quantified and reported.

Importantly, EEG theta/delta ratio is not a sensitive marker of REM sleep and if not considered as such by the community. REM sleep is evidenced by a high amplitude theta rhythm generated in the hippocampus (detected on the cortical EEG as well), a complete muscle atonia, Eye-movement, and fluctuating heartbeat and breathing rate (though the last 3 are difficult to see on classical EMG recordings from neck electrodes). Thus, authors should report theta amplitude using a consistent y-axis unit of EEG power spectrum values.

-Fig 7 h,j: The authors must report raw values of theta amplitude alone instead of theta/delta ratio, or theta percentage (percentage of total power ?), in accordance to the standards in the field.

I am not sure whether EMG (REM?NREM) ratio is useful here.

Finally, I am left wondering why the authors didn't use a similar retrograde targeting strategies as they use in the silencing experiment (Fig 8) in this set of experiment ? overall it makes results difficult to compare.

Figure 8 use GCAMP6 and fiber photometry to show that orexin neurons are silent during NREM sleep, slightly more active during REM sleep and strongly activated when the animal wake up. The higher

GCAMP6 activity during REM sleep is somewhat odd in comparison with previous literature (work from B. Jones group) which showed that orexin neurons increase their activity before a REM-to-wake transition, suggestive of an wake-promoting action (as shown with optogenetics in many publications). This may result from activity linked to muscle twitches. Is that the case? This is difficult to assess from the EMG integral.

How do the authors explain this elevated activity during REM sleep? This finding is very surprising.

In Fig 8e the authors report optogenetic silencing experiment during REM sleep, however, some experimental details are lacking from the manuscript and figure.

-the authors should report details regarding the targeting of silencing opsins to Orexin cells in vivo and evidence for optical silencing of their neuronal activity.

-The authors must report values of theta amplitude alone instead of theta/delta ratio, in accordance to the standards in the field.

As in Fig 7, I am not sure whether EMG (REM/NREM) ratio is useful here.

-Fig 8g-o show that DREADD silencing of orexin neurons projecting to SLD decreases the duration of REM sleep episodes. Fig 8n,o should be replaced as suggested for previous in vivo figure 7.

-Finally, the author must report a clear quantification of targeting methods used in this study (Fig 7, 8).

Reviewer #2 (Remarks to the Author):

This is an interesting paper, presenting important evidence for a causal role of orexin axon activity in SLD in promoting REM sleep.

The most important finding in the paper is that optogenetic silencing of SLD orexin cell axons disrupts REM sleep, while optogenetic activation of these axons promotes REM sleep. Overall I am enthusiastic about this paper. However, there are several important changes that would significantly improve it:

Abstract:

As far as I know, the efficiency of the retrotracer (i.e. what percentage of true connections it labels) is known. Therefore, the precise percent claim is unjustified ("Here, we found that ~7% orexin neurons projected to the sublaterodorsal tegmental nucleus (SLD) and exhibited REM sleep related activation"), since many more orexin neurons may project there but are not detected due to tracer inefficiency.

General language/terminology used:

Overall the paper is elegantly written and a pleasure to read.

However, the authors repeatedly use certain scientific terms without any justification, which makes their claims unsubstantiated and weakens the paper. I suspect this is a linguistic misunderstanding rather than over-interpretation of results, and I think it must be corrected by textual re-wording. Key examples:

- "Efficient" (eg in efficient output, output efficiency). Efficient means achieving maximum output with minimum resource - but the authors never study any resource allocation. So efficient must be removed, simply output would suffice.

- "Information" (eg orexin-elicited information). Information, in science, has a specific mathematical definition and the authors have not studied or defined this. So they should replace information with a word like effect or response.

The language is also often redundant to the point of making some sentences meaningless, the paper should be edited to fix this. For example "Membrane dynamics influences firing activity..besides

membrane potential level, sub threshold fluctuations are involved ..”

These 4 terms (Membrane (potential) dynamics, firing activity, fluctuations, membrane potential level) mean the same thing = membrane potential dynamics, so this is a very circular statement.

Results:

- Please report real neuron counts as well as %
- Gap junction section is not convincing. Are the gap junction pharmacological blockers specific? (Probably not..). Did biocytin spread to neighboring neurons from the patch-clamp neuron, as would be expected from gap junction? If not this argues against gap junctions.

References:

The paper is missing some key references in this field (I.e. link between orexin cell firing and brain electrophysiology). At the very least, references to the following key papers must be cited:

On effects of orexin cell activity on EEG:

Adamantidis et al, Nature 2007 Nov 15;450(7168):420-4

Hara et al, Neuron 2001 May;30(2):345-54.

Chemelli et al, Cell 1999 Aug 20;98(4):437-51

Tabuchi et al, J Neurosci 2014 May 7;34(19):6495-509

On effects of orexin cell activity on downstream circuits, including effects of other transmitters and inhibitory orexin signalling in sleep-wake relevant structures:

Kosse et al, PNAS 2017 Apr 25;114(17):4525-4530

Schone et al, Cell Reports 2014 May 8;7(3):697-70

Belle et al, J Neurosci. 2014 Mar 5;34(10):3607-21

Blomeley et al, Nat Neurosci. 2018 Jan;21(1):29-32

Reviewer #3 (Remarks to the Author):

In this exciting study Feng et al investigated the functions of a subpopulation of orexin neurons which project to the sublateralodorsal tegmental nucleus (SLD). Orexins directly excited a large proportion of SLD cells and increased probability of synchronized firing in the SLD. Fiber photometry experiments showed activation of Orx-SLD pathway during REM sleep and wakefulness, whereas opto- and chemogenetic manipulations of this pathway demonstrated its role in REM sleep episodes duration and REM sleep-associated atonia. These novel and important findings show that orexin-SLD pathway stabilizes REM sleep. The study expands our knowledge of sleep-regulating neural circuits. The authors employed an impressive combination of techniques. However, a number of mice is low in several groups, and some controls are not yet provided.

Specific concerns:

1. Number of animals in some groups is lower as usually used. In particular, for c-fos experiments (Fig.1) and for fiber photometry experiments (Fig.8d) 4 animals/group were used.

2. Fig 5 e- normalized values are shown on the figure. It would be useful to know how often does the coincident spiking typically occur. How high was the average coincident spiking probability?

3. For experiments shown on Fig.6 CBX should be used as a control instead of "Baseline", especially since the authors showed that CBX alone reduced coincidental spiking almost two-fold (Suppl.fig.5b).

4. Fig.7f - are SEM or SD shown on bar plots? If indeed SEM are shown, the variability looks a bit too high for a reported significance level ($p < 0.01^{**}$ for 20 Hz).

5. The authors showed that optogenetic activation of orexin-SLD projections prolonged the duration of

REM sleep episodes - did it also promote transitions from nonREM to REM if optogenetic stimulation was applied during non-REM sleep?

6. Did the proportion of sleep epochs resulted in waking change upon the optogenetic excitation or inhibition of Orx-SLD projections?

7. Was an automated sleep phase detection algorithm or visual inspection by an experimenter applied in the closed-loop optogenetic experiments? What was defined as e.g. "stable NREM sleep" - a continuous NREM sleep epoch without wakefulness or REM episodes?

Minor comments:

The paper would benefit from further proofreading. There are some typos, e.g. "Orxin" in a subtitle on page 25 , "optogentics" on page 50 etc.

I would add "lateral" to the title of Suppl. Fig. 1: SLD-projecting orexin neurons are sporadically distributed in the LATERAL hypothalamus

Response to Reviewers:

Reviewer #1 (Remarks to the Author):

The study by Feng and collaborators investigate the role of a subset of orexin neurons in stabilizing REM sleep in rodents. Although the findings are quite surprising - Orexins and their receptors are classically identified as an arousal/wake-promoting system-, the authors show that orexin activate SLD neurons, a brainstem REM sleep center where they increase the synchronous activity of SLD neurons through gap junctions. They further use optogenetic or DREADD approaches to activate either orexin neurons terminals in SLD, or silence SLD-projecting orexin neurons and claim that it increases, and decreases, REM sleep episode duration, respectively. The authors claim that "These findings reveal a stabilization role of orexin in REM sleep".

Issue 1: This manuscript reports original findings on the cellular mechanism of orexin activation of SLD neurons through gap junction. The behavioral experiments supporting the main claim of the study clearly contrast with previous studies in the field (several of which are missing from the reference list), and the current working model. If true, it would suggest that subpopulation of orexin neurons have opposite action (on REM sleep in particular), which is interesting per se, even if the behavioral changes observed here remains quite small (< 20%).

Response:

We thank for the comment and following suggestions to our work. As you mentioned, orexin neurons are found to promote wakefulness through exciting several downstream wake-promoting nuclei (Adamantidis et al., 2007; Sasaki et al., 2011; Carter et al., 2012; Schöne et al., 2012; Tsunematsu et al., 2013; Hasegawa et al., 2014; Ren et al., 2018). It is also interesting to note that, in addition to wakefulness maintaining deficits, REM sleep deficits are frequently reported in narcoleptic patients (with up to 60% of narcoleptic patients suffering from failure of muscle

tonia) (Knudsen et al., 2010; Dauvilliers et al., 2013). This strange phenomenon of disruptions in both states has aroused interests (Arrigoni et al., 2016; Mahoney et al., 2019). Intriguingly, although the inhibition effect of orexin on REM sleep through exciting Wake-on/REM-off regions are frequently proposed (Kaur et al., 2009; Bourgin et al., 2000; Arrigoni et al., 2016), excitatory orexinergic projections have also been found in REM sleep promoting brain regions, including the SLD (Peyron et al., 1998; Torterolo and Chase, 2013). Therefore, investigating the potential opposite actions of orexin on REM sleep may help provide more insights into deficits in both states after loss of orexin signaling.

Previously, an again elevated activity level of the orexinergic system during REM sleep has been reported both from the neuronal firing (Lee et al., 2005; Mileykovskiy et al., 2005; Takahashi et al., 2008) and orexin release (Kiyashchenko et al., 2002) aspects. These studies may provide the preliminary evidence for the REM sleep relevance of orexin signaling, in addition to its role in wakefulness. Importantly, as you suggested, the existence of a subpopulation of orexin neurons promoting REM sleep is a quite likely explanation for understating how exactly orexin system can orchestrate different brain states. Many studies have focused on distinguishing the functional subdivisions of the orexin neurons from different perspectives. Unfortunately, studies employing single-cell RNA-sequencing technique failed to detect differences in this system at the mRNA level (Mickelsen et al., 2019). Several findings have alternatively suggested another possible idea to distinguish subpopulation of orexin neurons based on their downstream projection patterns (Harris et al., 2005; Iyer et al., 2018). We have previously held the similar idea (Zhang et al., 2011; Hu et al., 2015; Yang et al., 2017). Our current study may also provide some clues for this. As you may see, the orexin neurons projecting to the SLD were largely non-overlapping from those to the locus coeruleus (LC), and a significant activation of the SLD-projecting orexin neurons emerged from the orexin neuronal entirety after the REM sleep rebound. In fact, the existence of projection-site based divisions in hypothalamic neurons seems to be understandable. A previous study on the hypothalamic galanin-expressing neurons has found this kind of divisions, which

serves to better orchestrate different aspects of complex parental behaviors (Kohl et al., 2018). Whether such kind of projection-based division is a common functional circuit architecture in the hypothalamus is still an open issue.

Following these clues, we have systemically investigated the electrophysiological modulations/mechanisms of orexin in the SLD neurons, the activity pattern of the SLD orexin projection, as well as its behavior consequences on REM sleep. All obtained data together have logically suggested a stabilization role of SLD orexin signaling in REM sleep regulation. With your helps, additional evidences have been provided in the Revised Manuscript, especially the target of orexin signaling in the SLD REM-on neurons and the physiological relevance of the elevated SLD orexin signaling in REM sleep. We found that orexin receptors were abundantly expressed on the SLD REM-on neurons (Revised Supplementary Fig. 4) and the GCaMP signals of the SLD orexin terminals elevated throughout the REM sleep episode, including both phasic and tonic REM sleep (Revised Supplementary Fig. 15). We hope that these new findings further support the REM sleep stabilization role of SLD orexin signaling. For details in these and other important issues, please see the following responses. After a careful revision following your instructions, we feel that the manuscript is better presented and the conclusions are still valid.

Issue 2: Some parts of the manuscript needs clarifications and re-organization. The authors used a mix of rats and mice in their experiments which should be clearly indicated throughout the manuscript (main text and figure legends). Some figure also include in vitro and in vivo data, which makes it difficult to understand the experimental findings and often bias the interpretation of the results. Similarly, the results on the SLD-Hippocampus connection is awkward and not supported by proper anatomical data. Importantly, the authors often relies on unusual quantification parameter (theta/delta amplitude ratio instead of theta amplitude alone) or different normalization of EEG amplitude (percentage vs mV²). This must be homogenized throughout the manuscript.

Response:

Thank for these suggestions. We have carefully examined the manuscript according to your suggestions, and revisions were made in the following aspects to improve the manuscript,

1. Necessary captions and descriptions have been added in the revised figures and corresponding texts, respectively, to more clearly clarify the experimental design and the findings (e.g. usage of rats/mice; in vitro/in vivo experiments). Besides, we have also explained the usage of animals in the method section (Page 40, Line 913-918) in the Revised Manuscript.

2. The original Fig. 5 and Fig. 6 related to the in vivo electrophysiological results including functional SLD-Hippocampus connections have been re-organized to better serve the main idea of this paper. Now the Revised Fig.5 and Revised Supplementary Fig. 8 and 9 are related to these results. For details, please see responses to Issue 9-13.

3. We have replaced all EEG theta/delta ratios by raw values of theta oscillation powers, and the usage of these parameters were explained and homogenized in the Revised Manuscript.

For details in these revisions, please see the following responses. We hope you will find that all these revisions made following your suggestions may help further confirm our conclusions that SLD orexin signaling stabilizes REM sleep.

Issue 3: Conceptually, I have two major concerns. First, the SLD encompasses both REM-ON and REM-OFF neurons (see work from Luppi & Saper group). Thus, to support the main authors's claim in this manuscript (Orexin synchronize neuronal activity in SLD to stabilize REM sleep), the authors should provide evidence that orexin increases the activity of REM-ON, rather than REM-OFF neurons. That would provide a specific mechanism for the finding and strengthen the study.

Response:

With great interest following this question you suggested, we have conducted

new experiments to examine the target of SLD orexin signaling among functionally distinguished SLD neurons. We found that a large majority of the SLD REM-on neurons indeed expressed orexin receptors, further providing a functional basis for the observed REM sleep stabilization effect by SLD orexin signaling. Please see the response to Issue 5 and the Revised Supplementary Fig. 4 for detailed experimental results.

The results in Revised Supplementary Fig.4 also showed that some c-Fos negative SLD neurons (after the REM sleep rebound) expressed orexin receptors. Unfortunately, because of the shortages in current models, the potential REM-off neurons could not simply be identified from these c-Fos negative SLD neurons. But according to works from Luppi group, the GABAergic neurons in the SLD may not be REM-on neurons and suspected that they might be REM-off interneurons (Boissard et al., 2003; Sapin et al., 2009). Therefore, considering that GABAergic neurons expressed orexin receptors and orexin signaling utilized volume transmission in the SLD in the present study, it is unlikely that orexin may have cell type selectivity in the SLD, no matter whether the GABAergic neurons are REM-off neurons or not (Lu et al., 2006).

In this condition, the potential excitation of orexin on the SLD REM-off neurons could not be excluded. But importantly, the SLD is considered as REM-promoting at the functional level, as inactivation of SLD using different methods reduced REM sleep amount (Sanford et al., 2005; Lu et al., 2006; Valencia Garcia et al., 2017). Consistently, large number of REM-on glutamatergic neurons have been identified in the SLD (Lu et al., 2006; Clément et al., 2011). Compared to the abundant glutamatergic neurons, previous studies (Clément et al., 2011; Boissard et al., 2003; Krenzer et al., 2011) and our immunostaining results (Supplementary Fig. 2) found that the number of GABAergic neurons in the SLD was relatively small. This is also supported by in situ hybridization (ISH) data from the Allen Brain Atlas (experiment 73818754 and 72081554 for glutamatergic and GABAergic neurons, respectively). Taking all these into account, the direct excitation of orexin on SLD REM-on glutamatergic neurons should at least be a major player that finally participate to

stabilize REM sleep, as revealed in the present study.

In comparisons, previous studies concerning orexin's modulation in REM sleep indeed focused on the Wake-on/REM-off neuron enriched regions, such as the LC and ventrolateral part of the periaqueductal grey matter (vlPAG)/lateral pontine tegmentum (LPT) area (Kaur et al., 2009; Bourgin et al., 2000; Arrigoni et al., 2016). They found an inhibitory role for orexin in REM sleep control. If considering this inhibition alone, it would not suffice to explain all aspects of REM sleep deficits in narcolepsy, which contains poor circadian timing and rapid onsets of REM sleep, as well as disruption of REM sleep physiology (for example, failure of muscle atonia and abnormal EEG theta oscillations) (Bastianini et al., 2012; Dauvilliers et al., 2013). The present study on the direct relationship between orexin and SLD may provide a more comprehensive understanding on the disrupted REM sleep physiology after loss of orexin signaling. All these findings together suggest that REM sleep is orchestrated by the hypothalamic orexinergic system in a complex manner.

Issue 4: Second, orexin neurons release both orexin peptide and glutamate (see work from Burdakov group). Thus, in optogenetic experiments described in this study, it is unclear whether orexin, or glutamate (or both), is responsible for the observed effect on REM sleep. Without clear supportive data, the strong claim from the authors is unsubstantiated and the behavioral effect may result from an indirect synaptic pathways.

Response:

We thank for this important comment. According to your suggestion, the possible involvement of the indirect synaptic pathways should also be examined. An additional group of SLD neurons from orexin-ChR2 mice was added to examine the optical-elicited excitatory responses in the presence of synaptic transmission blockers (DNQX 40 μ M, MK 801 20 μ M, and picrotoxin 100 μ M). In this condition, we found that light-delivery still elicited a depolarization of 2.9 ± 0.4 mV in the SLD neurons (n

= 6 neurons from orexin-ChR2 mice). Moreover, in our original manuscript, we have provided evidence that the excitatory responses of SLD neurons elicited by optical activation of the SLD orexin terminals in orexin-ChR2 mice can be totally blocked by the orexin receptor antagonist TCS 1102. In summary, the optical-elicited depolarization was not significantly different in the DNQX/MK/PTX and normal ACSF condition, and can be totally blocked by TCS 1102 (Please see the Supplementary Fig.10 for details). Together with the results from slice physiology in rats, these data suggest that the effects elicited by optical activation of SLD orexin terminals were mediated by the release of orexin, but not glutamate (Schöne and Burdakov, 2017).

Based on these new data, we have added a new panel in the original Supplementary Fig. 6 (now in the Revised Supplementary Fig. 10). Besides, the related descriptions have also been revised in the Revised Manuscript (Page 21, Line 479-484). The Revised Supplementary Fig. 10 is listed below,

Supplementary Fig. 10 Optogenetic activation of the SLD orexin terminals depolarizes SLD neurons, and this depolarization is mediated by the release of orexin

a Schematic for optogenetic activation of SLD orexin terminals in brain slices from the orexin-Cre;Ai27D offspring.

b The Ai27D mice express a ChR2(H134R)/tdTomato fusion protein in a Cre-dependent manner. ChR2-tdTomato expression in the LH orexin neurons was found at the age (8-14 days) of the used orexin-Cre;Ai27D offspring (scale bar, 20 μ m).

c Optical (473 nm) activation (20 Hz; pulse width: 5ms; duration: 1 s) of the SLD orexin terminals elicited a depolarization in the tested SLD neuron under normal ACSF condition.

d Optical activation of the SLD orexin terminals still elicited a depolarization in the tested SLD neuron after adding DNQX 40 μ M, MK 801 (MK) 20 μ M, and picrotoxin (PTX) 100 μ M into the ACSF.

e In the presence of TCS 1102 (10 μ M), optical activation of the SLD orexin terminals failed to elicit a depolarization in the tested SLD neuron.

f Group data showing the optical-elicited responses in the SLD neurons under normal ACSF, DNQX/MK/PTX, and TCS 1102 conditions, respectively (normal ACSF: 3.2 ± 0.9 mV, DNQX/MK/PTX: 2.9 ± 0.4 mV; TCS 1102: 0.2 ± 0.3 mV; $n = 6$ neurons for each group; one-way ANOVA, $F_{(2, 15)} = 8.216$, $P < 0.01$; post-hoc LSD comparison test; ACSF vs DNQX/MK/PTX, $P = 0.719$; ACSF vs TCS 1102, $P < 0.01$; DNQX/MK/PTX vs TCS 1102, $P < 0.01$).

Data are presented as mean \pm SEM. n.s., no significant difference; ** $P < 0.01$.

These concerns are described further below in the in vitro and in vivo section of the study.

In vitro study

Issue 5: -Fig. 1 reports the anatomical projections of orexin neurons to SLD including both excitatory and inhibitory SLD neurons. Could the authors identify the REM-ON and REM-OFF cells to further support their in vivo findings? According to the current model, those should be c-Fos positive after the REM sleep rebound conducted in Fig 1h.

Response:

We thank for this experimental suggestion that helps further examining the orexin's modulation on functional SLD neurons, and thereby, clarifying its role in REM sleep regulation (see also response to Issue 3). Following the suggestion, we have performed double immunostaining of orexin receptors (OXRs) and c-Fos to

examine the distribution of OXRs in SLD neurons after the REM sleep rebound. As shown in the following figure (Revised Supplementary Fig. 4), the large majority of SLD REM-on ($c\text{-Fos}^+$) neurons also expressed OXRs. These results suggest that a direct and noteworthy influence of orexin on SLD REM-on neurons indeed exist, supporting the behavioral findings.

Besides, some $c\text{-Fos}^-/\text{OXRs}^+$ SLD neurons were also observed. These neurons may include potential REM-off SLD inhibitory neurons. But according to the current model, the $c\text{-Fos}^-$ neurons can not be simply identified as REM-off neurons and may also include a certain amount of unlabeled REM-on neurons (Dragunow and Faull, 1989; Renouard, 2015). Considering these results and that orexin utilizes volume transmission in the SLD, cell type selectivity of orexin's effects in the SLD may be indeed lacking. The final behavioral consequences may thus account from the effects of orexin on the entire SLD neuronal network. The abundant existence of $\text{OXRs}^+/\text{c-Fos}^+$ SLD neurons further suggests that the orexin's modulations on SLD REM-on neurons are a major effect for the final behavioral consequences.

Based on these new data, we have constructed a new figure in the Revised Supplementary Information (Revised Supplementary Fig. 4). Besides, we have added related descriptions in the Revised Manuscript (Page 9, Line 191-194). The Revised Supplementary Fig. 4 is listed below,

Supplementary Fig. 4 Orexin receptors are expressed in SLD REM-on neurons after the rebound period from 72-h REM sleep deprivation.

a Double immunostaining of OXRs and c-Fos were performed in the SLD region of rats after 2.5-h rebound period from 72-h REM sleep deprivation (RSD). EEG/EMG recordings showed that the amount of REM sleep was enriched to $33.6 \pm 3.2\%$, during the 2.5-h rebound period after the RSD ($n = 3$ rats). The immunostaining results showed that a large majority of REM-on (c-Fos⁺) SLD neurons also expressed OXRs. Besides, OXRs⁺ SLD neurons without c-Fos expression were also found in this condition.

Issue 6: -Do MCH cells are retrogradely labelled? A comparison of orexin vs MCH population would be interesting to report, in particular due to their involvement in REM sleep (in particular MCH neurons).

Response:

To examine this interesting issue, the SLD-projecting MCH (MCH^{SLD}) neurons were retrogradely labeled by CTB/MCH-immunostaining, and the activities of them were examined by c-Fos after the REM sleep rebound in rats. We noticed that although MCH and orexin neurons are located in similar regions of the hypothalamus, the distribution range of MCH neurons is broader than orexin neurons. The MCH^{SLD} account for $8.5 \pm 0.7\%$ of the MCH neuron entirety (MCH^{entirety}), and they were also sporadically distributed in the hypothalamus. In the MCH^{entirety}, $71.4 \pm 1.8\%$ of them expressed c-Fos after the REM sleep rebound. As for the MCH^{SLD} neurons, $70.8 \pm 1.6\%$ of them was found to express c-Fos. The c-Fos expression level exhibited no significant difference between the MCH^{entirety} and the MCH^{SLD} neurons.

The existence of a subset of REM-on MCH neurons projecting to this REM-on (SLD) region is interesting, as the MCH neurons have been classically identified to facilitate REM sleep through exerting inhibitory modulations on REM-off regions (Varin et al., 2018; Luppi et al., 2013). Actually, a previous study has reported that microinjection of MCH into the SLD inhibited REM sleep in rats (Monti et al., 2016). These clues in MCH system may also be compatible with the idea that sub-divisions

exist in the hypothalamic neurons to orchestrate a behavior. Intriguingly, there is also a tempting to study the sub-divisions of MCH system. For example, Izawa et al (2019) have reported that both REM sleep-active and wake-active MCH neurons existed in the hypothalamus, and they were distinct populations.

We have constructed a new figure (Supplementary Fig. 3) to provide these new results for potential interests and added related descriptions on Page 9, Line 188 - 191 in the Revised Manuscript.

Revised Supplementary Figure 3

Supplementary Fig. 3 Identification of the SLD-projecting melanin-concentrating hormone (MCH) neurons and their REM sleep related activities.

a CTB-488 was injected in the unilateral SLD of rats. After the rebound from 72-hour REM sleep deprivation, MCH and c-Fos immunostaining were performed to identify the SLD-projecting MCH (MCH^{SLD}) neurons and compare c-Fos expression between the MCH^{SLD} neurons and the total MCH population ($MCH^{entirety}$). EEG/EMG recordings showed that the amount of REM sleep was enriched to $33.6 \pm 3.2\%$, during the 2.5-h rebound period after the RSD ($n = 3$ rats).

b Two coronal images showing the distribution of $MCH^{entirety}$ and MCH^{SLD} neurons, and their c-Fos expression after the REM sleep rebound. C-Fos expression were observed in both the $MCH^{entirety}$ and MCH^{SLD} (arrows indicated) neurons. Scale bars, 50 μm .

c Group data showing that the MCH^{SLD} neurons ($CTB-488^+MCH^+/MCH^+$) accounted for $8.5 \pm 0.7\%$ of the total MCH population.

d Group data showing that c-Fos expression was not significantly different between the $MCH^{entirety}$ and MCH^{SLD} neurons after the REM sleep rebound ($MCH^{entirety}$: $71.4 \pm 1.8\%$, MCH^{SLD} : $70.8 \pm 1.6\%$; $n = 3$ rats, paired t test; $t_2 = 0.167$, $P = 0.883$; analysis of 5607 MCH neurons and 468 MCH^{SLD} neurons from 3 rat brains).

Data are presented as mean \pm SEM. n.s., no significant difference.

Issue 7: -Why the authors use Mefloquine (Fig 3 d-e) or CBX (Fig 3 a-c) to block pannexin (and therefore gap junction) in similar experiments? Same compound should be used for direct comparisons (Fig 3).

Response:

We are sorry for not clearly describing these results. We used CBX (Fig 3 a-c), and then again mefloquine (Fig 3 d-e), aiming at confirming the involvement of gap junction (GJ) in the orexin-elicited broad excitation in SLD neurons. The use of different GJ blockers in the same experimental design to confirm the GJ-mediated effects has been suggested (Cruikshank et al., 2004). The effects of individual drugs (CBX and mefloquine, respectively) on the orexin's effects were indeed directly compared before and after application. After finding the consistent blockage effect of both drugs on the orexin-elicited excitatory responses, we noticed that CBX and mefloquine totally blocked the responses in 26.7% and 24.0% tested SLD neurons, respectively. Therefore, these data may further suggest that SLD orexin signaling directly targets on ~3/4 of SLD neurons. We have re-written the related descriptions to avoid misunderstanding to read "After the blockage of gap junction activities, around ~1/4 SLD neurons were found to be orexin-A non-responsive, suggesting that the SLD orexin signaling directly targets on ~3/4 of SLD neurons." (Page 13, Line 286-288).

Issue 8: -Legend of fig 3c: it should read :”showing that The orexin-elicited”

Response:

We thank for the help in language issue. This sentence has been re-written to read: "Group data showing that the orexin-elicited inward current was reduced in the presence of CBX." (Pages 14, lines 309-310). Similar descriptions were carefully checked in the Revised Manuscript.

In vivo studies

- Figures 5 & 6:

Issue 9: Fig 5d: if I understand well this results, it relates to paired cell recording in vitro. If so, it is confusing to have it side-by-side to in vivo data. These panels (Fig 5d, e) must be moved to the in vitro figure to avoid confusion.

Perhaps they could be merge with Fig 6a which also mix in vitro and in vivo datasets. This is very confusing. Could Figure 6 and figure 5 be consolidated into a single figure ?

Response:

Sorry for not describing these in vivo data clearly. According to your suggestions in this issue and Issues 10, we noticed that these results should be more clearly presented to better serve the main idea of this paper. Therefore, we have re-organized the main results from the original Fig. 5 and 6 into a new figure (Revised Fig.5). In this revised figure, only the in vivo modulations of orexin in SLD were retained, and the changes of hippocampal activities after SLD drug injections were all moved to the Revised Supplementary Fig. 9. Besides, necessary captions were added in these new figures to avoid confusions.

Data in the original Fig 5d,e and Fig 6a,b are now presented in the Revised Fig. 5e,f and 5h,i, respectively. These results are coincidental spiking analyses from the pairs of SLD units in multiple channel recordings. Through these analyses, gap junction activities can be identified in vivo when significant coincidental spiking activities occurred between the recorded pairs of units in SLD within ± 1 ms (Totah et al., 2018). In this condition, the modulation of orexin on the gap junction activities can be examined. As a result, all these data from the Revised Fig. 5 described that microinjection of orexin-A into the SLD increased the firing frequency and gap junction mediated coincident spiking activities of the SLD neuronal network, indicating that orexin elicited synchronized excitation in the SLD neuronal network in

vivo.

The corresponding descriptions in the main text and figure legends have also been re-written (Please see Page 17-19, Line 382-467). The Revised Figure 5 is listed below,

Revised Figure 5

Fig. 5 Microinjection of orexin-A into the SLD increases the frequency and synchrony level of spiking activities in the SLD neuronal network in vivo.

a Schematic of SLD multiple channel recordings and drug injections in urethane-anesthetized rats in vivo.

b The locations of electrode arrays (pre-covered by DiI, red) and drug canulas (red rectangle) from a tested rat. Scale bar: 500 μ m.

c Microinjection of orexin-A (30 μ M, 0.3 μ l) increased the firing frequency of a recorded SLD unit in vivo. Overlaid spike waveforms and raster plot of spikes in a 10-second epoch before and after orexin-A application were expanded at the top.

d Group data showing that orexin-A significantly increased the firing frequency of SLD neurons (n = 67/97 units, Wilcoxon signed rank test; z = 7.115, P < 0.01).

e Two example cross-correlograms with significant coincidental spiking on the timescale of \pm 1 ms before and after orexin-A microinjections from the same tested

rat in (c). Note that gap junction mediated interactions can occur either within the sharp time window of ± 1 ms (pair 1) or a larger time window of ± 5 ms (pair 2). We focused on the coincident activities within ± 1 ms. A significant interaction was counted if a pairwise 1% threshold (dotted gray lines) was crossed. The mean of the chance coincident (solid blue line) was computed from 30 shuffled cross-correlograms.

f Group data showing that orexin-A significantly increased the coincident spiking probability of SLD unit pairs with significant interactions in both baseline and orexin-A conditions ($n = 18/215$ pairs, Wilcoxon signed rank test; $z = 3.881$, $P < 0.01$). Note that only pairs of single units between the adjacent recording sites were analyzed.

g Group data showing that microinjection of orexin-A ($30 \mu\text{M}$, $0.3 \mu\text{l}$) into the SLD still increased the firing frequency of SLD neurons after the pre-injection of CBX (100 mM) ($n = 39/72$ units, Wilcoxon signed rank test; $z = 5.442$, $P < 0.01$).

h Two example cross-correlograms with significant coincidental spiking on the timescale of ± 1 ms before and after orexin-A microinjection in the presence of CBX.

i Group data showing that orexin-A failed to increase the coincident spiking probability of SLD unit pairs with significant interactions in both baseline and orexin-A conditions in the presence of CBX ($n = 13/182$ pairs, Wilcoxon signed rank test; $z = 0.622$, $P = 0.542$).

Data are presented as mean \pm SEM. n.s., no significant difference; ** $P < 0.01$.

Issue 10: Fig 5h: Phase locking analysis between spike recordings in (b) and hippocampal slow oscillations in (f) before and after SLD orexin injection does not reveal any significant locking. There is perhaps an increase of amplitude of urethane-induced low frequency-high amplitude oscillatory activities in the hippocampus but the phase locking of SLD neurons to hippocampal LFP remains minimal. Are those findings relevant for the main message of the paper? If so, the authors should provide anatomical evidence as a rationale to record LFP/single unit in those two areas.

Response:

According to your suggestions, the results about the hippocampal activity changes were all removed from figures in the main text, to help better deliver the main message of this paper. A Revised Supplementary Fig. 9 was used to include these experiments for potential interests. Through these experiments, we intended to

evaluate whether the orexin-enhanced SLD output may finally contribute to REM sleep via affecting activities of the downstream REM sleep circuitries. It has been reported that the SLD output is important for physiological state changes during REM sleep, including the hippocampal EEG activation (Lu et al., 2006). Therefore, in urethane-anesthetized rats, we recorded the changes of hippocampal slow oscillation as an index to reflect the orexin-elicited SLD output changes, and found that SLD orexin microinjections increased its amplitude.

These oscillation changes were also used as a quantification parameter for the correlation analysis with the drug modulated SLD neuronal activities, to probe the involved mechanisms and possible functional significance. We have looked up the literatures regarding the anatomical connections between the SLD and hippocampus before conducting this analysis and they were discussed in the manuscript (Page 16-17, Line 840-844). Both direct and indirect SLD-hippocampus neural pathways have been reported (Datta et al. 1998; Renouard et al., 2015; Koike et al., 2017; Yamada and Ueda, 2019). In our opinions, these evidences thus provided morphological bases for simultaneously recording LFP/single unit activities in these two areas, as indirect connections can also help to form the functional coherence activities between their nodes (Gent et al., 2018). In the present study, the recorded phase-locking between SLD spiking and hippocampal oscillations was found to be weak but above the significance level, indicating that a functional connection between them exist - perhaps indeed mainly via indirect pathways. More importantly, our data revealed that the phase-locking strength can be significantly increased by orexin-A, and this increase was blunted in the presence of CBX (Revised Supplementary Fig. 9). Therefore, we feel that this analysis fulfilled our purpose to examine the mechanisms underlying the orexin-enhanced hippocampal oscillatory activities, and suggested that an increase in the SLD-hippocampus correspondence was involved.

However, considering your important suggestions, we realized that some gaps still existed when using these results to suggest that the orexin-elicited synchronized excitation is essential for hippocampal activities in REM sleep. This even hindered the main message of the paper. Therefore, we have made the above mentioned

revisions. The Revised Fig. 5 has been listed in the response to Issue 9.

Supplementary Figure 9

Supplementary Fig. 9 The orexin-induced synchronized excitation contributes to the output of SLD neuronal network

a Power spectrogram and raw EEG traces of hippocampal recordings before and after microinjection of orexin-A (30 μ M, 0.3 μ l) into the SLD. During urethane anesthesia, the hippocampal activities were dominated by slow oscillations (0.3-2.5 Hz). Waveforms of this slow oscillations (in 10-second epoch) before and after orexin-A microinjection were expanded at the top. An occasionally appeared disruption of hippocampal slow oscillations by a transient desynchronized/active state in anesthesia tends to increase after orexin-A microinjections.

b Group data showing the power of hippocampal slow oscillations (0.3-2.5 Hz) was increased after microinjection of orexin-A into SLD (baseline: $11688.4 \pm 3224.7 \mu\text{V}^2$, orexin-A: $16959.5 \pm 4566.8 \mu\text{V}^2$; $n = 5$ rats, paired t test; $t_4 = 3.196$, $P < 0.05$).

c Phase locking analysis between SLD spiking activities and hippocampal slow oscillations (0.3 - 2.5 Hz) before and after orexin-A microinjections. Note that a

significant phase preference emerged in the presence of orexin-A.

d Group data showing that the locking strength between SLD spiking activities and hippocampal slow oscillations was significantly increased after microinjection of orexin-A into the SLD (baseline: 0.092 ± 0.009 , $n = 76/97$ pairs; orexin-A: 0.115 ± 0.009 , $n = 84/97$ pairs; Mann-Whitney rank sum test; $z = 3.335$, $P < 0.01$).

e Power spectrogram and raw EEG traces of hippocampal recordings before and after microinjection of CBX (100 mM, 0.3 μ l) into the SLD.

f Group data showing that CBX did not influence the power of hippocampal slow oscillation (0.3-2.5 Hz) (baseline: $20164.1 \pm 5403.9 \mu\text{V}^2$, CBX: $18934.5 \pm 5001.5 \mu\text{V}^2$; $n = 4$ rats, paired t test; $t_3 = 1.000$, $P = 0.391$).

g Phase locking analysis between SLD spiking activities and hippocampal oscillations (0.3 - 2.5 Hz) before and after microinjection of CBX into the SLD.

h Group data showing that the locking strength between SLD spiking activities and hippocampal slow oscillations remained unchanged after microinjection of CBX into the SLD (baseline: 0.091 ± 0.006 , $n = 65/89$ pairs; CBX: 0.099 ± 0.004 , $n = 59/89$ pairs; Mann-Whitney rank sum test; $z = 1.814$, $P = 0.070$).

i Power spectrogram and raw EEG traces of hippocampal recordings before and after microinjection of orexin-A (100 mM, 0.3 μ l) into the SLD in the presence of CBX (100 mM).

j Group data showing that orexin-A still increased the power of hippocampal oscillations (0.3-2.5 Hz) in the presence of CBX (CBX: $12680.7 \pm 3971.6 \mu\text{V}^2$ orexin-A: $14834.0 \pm 4641.0 \mu\text{V}^2$; $n = 5$ rats, paired t test; $t_4 = 2.958$, $P < 0.05$). But note that the percentage increase of this oscillation power in CBX/orexin-A condition was largely reduced, compared to that induced by orexin-A microinjection alone (orexin-A: $48.9 \pm 7.0\%$, CBX/orexin-A: $18.4 \pm 2.5\%$; $n = 5$ rats for each group, Mann-Whitney rank sum test; $z = 2.402$, $P < 0.05$), suggesting the involvement of the orexin-elicited synchronized excitation in the SLD output.

k Phase locking analysis between SLD spiking activities and hippocampal oscillations (0.3 - 2.5 Hz) before and after microinjection of orexin-A into the SLD in the presence of CBX.

l Group data showing that the orexin-A induced increase in the locking strength between SLD spiking activities and hippocampal slow oscillations was abolished in the presence of CBX (baseline: 0.086 ± 0.006 , $n = 40/72$; orexin: 0.084 ± 0.006 , $n = 55/72$; Mann-Whitney rank sum test; $z = 0.693$, $P = 0.488$).

Data are presented as mean \pm SEM. n.s., no significant difference; * $P < 0.05$; ** $P < 0.01$.

Issue 11: On the heat map (Fig 5f), there is an interruption of the main oscillatory activity around 1000s (after Orexin-A application) - What does this correspond to ?

Could it be that the animal emerges from anesthesia ?

Response:

In this series of in vivo multiple channel recording experiments, we have carefully monitored the state of animals after anesthesia to assure that they exhibited stable muscle tone inhibition. The phenomenon you mentioned indeed occurred in some of our recordings and is also observed in previous hippocampal LFP recordings under urethane anesthesia (Lockmann et al., 2016). This interruption exhibited characteristics with a lower amplitude and faster frequency towards the theta band, which may represent a transient desynchronized and active state of the hippocampus in anesthesia (Buzsáki et al., 2002). Although this kind of activities tended to occur after SLD orexin microinjections as indicated in this panel (Revised Supplementary Fig. 9a), it was not stably observed under our experimental conditions. We suspected that, similar to the enhanced slow oscillations, this phenomenon may also represent the increased activity level of hippocampus. As a result, we only provided the raw data here for potential interests. According to your suggestions, we have added explanations for this phenomenon in the figure legends to read: "An occasionally appeared disruption of hippocampal slow oscillations by a transient desynchronized/active state in anesthesia tends to increase after orexin-A microinjections."

Issue 12: -Panels from Fig 6e, f must be represented on the same figure since they compared similar experimental conditions. For consistency, the authors must represent their data using the same Y-axis units for power/amplitude quantification (either percentage or mV²).

Response:

Sorry for not presenting this part of results clearly. As described in Issue 9 and 10, we have re-organized the related figures. Panels from the original Fig 6d-h are now moved to the Revised Supplementary Fig. 9. As for original Fig. 5g (now in

Supplementary Fig. 9b), we first found that orexin-A increased the power of hippocampal slow oscillations. The original Fig. 6e (now in Supplementary Fig. 9j) next described that the hippocampal slow oscillation power was still increased by orexin-A in the presence of CBX. As for the original Fig. 6f, we found that the power increase (%) was significantly higher in orexin-A group than orexin-A/CBX group, suggesting the involvement of the gap junction activity in this orexin-elicited power increase. The data now was reported in the main text to avoid misunderstanding. Besides, the same Y-axis units have been used for the LFP slow oscillation power comparisons in this revised figure.

Issue 13: I would encourage the authors to show raw EEG traces (with magnification) in addition to power spectrum heatmaps (Fig. 5 & 6).

Response:

As suggested, the raw EEG traces with magnification has been added in all related panels. Please see the Revised Supplementary Figure 9 listed in issue 10.

Issue 14:- Figure 7 & 8: My interpretation of these results is quite different than the authors's claims.

From Fig 7c(d and also g), the optogenetic activation of orexin terminals in SLD region during NREM sleep induced a cortical activation (left panel) that is likely results in some cases in a (short) awakening (as clearly shown by the EMG signals on the left panel). This cortical 're-activation' reflect a decrease of slow wave activities (0-5-4.5 Hz) and a concomitant increase of fast oscillations including theta and gamma that are typical of wake states. Therefore, my interpretation of these data is that the optogenetic stimulation wake up the animal but certainly do not prolong REM sleep. This latency to wake should be quantified and reported.

Response:

We thank for this comment. As you mentioned and shown in the original Fig 7c-h (now in the Revised Fig. 6c-g), if optogenetic activations of the SLD orexin terminals were applied in NREM sleep, immediate decreases in the EEG amplitude followed by late-onset decreases in EMG amplitude were observed, and these changes were disrupted by transitions to wakefulness in some of the tested trials (purple line indicated in the figure). An increase in EEG theta component was found before the disruption of wakefulness, suggesting that the activation of brain state has already occurred in this period. Considering that brain state activation in NREM sleep may lead to wakefulness or REM sleep, your suggestion on analyzing the latency to wakefulness was quite necessary. We found that the mean latencies to REM sleep (orexin^{mCherry}: 16.9 ± 5.0 s, n = 5 mice; orexin^{Chr2}: 15.8 ± 4.3 s, n = 5 mice; unpaired t test, $t_8 = 0.176$, P = 0.864) or wakefulness (orexin^{mCherry}: 29.0 ± 3.0 s, n = 5 mice; orexin^{Chr2}: 25.1 ± 3.9 s, n = 6 mice; unpaired t test, $t_9 = 0.749$, P = 0.473) were not significantly affected by light stimulations. We also analyzed the percentage distribution of sleep-wakefulness stages during light delivery, and noted that only between ~20 to ~30 s after the delivery, the percentage of wakefulness occurrences was slightly increased at the expense of NREM sleep (Revised Supplementary Fig. 13). This indeed indicates a short awakening effect of the light activation in NREM sleep, just as you pointed out.

We would like to remind that on the observations that a decrease of EMG amplitude accompanying the brain state activation occurred during NREM sleep before the transitions to wakefulness. This kind of EEG/EMG changes reminisces what happened during NREM to REM sleep transitions that depends on the SLD output. The ability of SLD orexin signaling to induce this kind of EEG/EMG changes suggests its potential contributions to physiological REM sleep besides the awakening effect, though not to the direct transitions from NREM sleep. To examine this potential contribution, light stimulation was thus performed during REM sleep in the next set of experiments shown in the original Fig.7 i, j (Revised Fig.6h-j). Through this protocol, we found that the orexin-enhanced SLD output indeed contributed to prolong the REM sleep episodes. Moreover, optogenetic/chemogenetic inhibition of

SLD orexin signaling had the opposite effects, confirming its physiological involvement. Therefore, the SLD orexin signaling induced changes in EEG/EMG pattern were preferred to stabilize REM sleep, rather than induce wakefulness, during REM sleep.

All these findings support the notion that sleep/wakefulness stages were determined by the coordination of brain states and motor activities. Intriguingly, a perspective has been recently proposed that, for the orchestration of sleep/wakefulness cycles, common hubs regulating brain states and motor activities may be a general mechanism involved (Liu et al., 2020). It seems that the SLD (Torontali et al., 2019) and SLD orexin signaling may also act in this way. Furthermore, the orexin-enhanced SLD output must interact with other brain signals to influence the sleep/wakefulness stages. The consequences of the enhanced output thus depend on the functional condition of the brain, as we observed after light delivery during NREM and REM sleep, respectively.

Following the suggested analysis, we realized that we should not only focused on the descriptions of REM sleep state analysis in the manuscript. The related descriptions in this part of experiments has been re-written (Page 21-22, Line 495-517) to add these wakefulness state related analysis. Besides, necessary schematic for the light delivery during REM sleep was added into the Revised Fig. 6h, to better distinguish from that of NREM sleep. After these revisions, we feel that more comprehensive effects after optogenetic activation of SLD orexin signaling were presented according to your suggestions.

Issue 15: Importantly, EEG theta/delta ratio is not a sensitive marker of REM sleep and if not considered as such by the community. REM sleep is evidenced by a high amplitude theta rhythm generated in the hippocampus (detected on the cortical EEG as well), a complete muscle atonia, Eye-movement, and fluctuating heartbeat and breathing rate (though the last 3 are difficult to see on classical EMG recordings from neck electrodes). Thus, authors should report theta amplitude using a consistent y-axis unit of EEG power spectrum values.

-Fig 7 h,j: The authors must report raw values of theta amplitude alone instead of theta/delta ratio, or theta percentage (percentage of total power ?), in accordance to the standards in the field.

I am not sure whether EMG (REM/NREM) ratio is useful here.

Response:

We thank for this important suggestion. In the Revised Manuscript, we have replaced all EEG theta/delta ratios of REM sleep by raw values of theta oscillation powers according to your suggestions, and more clearly explained the reason to use $EMG_{REM/NREM}$ ratio as the index to examine the state of muscle atonia. In addition, the unit of raw values of field potentials including the hippocampal LFP and cortical EEG were unified as μV^2 .

As for the original Fig. 7h (now in the Revised Fig. 6g), the EEG theta percentage of total power was used to reflect the relative ratio of fast band EEG activities in NREM sleep. Considering that light-stimulation in NREM sleep decreased the EEG total power without inducing consecutive theta oscillations, we thought it was useful to use the change of this percentage comparison to analyze whether the brain state was activated. In this process, we found that the absolute values of individual EEG components may have a high variation, and several previous studies have suggested to use normalization in EEG analysis to reflect EEG component changes under different experimental conditions (Hayashi et al., 2015; Marquis et al., 2017). Therefore, in the original manuscript, after identifying REM sleep state with a high amplitude theta rhythm and muscle atonia, the insensitive EEG theta/delta power ratio was employed to probe the detailed changes of REM sleep EEG signatures in different experimental groups. Fortunately, following your advice, we realized that the high amplitude theta oscillations are unique EEG activities characterizing REM sleep, and direct raw value comparisons should be used. Therefore, raw values of theta oscillation powers alone have been used for behavioral analysis and a significant influence of SLD orexin signaling on them were observed in both optogenetic and chemogenetic experiments. Please see the Revised Fig. 6i, Fig.

7f and Fig. 8j for details. The related descriptions in the main text have also been revised according to these data.

Muscle atonia is another specific feature of REM sleep, as you mentioned. In the present study, we also aimed at observing the state of muscle atonia during REM sleep, and thereby, examining the influence of SLD orexin signaling on this state. Therefore, as a standard index to reflect muscle atonia that eliminated biased EMG modifications induced by posture changes of different sleep epochs (Chen et al., 2017; Valencia Garcia et al., 2017; Valencia Garcia et al., 2018), the analysis of $EMG_{REM/NREM}$ ratio should benefit for our aim. Results from using this index reflected the experimental phenomenon we observed that an abnormal behavior phenotype characterized by severe and constant disruption of muscle atonia in the majority of REM sleep episodes occurred after silencing of SLD orexin signaling. Descriptions have been added in the main text on Page 23, Line 527-529 to more clearly explain the reason to use this index

Issue 16: Finally, I am left wondering why the authors didn't use a similar retrograde targeting strategies as they use in the silencing experiment (Fig 8) in this set of experiment ? overall it makes results difficult to compare.

Response:

The purpose in our behavioral experiments was to manipulate the orexin-SLD pathway, and thereby, observe its influences on REM sleep. Accordingly, the used virus strategies all aimed at selectively label this pathway in both optogenetics and chemogenetics. For chemogenetics, the retrograde targeting strategy is better to fulfill the aim. For optogenetics, considering that the SLD-projecting orexin somas are sporadically distributed in the large area of LH (Supplementary Fig.1) and the spatial-confined range of light delivery, we implanted the optical fibers above the relatively concentrated orexin terminals in the SLD to effectively manipulate this pathway. In our opinions, both of these strategies largely fulfill the aim to selectively

manipulate the orexin-SLD pathway. When using these different but equally valid strategies to inhibit the orexin-SLD pathway (optogenetics v.s. chemogenetics), we revealed a consistent behavioral effect in decreasing the REM sleep episode duration (opto.: 18.2% v.s. chemo.: 22.0%, Fig. 7h and Fig. 8i), although the actions of optogenetics and chemogenetics may influence the orexin-SLD pathway in an acute and a prolonged manner, respectively.

Issue 17: Figure 8 use GCAMP6 and fiber photometry to show that orexin neurons are silent during NREM sleep, slightly more active during REM sleep and strongly activated when the animal wake up. The higher GCAMP6 activity during REM sleep is somewhat odd in comparison with previous literature (work from B. Jones group) which showed that orexin neurons increase their activity before a REM-to-wake transition, suggestive of an wake-promoting action (as shown with optogenetics in many publications). This may results from activity link to muscle twitches. Is that the case? this is difficult to assess from the EMG integral.

How do the authors explain this elevated activity during REM sleep? this finding is very surprising.

Response:

We thank for this comment. In the present study, jGCaMP7b activities of the SLD orexin terminals at the population level were recorded by fiber photometry method and the raw EMG recordings were now aligned with them (Revised Fig. 7d). As you may see from this revised figure panel, no muscle twitches occurred in the example recording. To further examine your concerns, we have also comprehensively checked the fiber photometry recordings during REM sleep with muscle twitches (Revised Supplementary Fig. 15). From all these recordings, we found that the GCaMP signals elevated during the time course of the entire REM sleep episodes. We further marked the time windows of population calcium transients during REM sleep (larger than 2 fold of the averaged signals), which definitely came from the calcium

activities of the SLD orexin terminals rather than the random fluctuation of the baseline, and found that some of them seem to have some degree of overlap with muscle twitches. In contrast, many of them are distributed in the tonic REM sleep period, and have distinct distance with muscle twitches. All these results suggest that the elevated GCaMP signals during REM sleep may be not specifically correlated with muscle twitches.

Actually, as you mentioned, a lot of attentions has been paid on the wake-promoting role of orexin neurons in previous single unit recordings and optogenetic studies (Lee et al., 2005; Mileykovskiy et al., 2005; Takahashi et al., 2008; Adamantidis et al., 2007). Intriguingly, they also reported the existence of elevated activities of orexin neurons during the entire period of REM sleep. From these reports, the exact physiological relevance of the elevated SLD orexin signaling during REM sleep and underlying mechanisms seem to be complex. While they all reported increased activities of orexin neurons before REM sleep to wake transitions, modest activities of orexin neurons were also found beyond this transition period within the REM sleep. Furthermore, different views on the relationship between orexin neuronal discharges and EMG changes (twitches of neck muscle or whiskers) during REM sleep have been proposed (Lee et al., 2005; Mileykovskiy et al., 2005; Takahashi et al., 2008;). Our analysis on the REM sleep related SLD orexin signaling was thus in consistent with these findings, and the REM sleep relevance may clearly emerge in the SLD-projecting orexin neurons. Thus, as you mentioned above, the activity patterns of projection-specific orexin neuron subgroups need attentions.

We have added related descriptions in the main text on Page 26 Line 604-606, and the Revised Supplementary Fig. 15 is listed below,

Supplementary Fig. 15 The jGCaMP transients of SLD orexin terminals were observed during the entire REM sleep episodes.

a Representative jGCaMP fluorescence traces of the SLD orexin terminals, spectrogram of EEG recordings, and raw EMG recorded simultaneously in an episode of REM sleep (R), the preceding NREM sleep (NR), and the following wakefulness (W). The red boxes labeled the time windows of ten detected jGCaMP transients (larger than 2 fold of the averaged signals), and the two blue boxes in the EMG recordings labeled the time windows of two set of muscle twitches. Note that the time windows in some jGCaMP transients (No. 7 and No. 10) seem to have a certain degree of overlap with that of muscle twitches, while others did not, indicating that the jGCaMP signals of the SLD orexin terminals elevated during the time course of the entire REM sleep episodes, including both tonic and phasic periods.

Issue 18: In Fig 8e the authors report optogenetic silencing experiment during REM sleep, however, some experimental details are lacking from the manuscript and figure. -the authors should report details regarding the targeting of silencing opsins to Orexin cells in vivo and evidence for optical silencing of their neuronal activity.

Response:

We thank for this suggestion. The schematics for targeting strategy of ArchT-GFP to the orexin neurons in vivo and experimental design of optogenetic inhibition have been added in the Revised Fig.7e. We have previously provided morphological data showing the successful expression of light-sensitive opsins in the LH orexin neurons by virus infections in the original Supplementary Fig. 6e, f. With your suggestions, we further provided electrophysiological results concerning the effectiveness of optogenetic exciting or silencing of orexin neuron activities. These electrophysiological results were combined with the morphological data to form a new Figure - Supplementary Fig.11, to provide the methodological support for the optogenetics. Besides, related descriptions on these issues have been also added in the Revised Manuscript on Page 21, Line 484-487 and Page 26, Line 607-611.

Supplementary Fig. 11 Morphological and electrophysiological validation of optogenetic manipulations.

a AAV-Ef1 α -DIO-ChR2-mCherry or AAV-CAG-FLEX-ArchT-GFP were injected into the LH of orexin-Cre mice, to selectively express ChR2-mCherry (orexin^{ChR2-mCherry}) or ArchT-GFP (orexin^{ArchT-GFP}) in the orexin-Cre mice, respectively.

b Example images showing the expression of opsins in the LH orexin neurons (ArchT-GFP in this mice). Scale bar, 100 μ m. fx, fornix.

c Quantification of the percentage of Orexin-A⁺ and ArchT/ChR2⁺ neurons in the ArchT/ChR2⁺ neurons and Orexin-A⁺ neurons, respectively, in orexin-Cre mice after virus injections (n = 6 mice; 3 orexin^{ArchT-GFP} and 3 orexin^{ChR2-mCherry} mice).

d Light (589 nm) stimulation reliably abolished the firing activities in the orexin neurons of the orexin^{ArchT} mice (experiments were repeated in 6 orexin-ArchT⁺ neurons).

e 5, 10 and 20 Hz light (473 nm) stimulation reliably evoked action potentials in the orexin neurons of the orexin^{ChR2} mice (experiments were repeated in 6 orexin-ChR2⁺ neurons).

Data are presented as mean \pm SEM.

Issue 19: -The authors must report values of theta amplitude alone instead of theta/delta ratio, in accordance to the standards in the field.

As in Fig 7, I am not sure whether EMG (REM/NREM) ratio is useful here.

-Fig 8g-o show that DREADD silencing of orexin neurons projecting to SLD decreases the duration of REM sleep episodes. Fig 8n,o should be replaced as suggested for previous in vivo figure 7.

Response:

As mentioned in Issue 15, we have replaced the theta/delta ratios in the original Fig. 8 by the raw values of theta oscillation power. These data are now presented in the Revised Fig. 7f and Fig. 8j. The corresponding descriptions in the main text were also revised. Please see Page 27, Line 612-613 and Page 30, Line 687-688.

Issue 20: -Finally, the author must report a clear quantification of targeting methods used in this study (Fig 7, 8).

Response:

We thank for this suggestion. As suggested, we have provided more details for the targeting methods in chemogenetics in the Revised Supplementary Fig. 16, which

is also listed below. The quantification for the specificity and efficiency of the viruses used in the optogenetics has been reported in the original Supplementary Fig.6, and more information was added to compose the Revised Supplementary Fig. 11, which is also listed in the response to Issue 18. The Revised Supplementary Fig. 16 is listed below. The related descriptions have been added in the main text on Page 28, Line 656-658.

Supplementary Fig. 16 Morphological and electrophysiological validation of chemogenetic manipulations.

a In chemogenetics, AAV-retro-hSyn-DIO-hM4D (Gi)-mCherry was bilaterally injected into the SLD of the orexin-Cre mice, to selectively label the orexin-SLD pathway.

b An example coronal image showing that virus injection in the SLD resulted in the expression of hM4D (Gi)-mCherry in the SLD-projecting orexin neurons in the LH. Scale bar, 100 μm.

c Quantification of the percentage of orexin-A⁺/hM4D⁺ neurons in all orexin-A⁺ neurons (SLD-projecting orexin neurons, 15.2 ± 1.4%) and the percentage of orexin-A⁺/hM4D⁺ neurons in all hM4D⁺ neurons (92.4 ± 4.1%) after the virus infections (n = 4 orexin^{SLD-hM4D} mice).

d Patch-clamp recordings were performed on the hM4D (Gi)-mCherry⁺ LH orexin neurons after 2-month of the virus injections in the brain slices. A 2-min bath application of CNO (10 μM) totally abolished the firing activities in all 6 identified hM4D (Gi)-mCherry⁺ LH orexin neurons.

Data are presented as mean ± SEM.

Reviewer #2 (Remarks to the Author):

This is an interesting paper, presenting important evidence for a causal role of orexin axon activity in SLD in promoting REM sleep. The most important finding in the paper is that optogenetic silencing of SLD orexin cell axons disrupts REM sleep, while optogenetic activation of these axons promotes REM sleep. Overall I am enthusiastic about this paper. However, there are several important changes that would significantly improve it:

Response: We would like to thank the reviewer for the encouraging comments on the present study. To improve the presentation and language issue of our study, we have made careful revisions of our manuscript with the help of several native speakers from the *American Journal Expert (AJE, USA)*. Besides, new experiments were conducted to further consolidate conclusions in the gap junction section based on your suggestions.

Issue 1: Abstract: As far as I know, the efficiency of the retrotracer (i.e. what percentage of true connections it labels) is known. Therefore, the precise percent claim is unjustified ("Here, we found that ~7% orexin neurons projected to the sublaterodorsal tegmental nucleus (SLD) and exhibited REM sleep related activation"), since many more orexin neurons may project there but are not detected due to tracer inefficiency.

Response:

We thank for this professional suggestion. As suggested, the descriptions on the tracing experiments need to be more objective. We have re-written this sentence in the abstract to read "Here, we found that a proportion of orexin neurons projected to the sublaterodorsal tegmental nucleus (SLD) and exhibited REM sleep related activation". Besides, we have examined descriptions related to the tracing experiments to avoid

unjustified percent claim in the Revised Manuscript.

Issue 2: General language/terminology used:

Overall the paper is elegantly written and a pleasure to read.

However, the authors repeatedly use certain scientific terms without any justification, which makes their claims unsubstantiated and weakens the paper. I suspect this is a linguistic misunderstanding rather than over-interpretation of results, and I think it must be corrected by textual re-wording. Key examples:

- “Efficient” (eg in efficient output, output efficiency). Efficient means achieving maximum output with minimum resource - but the authors never study any resource allocation. So efficient must be removed, simply output would suffice.
- “Information” (eg orexin-elicited information). Information, in science, has a specific mathematic definition and the authors have not studied or defined this. So they should replace information with a word like effect or response.

Response:

We thank for the comment and your help in the language issue. As suggested, "efficient/efficiency" has been removed, and "orexin-elicited information" has been replaced by "orexin-elicited effects" in the Revised Manuscript. Besides, we have carefully carried out textual re-wording to avoid inappropriate use of scientific terms in the Revised Manuscript.

Issue 3: The language is also often redundant to the point of making some sentences meaningless, the paper should be edited to fix this. For example “Membrane dynamics influences firing activity..besides membrane potential level, sub threshold fluctuations are involved ..”

These 4 terms (Membrane (potential) dynamics, firing activity, fluctuations, membrane potential level) mean the same thing = membrane potential dynamics, so

this is a very circular statement.

Response:

Thanks again for the help in the language issues. We have re-written these two sentences to read "Intriguingly, in addition to the readily-observed membrane potential level of the SLD neurons, several forms of sub-threshold fluctuations, such as gamma band and spikelet activities, are also involved in shaping their firings." In addition, we have made efforts to precisely describe our ideas without redundant expressions in the Revised Manuscript.

Issue 4: -Please report real neuron counts as well as %

Response:

As suggested, in addition to the percentage value, we have added real neuron counts of related experiments in the figure legends of the Revised Manuscript. For double CTB tracing experiments (Fig. 1d), analysis was made in a total of 3902 orexin neurons from 6 rat brains to compare the distribution of the orexin neurons projecting to the SLD and the LC. As for CTB/c-Fos experiments (Fig. 1k), 2767 orexin neurons from 4 rat brains and 6012 orexin neurons from 7 rat brains were analyzed in the control group and REM sleep deprivation (RSD) group, respectively, to compare their c-Fos expression. In this condition, 178 and 370 SLD-projecting neurons were further extracted from the control and RSD group, respectively, to compare c-Fos expression in the SLD-projecting orexin neurons in these two groups.

Issue 5: -Gap junction section is not convincing. Are the gap junction pharmacological blockers specific? (Probably not..). Did biocytin spread to neighboring neurons from the patch-clamp neuron, as would be expected from gap junction? If not this argues against gap junctions.

Response:

We thank for this important comment. In light of your suggestions, we have repeated dye-coupling experiments with biocytin in the SLD neurons. According to previous reports on this method (Blivis et al., 2019; Lee et al., 2014), a 4-hour post-incubation handling of the slice in 32 °C ACSF after loading biocytin (15 min) into the patched SLD neuron was added to facilitate the spread of biocytin via gap junctions (Revised Supplementary Fig. 6b), instead of immediate paraformaldehyde-fixation in the original biocytin-labeling experiments. In this condition, the spread of biocytin from the patched SLD neuron to its neighboring neuron was revealed (Supplementary Fig. 6c, d), indicating the functional expression of gap junctions in the SLD network. As you may see from this revised figure listed below, the spread of biocytin was observed in 3 of 6 tested SLD neurons and the number of the labeled neurons nearby seems to be relatively low. This kind of phenomenon has also been reported in other neuronal networks known to be extensively connected by Cx-36 (Logan et al., 1996; Gibson et al., 1999), which is perhaps due to the slow diffusion rate of the dye through small or distally located gap junctions (Nagy et al., 2019).

We appreciated your suggestions on using different methods including dye-spreading and pharmacological blockers to examine the gap junction activities, as each method may indeed have its shortages (Manjarrez-Marmolejo and Franco-Pérez, 2016; Nagy et al., 2018). Considering these facts, we also used double patch-clamp recordings to directly record pairs of electrically coupled SLD neurons, and found that orexin increased their coupling conductance. Taken all these data together, we hope that our conclusions about the gap junction section are more convincing.

We have constructed a new figure (Revised Supplementary Fig. 6) to include these new data and added descriptions related to these results in the Revised Manuscript (Page 10, Line 210-214). The Revised Supplementary Fig. 6 is listed below,

Supplementary Fig. 6 Cx-36 is expressed in the SLD glutamatergic neurons, and the spread of biocytin from the patched neuron to its neighboring neuron was observed after a 4-h post incubation

a Double immunostaining of Cx-36 and glutamate indicated that Cx-36 expressed at the cell membrane of SLD glutamatergic neurons in adulthood. In addition, Cx-36 expression was also detected in the arbors of the SLD. Scale bars: 40 μm .

b Biocytin was loaded into a SLD neuron for 15 minutes. After that, the slice was post-incubated in 32 °C ACSF for 4 hours to facilitate the gap junction mediated spread of biocytin. Then, the slice was fixed, dehydrated and re-sectioned for the immunostaining process.

c,d After the immunostaining process, the spread of biocytin from the patched neuron (arrows) to its neighboring neuron (arrow heads) was observed in the SLD in 3 of 6 tested neurons, suggesting the existence of electrical coupling in the SLD. Scale bars: 10 μm .

Issue 6: References:

The paper is missing some key references in this field (I.e. link between orexin cell firing and brain electrophysiology). At the very least, references to the following key papers must be cited:

On effects of orexin cell activity on EEG:

Adamantidis et al, Nature 2007 Nov 15;450(7168):420-4

Hara et al, Neuron 2001 May;30(2):345-54.

Chemelli et al, Cell 1999 Aug 20;98(4):437-51

Tabuchi et al, J Neurosci 2014 May 7;34(19):6495-509

de lecea current top behav neurosci. 2016

On effects of orexin cell activity on downstream circuits, including effects of other transmitters and inhibitory orexin signalling in sleep-wake relevant structures:

Kosse et al, PNAS 2017 Apr 25;114(17):4525-4530

Schone et al, Cell Reports 2014 May 8;7(3):697-70

Belle et al, J Neurosci. 2014 Mar 5;34(10):3607-21

Blomeley et al, Nat Neurosci. 2018 Jan;21(1):29-32

Response

We thank for this suggestion. These important literatures have been cited in the Revised Manuscript to benefit the introduction and discussion of the present study.

For references on orexin cell activity on downstream sleep-wake relevant structures, please see references 49-52 discussed on Page 34, Line 781-784.

Besides, the references related to the orexin cell firing on brain EEG activities have also been cited in references 4-6 in the introduction (Page 3, Line 44-46), and references 58 and 63 in the discussion (Page 38, Line 864-869).

Reviewer #3 (Remarks to the Author):

In this exciting study Feng et al investigated the functions of a subpopulation of orexin neurons which project to the sublaterodorsal tegmental nucleus (SLD). Orexins directly excited a large proportion of SLD cells and increased probability of synchronized firing in the SLD. Fiber photometry experiments showed activation of Orx-SLD pathway during REM sleep and wakefulness, whereas opto- and chemogenetic manipulations of this pathway demonstrated its role in REM sleep episodes duration and REM sleep-associated atonia. These novel and important findings show that orexin-SLD pathway stabilizes REM sleep. The study expands our knowledge of sleep-regulating neural circuits. The authors employed an impressive combination of techniques. However, a number of mice is low in several groups, and some controls are not yet provided.

Response: We would like to thank for the encouraging comments on our work. Following your suggestions, we have repeated experiments you mentioned to further examine our conclusions. In addition, the analysis of sleep/wakefulness states transitions in both the experimental and control groups were performed in the optogenetics to more comprehensively analyze the light effects. We also carefully revised the manuscript to avoid typos and inappropriate descriptions with the help of several native speakers from the *American Journal Expert* (AJE, USA).

Specific concerns:

Issue 1: Number of animals in some groups is lower as usually used. In particular, for c-fos experiments (Fig.1) and for fiber photometry experiments (Fig.8d) 4 animals/group were used.

Response:

We thank this suggestion. As suggested, we have repeated the c-Fos experiments

and fiber photometry experiments, to re-examine the REM sleep related activities of the orexin-SLD pathway with care. A total of 7 rats and 7 orexin-Cre mice were now included in the c-Fos RSD group and fiber photometry experiments, respectively. After increasing the number of the tested animals, the results of both experiments confirmed that the orexin-SLD pathway showed activation during REM sleep. For the detailed results, please see Page 8-9, Line 178-188 and Page 26, Line 596-604 in the Revised Manuscript for the c-Fos (Revised Fig.1k) and Fiber photometry (Revised Fig.7d) experiment, respectively.

Issue 2: Fig 5 e- normalized values are shown on the figure. It would be useful to know how often does the coincident spiking typically occur. How high was the average coincident spiking probability?

Response:

We thank for this important suggestion. According to your suggestions, the raw values have been provided to replace the normalized comparisons. For the record, 43 of 465 (9.2%) SLD unit pairs were found to exhibit significant gap junction mediated interactions within ± 1 ms (recorded from the baseline of CBX and orexin-A microinjection experiments, respectively). The mean coincidental spiking probability of them was calculated to be 1.7 ± 0.1 % (ranging from 0.1% - 6.8 %). And because we used a recording probe with a site-interval of 200 μ m, this value may not aid in completely reflecting all SLD gap junction activities in vivo, in contrast to the high percentage (65.6%) of SLD neurons exhibiting gap junction mediated spikelet activities in slice recordings (Fig. 2b and d). But these available gap junction activities in vivo can be still used as an index to reflect the drug effects and measure the relative changes induced by them. Therefore, we previously used normalized values to more intuitively reflect the drug-elicited relative changes on SLD gap junction activities.

However, as you suggested, the raw values may help a more comprehensive interpretation of the SLD gap junction activities in vivo. Therefore, in addition to the

number of SLD unit pairs with significant interactions, we have reported the raw coincident spiking probability values in the Revised Manuscript. The descriptions related to this issue have been re-written to read (Page17, Line 391-397, Revised Fig. 5f) "In addition, the gap junction activities were reflected in vivo by coincidental spiking within a sharp time window of ± 1 ms between tested pairs of SLD neurons. In this condition, 8.4% recorded pairs of SLD units (18 of 215 pairs) showed significant coincident spiking activities. After microinjection of orexin-A into the SLD, the number of unit pairs with significant coincident spiking activities in SLD increased to 13.0% (28 of 215 pairs). Moreover, we found that the coincident probability from the 18 pairs of SLD units with significant interactions in both baseline and orexin-A conditions increased from 1.5 ± 0.2 % to 2.4 ± 0.3 % ($P < 0.01$)."

Similar revisions related to this issue were also made in the original Fig. 6e (now in the Revised Fig. 5i) and Supplementary Fig. 5c (now in the Revised Supplementary Fig. 8b).

Issue 3: For experiments shown on Fig.6 CBX should be used as a control instead of "Baseline", especially since the authors showed that CBX alone reduced coincidental spiking almost two-fold (Suppl.fig.5b).

Response:

We are sorry for the inappropriate captions. In the original Fig.6, CBX was pre-injected into the SLD before orexin-A microinjections. In this condition, data obtained in the presence of CBX were used as the control. The caption "baseline" has been replaced by "CBX" in the related figures, main text, and figure legends.

Issue 4: Fig.7f - are SEM or SD shown on bar plots? If indeed SEM are shown, the variability looks a bit too high for a reported significance level ($p < 0.01$ ** for 20 Hz).

Response:

In this figure, EMG data were reported as mean \pm SEM. For 20 Hz recordings, the mean EMG amplitude was $6.6 \pm 1.7 \mu\text{V}$ (ranging from 3.1 to $13.2 \mu\text{V}$) and $5.6 \pm 1.6 \mu\text{V}$ (ranging from 2.6 to $12.2 \mu\text{V}$) before and after light delivery. We indeed observed that the EMG values varies from animal to animal. This variability may be caused by the recording methods - which use two electrodes inserted in between neck muscles for a differential EMG recording. The relative large SEM of EMG recordings also occurred in some of previous reports (Chen et al., 2017; Valencia Garcia et al., 2018). For analysis on this series of data, we have first used Shapiro-Wilk test to examine their normality distributions. As these data passed the normality test, we used paired t test for further comparisons, and a $P < 0.01$ was reported.

Issue 5:The authors showed that optogenetic activation of orexin-SLD projections prolonged the duration of REM sleep episodes - did it also promote transitions from nonREM to REM if optogenetic stimulation was applied during non-REM sleep?

Response:

We thank for this important suggestion. Although the experiments delivering light stimulation (20 Hz, 1 s in every 3 s) after at least 30 s NREM sleep were used to observe direct EEG/EMG changes and reported non-induction of consecutive EEG theta oscillations, the transition probability analysis with these trails may indeed help further and comprehensively examine the light effects on brain state transitions, including NREM to REM sleep. Therefore, we applied related analysis and constructed new figures based on these results. Following works from de Lecea groups (Adamantidis et al., 2007) with similar experimental designs, we constructed the percentage distribution of sleep-wakefulness stage occurrences within the time course of light delivery, and thus the transition probability changes of the brain states can be assessed. The result showed that only the percentage of short wakefulness bout occurrence slightly increased between ~ 20 to ~ 30 s after light delivery at the expense of NREM sleep (Revised Supplementary Fig. 13b-d), indicating the increased

transition probability from NREM sleep to wakefulness rather than to REM sleep. However, we can not calculate the absolute value of the transition probability, as provided by other studies (Weber et al., 2018), because we did not delivery light stimulations at random time points in the sleep/wakefulness cycle.

We also found that increasing the stimulation intensity (20 Hz lasting for 20 s) during NREM sleep largely increased the occurrence percentage of wakefulness (Revised Supplementary Fig. 14a-d). In this condition, the occurrences of NREM and REM sleep decreased in similar proportions, consistently indicating that only the transition probability from NREM to wakefulness increased. Therefore, taking together with our findings that the activation of SLD orexin terminals during the entire REM sleep episode prolonged its duration, the effects of SLD orexin signaling on sleep/wakefulness stages are complex, but probably had no ability to induce direct NREM to REM sleep transitions.

In fact, all these data demonstrate that the consequences of the orexin-enhanced SLD output may depend on the functional condition of the brain. Intriguingly, a perspective has been recently proposed that, for the orchestration of sleep/wakefulness cycles, common hubs regulating brain states and motor activities may be a general mechanism involved (Liu et al., 2020). Considering that the decrease of EMG amplitude accompanying the brain state activation occurred during light stimulation of the SLD orexin terminals, it seems that SLD orexin signaling may also act in this way and the opposite directions of its actions on motor activity and brain state are interesting. In this condition, the interaction of the orexin-enhanced SLD output with other brain signals' effects would finally determine the sleep/wakefulness stages.

Based on these results, we have constructed the Revised Supplementary Fig. 13 and 14. The related descriptions have also been added in the main text (Page 22, Line 505-517). The Revised Supplementary Fig. 13 and 14 are listed below.

Supplementary Fig. 13 Percentage distributions of sleep/wakefulness occurrences after optogenetic activation (20 Hz, 20 pulses every 3 s) of the SLD orexin terminals during NREM sleep

a Group data showing the mean latency to REM sleep was not changed after light delivery in NREM sleep (orexin^{mCherry}: 16.9 ± 5.0 s, $n = 5$ mice; orexin^{ChR2}: 15.8 ± 4.3 s, $n = 5$ mice; unpaired t test, $t_8 = 0.176$, $P = 0.864$).

b Group data showing the mean latency to wakefulness was not changed after light delivery in NREM sleep (orexin^{mCherry}: 29.0 ± 3.0 s, $n = 5$ mice; orexin^{ChR2}: 25.1 ± 3.9 s, $n = 6$ mice; unpaired t test, $t_9 = 0.749$, $P = 0.473$).

c Percentage distribution of NREM sleep occurrences in 0.5 s bins showed that the NREM sleep occurrences tended to decrease between ~20 s - ~30 s after the light delivery (orexin^{mCherry}: $n = 5$ mice; orexin^{ChR2}: $n = 6$ mice; one-way repeated measure ANOVA reported no significant main effect of light or time point among this time range, but a significant interaction between them; light: $F_{(1,9)} = 2.105$, $P = 0.181$; time point: $F_{(20,180)} = 1.099$, $P = 0.353$; interaction: $F_{(20,180)} = 2.050$, $P < 0.01$; post hoc LSD comparison test revealed a significant change at the 25 s, $P < 0.05$). The period of light delivery was indicated by the blue background.

d Percentage distribution of wakefulness occurrences in 0.5 s bins showed that the wakefulness occurrences tended to increase between ~20 s - ~30 s after the light delivery (orexin^{mCherry}: $n = 5$ mice; orexin^{ChR2}: $n = 6$ mice; one-way repeated measure ANOVA reported no significant main effect of light or time point among this time range, but a significant interaction between them; light: $F_{(1,9)} = 2.646$, $P = 0.138$; time point: $F_{(20,180)} = 0.908$, $P = 0.578$; interaction: $F_{(20,180)} = 1.670$, $P < 0.05$; post hoc LSD comparison test revealed a significant change at the 24.5 s, 25s and 26 s, $P < 0.05$).

e Percentage distribution of REM sleep occurrences in 0.5 s bins showing that the

REM sleep occurrences were not changed after the light delivery (one-way repeated measure ANOVA reported no significant main effect of light or time point among this time range, and no significant interaction between them; light: $F_{(1, 9)} = 0.023$, $P = 0.882$; time point: $F_{(20, 180)} = 0.820$, $P = 0.687$; interaction: $F_{(20, 180)} = 0.820$, $P = 0.687$). Data are presented as mean \pm SEM. n.s., no significant difference; * $P < 0.05$

Supplementary Fig. 14 Percentage distribution of sleep/wakefulness occurrences after optogenetic activation (20 Hz for 20 s) of the SLD orexin terminals during NREM sleep

a An example trial from a tested mouse showing the changes of EMG and EEG recordings induced by optical activation (20 Hz for 20 seconds) of SLD orexin terminals during NREM sleep. The period of light delivery was indicated by the blue background.

b Percentage distribution of NREM sleep occurrences in 0.5 s bins showed that the NREM sleep occurrences decreased after the light delivery.

c Percentage distribution showing that wakefulness occurrences was high during the optical stimulation, suggesting that the brain state activation was intensive in this condition. Consistently, several previous studies have reported that strong excitation of SLD through direct optogenetic activation or inhibition of vIPAG-SLD GABA signaling indeed facilitated brain state activation and may lead to wakefulness from NREM sleep (Weber et al., 2018; Torontali et al., 2019).

e Percentage distribution of REM sleep occurrences in 0.5 s bins showing that the REM sleep occurrences decreased after the light delivery.

Data are presented as mean \pm SEM.

Issue 6: Did the proportion of sleep epochs resulted in wakening change upon the optogenetic excitation or inhibition of Orx-SLD projections?

Response:

We thank for this suggestion. It is very interesting to investigate the potential influences on subsequent sleep/wakefulness stage changes after the prolonged REM sleep episodes by SLD orexin signaling, as one of the important functions of REM sleep is proposed to influence the subsequent sleep/wakefulness homeostasis (Hayashi et al., 2015). In the present study, we found that in the optogenetic excitation/inhibition experiments during REM sleep, all tested and control REM sleep episodes ended by transitions out to wakefulness with a fairly high probability, suggesting that the usual exit to wakefulness after REM sleep termination may not be affected. However, because the light was manually delivered and terminated during REM sleep, the light delivery might slightly extend the tested REM sleep episodes. Therefore, the analysis of the exit from REM sleep may be biased in the present study, though not affecting our analysis for the light influences on REM sleep.

Issue 7: Was an automated sleep phase detection algorithm or visual inspection by an experimenter applied in the closed-loop optogenetic experiments? What was defined as e.g. "stable NREM sleep" - a continuous NREM sleep epoch without wakefulness or REM episodes?

Response:

The closed-loop optogenetic experiments in REM sleep were performed by visual inspection by an experienced experimenter, following the methods in previous studies (Boyce et al., 2016; Weber et al., 2018). The EEG/EMG signals were visually

inspected on-line. An ~10 s delay existed for manual detection to ensure that the mice were in REM sleep. Lasers were manually turned on by the experimenter with 50% probability after detection, and turned off when the REM sleep episode ended. After experiments, we have analyzed the EEG/EMG recordings again to make sure that light was delivered during REM sleep. We have revised the descriptions on this issue for clarity in the Revised Manuscript (Page 52, Line 1173-1179).

For light delivery during NREM sleep, one of our aims was to examine whether REM sleep state can be directly induced, and prolonged NREM sleep was preferred. As a result, only when the EEG/EMG signatures of NREM sleep were visually identified to stably appear for ~30 s, light delivery was performed. We intended to use the "stable NREM sleep" to reflect this situation. According to your suggestions, we realized this description is not precise and may lead to confusions. Therefore, we have examined the related issues and revisions were made to objectively describe the experiments in the Revised Manuscript (Page 51, Line 1162-1163).

Issue 8: Minor comments:

The paper would benefit from further proofreading. There are some typos, e.g. "Orxin" in a subtitle on page 25 , "optogentics" on page 50 etc.

I would add "lateral" to the title of Suppl. Fig. 1: SLD-projecting orexin neurons are sporadically distributed in the LATERAL hypothalamus.

Response:

We are sorry for the typos. These typos have been corrected. Besides, the location of orexin neurons has been corrected to "lateral hypothalamus (LH)" throughout the Revised Manuscript and Supplementary Information. We have made a careful proofreading in the revised manuscript, and other typos and inappropriate descriptions have also been revised.

References

1. Adamantidis, A. R., Zhang, F., Aravanis, A. M., Deisseroth, K., & de Lecea, L. (2007). Neural substrates of awakening probed with optogenetic control of hypocretin neurons. *Nature*, 450(7168), 420-424.
2. Arrigoni, E., Chen, M. C., & Fuller, P. M. (2016). The anatomical, cellular and synaptic basis of motor atonia during rapid eye movement sleep. *The Journal of physiology*, 594(19), 5391-5414.
3. Bourgin, P., Huitrón-Réndiz, S., Spier, A. D., Fabre, V., Morte, B., Criado, J. R., Sutcliffe, J. G., Henriksen, S. J., & de Lecea, L. (2000). Hypocretin-1 modulates rapid eye movement sleep through activation of locus coeruleus neurons. *The Journal of neuroscience : the official journal of the Society for Neuroscience*, 20(20), 7760-7765.
4. Buzsáki G. (2002). Theta oscillations in the hippocampus. *Neuron*, 33(3), 325-340.
5. Boissard, R., Fort, P., Gervasoni, D., Barbagli, B., & Luppi, P. H. (2003). Localization of the GABAergic and non-GABAergic neurons projecting to the sublaterodorsal nucleus and potentially gating paradoxical sleep onset. *The European journal of neuroscience*, 18(6), 1627-1639.
6. Bastianini, S., Silvani, A., Berteotti, C., Lo Martire, V., & Zoccoli, G. (2012). High-amplitude theta wave bursts during REM sleep and cataplexy in hypocretin-deficient narcoleptic mice. *Journal of sleep research*, 21(2), 185-188.
7. Boyce, R., Glasgow, S. D., Williams, S., & Adamantidis, A. (2016). Causal evidence for the role of REM sleep theta rhythm in contextual memory consolidation. *Science (New York, N.Y.)*, 352(6287), 812-816.
8. Blivis, D., Falgairolle, M., & O'Donovan, M. J. (2019). Dye-coupling between neonatal spinal motoneurons and interneurons revealed by prolonged back-filling of a ventral root with a low molecular weight tracer in the mouse. *Scientific reports*, 9(1), 3201.
9. Cruikshank, S. J., Hopperstad, M., Younger, M., Connors, B. W., Spray, D. C., & Srinivas, M. (2004). Potent block of Cx36 and Cx50 gap junction channels by mefloquine. *Proceedings of the National Academy of Sciences of the United States of America*, 101(33), 12364-12369.
10. Clément, O., Sapin, E., Bédod, A., Fort, P., & Luppi, P. H. (2011). Evidence that neurons of the sublaterodorsal tegmental nucleus triggering paradoxical (REM) sleep are glutamatergic. *Sleep*, 34(4), 419-423.
11. Carter, M. E., Brill, J., Bonnavion, P., Huguenard, J. R., Huerta, R., & de Lecea, L. (2012). Mechanism for Hypocretin-mediated sleep-to-wake transitions. *Proceedings of the National Academy of Sciences of the United States of America*, 109(39), E2635-E2644.
12. Chen, M. C., Vetrivelan, R., Guo, C. N., Chang, C., Fuller, P. M., & Lu, J. (2017). Ventral medullary control of rapid eye movement sleep and atonia. *Experimental neurology*, 290, 53-62.
13. Datta, S., Siwek, D. F., Patterson, E. H., & Cipolloni, P. B. (1998). Localization of pontine PGO wave generation sites and their anatomical projections in the rat. *Synapse (New York, N.Y.)*, 30(4), 409-423.
14. Dragunow, M., & Faull, R. (1989). The use of c-fos as a metabolic marker in neuronal pathway tracing. *Journal of neuroscience methods*, 29(3), 261-265.
15. Dauvilliers Y, Jennum P, Plazzi G. Rapid eye movement sleep behavior disorder and rapid eye movement sleep without atonia in narcolepsy. *Sleep Med.* 2013;14(8):775-781.
16. Gibson, J. R., Beierlein, M., & Connors, B. W. (1999). Two networks of electrically coupled inhibitory neurons in neocortex. *Nature*, 402(6757), 75-79.
17. Gent, T. C., Bandarabadi, M., Herrera, C. G., & Adamantidis, A. R. (2018). Thalamic dual control of sleep and wakefulness. *Nature neuroscience*, 21(7), 974-984.

18. Harris, G. C., Wimmer, M., & Aston-Jones, G. (2005). A role for lateral hypothalamic orexin neurons in reward seeking. *Nature*, 437(7058), 556-559.
19. Hasegawa, E., Yanagisawa, M., Sakurai, T., & Mieda, M. (2014). Orexin neurons suppress narcolepsy via 2 distinct efferent pathways. *The Journal of clinical investigation*, 124(2), 604-616.
20. Hu, B., Yang, N., Qiao, Q. C., Hu, Z. A., & Zhang, J. (2015). Roles of the orexin system in central motor control. *Neuroscience and biobehavioral reviews*, 49, 43-54.
21. Hayashi, Y., Kashiwagi, M., Yasuda, K., Ando, R., Kanuka, M., Sakai, K., & Itohara, S. (2015). Cells of a common developmental origin regulate REM/non-REM sleep and wakefulness in mice. *Science (New York, N.Y.)*, 350(6263), 957-961.
22. Iyer, M., Essner, R. A., Klingenberg, B., & Carter, M. E. (2018). Identification of discrete, intermingled hypocretin neuronal populations. *The Journal of comparative neurology*, 526(18), 2937-2954.
23. Kiyashchenko, L. I., Mileykovskiy, B. Y., Maidment, N., Lam, H. A., Wu, M. F., John, J., Peever, J., & Siegel, J. M. (2002). Release of hypocretin (orexin) during waking and sleep states. *The Journal of neuroscience : the official journal of the Society for Neuroscience*, 22(13), 5282-5286.
24. Kaur, S., Thankachan, S., Begum, S., Liu, M., Blanco-Centurion, C., & Shiromani, P. J. (2009). Hypocretin-2 saporin lesions of the ventrolateral periaqueductal gray (vlPAG) increase REM sleep in hypocretin knockout mice. *PloS one*, 4(7), e6346.
25. Knudsen, S., Gammeltoft, S., & Jennum, P. J. (2010). Rapid eye movement sleep behaviour disorder in patients with narcolepsy is associated with hypocretin-1 deficiency. *Brain : a journal of neurology*, 133(Pt 2), 568-579.
26. Krenzer, M., Anaclet, C., Vetrivelan, R., Wang, N., Vong, L., Lowell, B. B., Fuller, P. M., & Lu, J. (2011). Brainstem and spinal cord circuitry regulating REM sleep and muscle atonia. *PloS one*, 6(10), e24998.
27. Kohl, J., Babayan, B. M., Rubinstein, N. D., Autry, A. E., Marin-Rodriguez, B., Kapoor, V., Miyamishi, K., Zweifel, L. S., Luo, L., Uchida, N., & Dulac, C. (2018). Functional circuit architecture underlying parental behaviour. *Nature*, 556(7701), 326-331.
28. Logan, S. D., Pickering, A. E., Gibson, I. C., Nolan, M. F., & Spanswick, D. (1996). Electrotonic coupling between rat sympathetic preganglionic neurones in vitro. *The Journal of physiology*, 495 (Pt 2)(Pt 2), 491-502.
29. Lee, M. G., Hassani, O. K., & Jones, B. E. (2005). Discharge of identified orexin/hypocretin neurons across the sleep-waking cycle. *The Journal of neuroscience : the official journal of the Society for Neuroscience*, 25(28), 6716-6720.
30. Lu, J., Sherman, D., Devor, M., & Saper, C. B. (2006). A putative flip-flop switch for control of REM sleep. *Nature*, 441(7093), 589-594.
31. Luppi, P. H., Clément, O., Sapin, E., Gervasoni, D., Peyron, C., L'égier, L., Salvert, D., & Fort, P. (2011). The neuronal network responsible for paradoxical sleep and its dysfunctions causing narcolepsy and rapid eye movement (REM) behavior disorder. *Sleep medicine reviews*, 15(3), 153-163.
32. Luppi, P. H., Peyron, C., & Fort, P. (2013). Role of MCH neurons in paradoxical (REM) sleep control. *Sleep*, 36(12), 1775-1776.
33. Lee, S. C., Patrick, S. L., Richardson, K. A., & Connors, B. W. (2014). Two functionally distinct networks of gap junction-coupled inhibitory neurons in the thalamic reticular nucleus. *The Journal of neuroscience : the official journal of the Society for Neuroscience*, 34(39), 13170-13182.
34. Lockmann, A. L., Laplagne, D. A., Le ã, R. N., & Tort, A. B. (2016). A Respiration-Coupled Rhythm in the Rat Hippocampus Independent of Theta and Slow Oscillations. *The Journal of neuroscience : the official*

- journal of the Society for Neuroscience, 36(19), 5338-5352.
35. Liu, D., Li, W., Ma, C., Zheng, W., Yao, Y., Tso, C. F., Zhong, P., Chen, X., Song, J. H., Choi, W., Paik, S. B., Han, H., & Dan, Y. (2020). A common hub for sleep and motor control in the substantia nigra. *Science (New York, N.Y.)*, 367(6476), 440-445.
 36. Mileykovskiy, B. Y., Kiyashchenko, L. I., & Siegel, J. M. (2005). Behavioral correlates of activity in identified hypocretin/orexin neurons. *Neuron*, 46(5), 787-798.
 37. Monti, J. M., Torterolo, P., Jantos, H., & Lagos, P. (2016). Microinjection of the melanin-concentrating hormone into the sublateralodorsal tegmental nucleus inhibits REM sleep in the rat. *Neuroscience letters*, 630, 66-69.
 38. Manjarrez-Marmolejo, J., & Franco-Pérez, J. (2016). Gap Junction Blockers: An Overview of their Effects on Induced Seizures in Animal Models. *Current neuropharmacology*, 14(7), 759-771.
 39. Marquis, L. P., Paquette, T., Blanchette-Carrière, C., Dumel, G., & Nielsen, T. (2017). REM Sleep Theta Changes in Frequent Nightmare Recallers. *Sleep*, 40(9), zsx110.
 40. Mickelsen, L. E., Bolisetty, M., Chimileski, B. R., Fujita, A., Beltrami, E. J., Costanzo, J. T., Naparstek, J. R., Robson, P., & Jackson, A. C. (2019). Single-cell transcriptomic analysis of the lateral hypothalamic area reveals molecularly distinct populations of inhibitory and excitatory neurons. *Nature neuroscience*, 22(4), 642-656.
 41. Mahoney, C. E., Cogswell, A., Koralnik, I. J., & Scammell, T. E. (2019). The neurobiological basis of narcolepsy. *Nature reviews. Neuroscience*, 20(2), 83-93.
 42. Nagy, J. I., Pereda, A. E., & Rash, J. E. (2019). On the occurrence and enigmatic functions of mixed (chemical plus electrical) synapses in the mammalian CNS. *Neuroscience letters*, 695, 53-64.
 43. Peyron, C., Tighe, D. K., van den Pol, A. N., de Lecea, L., Heller, H. C., Sutcliffe, J. G., & Kilduff, T. S. (1998). Neurons containing hypocretin (orexin) project to multiple neuronal systems. *The Journal of neuroscience : the official journal of the Society for Neuroscience*, 18(23), 9996-10015.
 44. Renouard, L., Billwiller, F., Ogawa, K., Clément, O., Camargo, N., Abdelkarim, M., Gay, N., Scoté-Blachon, C., Touré R., Libourel, P. A., Ravassard, P., Salvart, D., Peyron, C., Claustat, B., L'éger, L., Salin, P., Malleret, G., Fort, P., & Luppi, P. H. (2015). The supramammillary nucleus and the claustrum activate the cortex during REM sleep. *Science advances*, 1(3), e1400177.
 45. Ren, S., Wang, Y., Yue, F., Cheng, X., Dang, R., Qiao, Q., Sun, X., Li, X., Jiang, Q., Yao, J., Qin, H., Wang, G., Liao, X., Gao, D., Xia, J., Zhang, J., Hu, B., Yan, J., Wang, Y., Xu, M. et al. (2018). The paraventricular thalamus is a critical thalamic area for wakefulness. *Science (New York, N.Y.)*, 362(6413), 429-434.
 46. Sanford, L. D., Yang, L., Tang, X., Ross, R. J., & Morrison, A. R. (2005). Tetrodotoxin inactivation of pontine regions: influence on sleep-wake states. *Brain research*, 1044(1), 42-50.
 47. Sapin, E., Lapray, D., B'érod, A., Goutagny, R., L'éger, L., Ravassard, P., Clément, O., Hanriot, L., Fort, P., & Luppi, P. H. (2009). Localization of the brainstem GABAergic neurons controlling paradoxical (REM) sleep. *PloS one*, 4(1), e4272.
 48. Sasaki, K., Suzuki, M., Mieda, M., Tsujino, N., Roth, B., & Sakurai, T. (2011). Pharmacogenetic modulation of orexin neurons alters sleep/wakefulness states in mice. *PloS one*, 6(5), e20360.
 49. Schöne, C., Cao, Z. F., Apergis-Schoute, J., Adamantidis, A., Sakurai, T., & Burdakov, D. (2012). Optogenetic probing of fast glutamatergic transmission from hypocretin/orexin to histamine neurons in situ. *The Journal of neuroscience : the official journal of the Society for Neuroscience*, 32(36), 12437-12443.
 50. Schöne, C., & Burdakov, D. (2017). Orexin/Hypocretin and Organizing Principles for a Diversity of Wake-Promoting Neurons in the Brain. *Current topics in behavioral neurosciences*, 33, 51-74.

51. Takahashi, K., Lin, J. S., & Sakai, K. (2008). Neuronal activity of orexin and non-orexin waking-active neurons during wake-sleep states in the mouse. *Neuroscience*, 153(3), 860-870.
52. Torterolo, P., Sampogna, S., & Chase, M. H. (2013). Hypocretinergic and non-hypocretinergic projections from the hypothalamus to the REM sleep executive area of the pons. *Brain research*, 1491, 68-77.
53. Tsunematsu, T., Tabuchi, S., Tanaka, K. F., Boyden, E. S., Tominaga, M., & Yamanaka, A. (2013). Long-lasting silencing of orexin/hypocretin neurons using archaerhodopsin induces slow-wave sleep in mice. *Behavioural brain research*, 255, 64-74.
54. Totah, N. K., Neves, R. M., Panzeri, S., Logothetis, N. K., & Eschenko, O. (2018). The Locus Coeruleus Is a Complex and Differentiated Neuromodulatory System. *Neuron*, 99(5), 1055-1068.e6.
55. Torontali, Z. A., Fraigne, J. J., Sanghera, P., Horner, R., & Peever, J. (2019). The Sublaterodorsal Tegmental Nucleus Functions to Couple Brain State and Motor Activity during REM Sleep and Wakefulness. *Current biology : CB*, 29(22), 3803-3813.e5.
56. Varin, C., Luppi, P. H., & Fort, P. (2018). Melanin-concentrating hormone-expressing neurons adjust slow-wave sleep dynamics to catalyze paradoxical (REM) sleep. *Sleep*, 41(6), 10.1093/sleep/zsy068.
57. Valencia Garcia, S., Libourel, P. A., Lazarus, M., Grassi, D., Luppi, P. H., & Fort, P. (2017). Genetic inactivation of glutamate neurons in the rat sublaterodorsal tegmental nucleus recapitulates REM sleep behaviour disorder. *Brain : a journal of neurology*, 140(2), 414-428.
58. Valencia Garcia, S., Brischoux, F., Clément, O., Libourel, P. A., Arthaud, S., Lazarus, M., Luppi, P. H., & Fort, P. (2018). Ventromedial medulla inhibitory neuron inactivation induces REM sleep without atonia and REM sleep behavior disorder. *Nature communications*, 9(1), 504.
59. Weber, F., Hoang Do, J. P., Chung, S., Beier, K. T., Bikov, M., Saffari Doost, M., & Dan, Y. (2018). Regulation of REM and Non-REM Sleep by Periaqueductal GABAergic Neurons. *Nature communications*, 9(1), 354.
60. Yang N, Wang GZ, Wen SY, Qiao QC, Liu YH, Zhang J. Orexin exerts excitatory effects on reticulospinal neurons in the rat gigantocellular reticular nucleus through the activation of postsynaptic orexin-1 and orexin-2 receptors. *Neurosci Lett*. 2017;653:146-151.
61. Yamada, R. G., & Ueda, H. R. (2020). Molecular Mechanisms of REM Sleep. *Frontiers in neuroscience*, 13, 1402.
62. Zhang, J., Li, B., Yu, L., He, Y. C., Li, H. Z., Zhu, J. N., & Wang, J. J. (2011). A role for orexin in central vestibular motor control. *Neuron*, 69(4), 793-804.

REVIEWER COMMENTS

Reviewer #1 (Remarks to the Author):

The authors answered all my concerns, however the final data failed to convince me that this circuit is a major player in REM sleep control, and in particular REM sleep facilitation or REM sleep stability as stated in the Abstract: "Using optogenetics and fiber-photometry, we consequently found that orexin-enhanced SLD output was sufficient to prolong REM sleep episodes through consolidating brain state activation/muscle tone inhibition.")

Furthermore, the authors acknowledged, and now provide experimental evidence, that activation of Ox-SLD circuit during NREM sleep induce arousal and not REM sleep. So, this circuit DOES NOT facilitate REM sleep.

In addition, the authors showed that activation of this circuit during REM sleep prolong REM sleep by ~ 10 % (~65 s to ~72 s) and has no effect on EMG activity, while the necessity experiment reports a decrease of ~ 17 % upon silencing of Ox-SLD circuit and no change on EMG activity. Overall, these changes are marginal as compared to other reports in the literature. Therefore, the claim on "necessity" and "sufficiency" of this circuit in REM stability are rather strong and weakly supported by the present data.

Please correct the following:

Fig 6

-panel d: y-axis unit and label must be added to the blue traces. What do these represent ? total EEG power ? why ?

-Panel I & j: controls are missing. Light off can not be considered as a control condition.

Fig 8

-panel e: are those SEM ? it does not seem significant – please revise statistics

As a side note, the authors are misleading when referencing to Lee et al., 2005; Mileykovskiy et al., 2005; Takahashi et al., 2008 in their statement: "Intriguingly, they also reported the existence of elevated activities of orexin neurons during the entire period of REM sleep". These 3 papers do not show elevated activity of orexin neurons during REM sleep. At best, their firing is similar than during NREM sleep, and close to zero. This is very different than "elevated" or "modest" activity as claimed by the authors.

Reviewer #2 (Remarks to the Author):

The authors addressed my points to my satisfaction.

Reviewer #3 (Remarks to the Author):

The authors have addressed my concerns, and have provided the substantial additional dataset. The paper has been substantially improved. I do not have further concerns or comments.

Response to Reviewers' Comments

We, all of the authors, would like to sincerely express our appreciations for the reviewers' time and professional advice in helping improve our manuscript.

Reviewer #1 (Remarks to the Author):

Issue 1: The authors answered all my concerns, however the final data failed to convince me that this circuit is a major player in REM sleep control, and in particular REM sleep facilitation or REM sleep stability as stated in the Abstract: “Using optogenetics and fiber-photometry, we consequently found that orexin-enhanced SLD output was sufficient to prolong REM sleep episodes through consolidating brain state activation/muscle tone inhibition.”

Response:

We thank for the reviewer's recognition on our efforts in revising the manuscript. We also understand your concerns that our descriptions should be more objective to reflect the facts and avoid controversial expression, such as "sufficient". Therefore, the sentence you mentioned has been revised to "Using optogenetics and fiber-photometry, we consequently found that orexin-enhanced SLD output prolonged REM sleep episodes through consolidating brain state activation/muscle tone

inhibition.". In addition, to present our experimental data objectively, efforts have been made to avoid controversial descriptions throughout the manuscript. For details, please see the responses to issues 2 and 3.

As for the data from present study, the behavioral tests finally demonstrate that the amount, episode duration, as well as the hallmark features (i.e. theta oscillation/muscle atonia) of REM sleep were affected when manipulating SLD orexin signaling in REM sleep, supporting that this signaling is involved in the maintenance of REM sleep. And it is noteworthy that the failure of muscle atonia after loss of this signaling reminds the symptoms of REM sleep behavioral disorder (RBD) in narcolepsy.

Moreover, in our opinion, in order to well orchestrate REM sleep, diverse brain circuitries and neuromodulators, including the SLD orexin signaling, should work together, and it seems that different brain circuitries or neuromodulators have divisions in regulating different aspects of REM sleep. For example, it has been reported that optogenetic inhibition of MCH neurons did not influence REM sleep episode duration and only reduced hippocampal theta rhythm, although activation of these neurons extended the duration of REM sleep (Jego et al., *Nature Neuroscience*, 2013). In addition, genetic ablation of vIPAG GABAergic neurons had no effects on the REM sleep episode duration, and preferentially affected the gating of REM sleep (i.e. transitions) (Weber,

Nature Communications, 2018). Also, the blockage of cholinergic input to the SLD only produced a modest shortening of REM sleep bout length, but attenuated the normal increase in hippocampal theta oscillations (Grace, J Neurosci., 2014). Therefore, the revealed stabilization role of SLD orexin signaling in the present study may also help expand our knowledge in REM sleep control.

Issue 2: Furthermore, the authors acknowledged, and now provide experimental evidence, that activation of Ox-SLD circuit during NREM sleep induce arousal and not REM sleep. So, this circuit DOES NOT facilitate REM sleep.

Response:

We feel sorry for not clearly describing the related experimental results. We did not conclude that SLD orexin signaling facilitate or generate REM sleep from NREM sleep. Actually, the optogenetic activation of the SLD orexin signaling during NREM sleep revealed that this manipulation not only induced a brief arousal as you mentioned, but also induced brain state activation with increased theta activities/muscle tone decrease before arousal (Fig. 6c). Upon these observations, our conclusion reads "these data demonstrate that the activation of SLD orexin signaling does not induce complete REM sleep transitions, but

indeed causes the activation of brain state with increased EEG theta component and muscle tone decrease. In NREM sleep, this activation may lead to a short awakening effect."

Furthermore, in our opinion, these results have provided important clues for SLD orexin signaling in REM sleep regulation, since this kind of EEG/EMG changes (e.g. brain state activation with muscle tone decrease) reminisces what occurs during REM sleep that depends on the SLD output. The contributions of this signaling in REM sleep are finally proved by the following behavioral experiments (Figs. 6h-j, 7, and 8). To further avoid misunderstanding, we have thoroughly checked related descriptions in the manuscript,

-a word "complete" in Results has been removed from the summary on Page 22, Line 507-509.

-an expression "for the facilitation of" has been changed to "to regulate" in Discussion on Page 35, Line 817.

-an expression "effectively facilitate" has been changed to "contribute to" in Discussion on Page 36, Line 838.

Issue 3: In addition, the authors showed that activation of this circuit during REM sleep prolong REM sleep by ~ 10 % (~65 s to ~72 s) and has no effect on EMG activity, while the necessity experiment reports a decrease of ~ 17 % upon silencing of Ox-SLD circuit and no change on

EMG activity. Overall, these changes are marginal as compared to other reports in the literature. Therefore, the claim on “necessity” and “sufficiency” of this circuit in REM stability are rather strong and weakly supported by the present data.

Response:

We appreciate your suggestions on comprehensive consideration of different regulation factors for REM sleep control, including SLD orexin signaling, and thereby, dissecting important factors in REM sleep control. As mentioned above, in our opinion, diverse brain circuitries and neuromodulators, including the SLD orexin signaling, should work together to well orchestrate REM sleep state, and it seems that different brain circuitries or neuromodulators have divisions in regulating different aspects of REM sleep.

As for the role of SLD orexin signaling in REM sleep, considering your concerns that SLD orexin signaling alone can not induce REM sleep, we have carefully revised the usage of "sufficiency" or "sufficient" throughout the manuscript to avoid misunderstanding,

-an expression "is sufficient to prolong" in Results has been changed to "contribute to" in the summary on Page 23, Line 534-537.

-a word "causal" has been removed from the Discussion on Page 27, Line 618.

-a word "causal" has been removed from the Discussion on Page 37, Line 859.

-an expression "are crucially required in" has been changed to "contribute to" in the Discussion on Page 37, Line 860.

However, it is noteworthy that the failure of muscle atonia after chemogenetic silencing this signaling closely mimics what occurs in narcolepsy. Therefore, we think SLD orexin signaling is necessary for the stabilization of REM sleep. In addition, the theta oscillation, episode duration, total amount of REM sleep are also influenced by SLD orexin signaling. Intriguingly, it has been previously reported that genetic inactivation of SLD glutamate neurons produced ~30% reduction of daily REM sleep quantities (Valencia Garcia et al., Brain, 2017). Considering that the total time of REM sleep was reduced by ~17% after loss of SLD orexin signaling, the contribution of the orexin's modulation on SLD glutamatergic neurons may also provide some insights for REM sleep amount regulation.

Please correct the following:

Issue 4: Fig 6

-panel d: y-axis unit and label must be added to the blue traces. What do these represent? total EEG power? why?

Response:

Thanks for this suggestion. In order to more directly present the latency of the optical effects on the EEG/EMG changes, we normalized and averaged EEG and EMG recordings of all tested trials in the example animal before wakefulness, as shown above, to form the example blue traces in Fig. 6d. In addition, the statistic analysis of the EEG total power and EMG amplitude from all tested animals was put below in Fig. 6e, f. These explanations were already included in the legends of Fig. 6d. According to your suggestions, to better illustrate this figure, we have further added y-axis unit and label to this panel.

Issue 5: -Panel I & j: controls are missing. Light off can not be considered as a control condition.

Response:

In experiments using optogenetics during REM sleep, we adopted the closed-loop designs that have been reported in previous studies (Boyce et al., Nature Neuroscience, 2016; Weber et al., Nature Communications, 2018). The detailed method has been included in the method section on Page 52, Line 1173-1184. In this experiment, sleep/wakefulness states were classified based on EEG and EMG on-line. When REM sleep was detected, the laser was turned on with 50%

probability and turned off only when the REM episode ended. This allowed comparison of the REM episode durations with and without laser stimulation within the same recording session, and light-off is used as the control in this experimental design (Boyce et al., Nature Neuroscience, 2016; Weber et al., Nature Communications, 2018). In addition, we have tested the mCherry control for the selectivity of the opsin effects in optogenetics during NREM sleep, and no effects were found in the mCherry group.

Issue 6: Fig 8

-panel e: are those SEM? it does not seem significant – please revise statistics

Response:

We have doubled-checked plots in this figure. This panel e reports that CNO significantly reduced the REM sleep amount, compared to the vehicle injections (n = 8 mice). In the dataset describing REM sleep amount changes, the hollow circles represent data from 8 animals in vehicle (black) and CNO group (red). The mean and SEM value from these data are presented in the solid circles. In statistic analysis, we first examined the normality distribution of each dataset using Shapiro-Wilk test. As all these data passed the normality test and this is a pairwise

experiment design, we used paired t test for the following comparisons. The result reported a significant reduction of REM sleep amount.

Issue 7: As a side note, the authors are misleading when referencing to Lee et al., 2005; Mileykovskiy et al., 2005; Takahashi et al., 2008 in their statement: “Intriguingly, they also reported the existence of elevated activities of orexin neurons during the entire period of REM sleep”. These 3 papers do not show elevated activity of orexin neurons during REM sleep. At best, their firing is similar than during NREM sleep, and close to zero. This is very different than “elevated” or “modest” activity as claimed by the authors.

Response:

Many thanks for pointing out our language issue in our last response letter. Based on these 3 literatures and several reviews discussing on them (Tortorolo and Chase, Sleep Sci. 2014; Arrigoni et al., J. Physiol., 2016; Azeez et al., Front Pharmacol., 2018), we intended to express that a certain level of increase of orexin neuronal activities exists from NREM to REM sleep, e.g. from "silent" to "occasional burst discharge" (Mileykovskiy et al, 2005), or to "transient discharges" (Takahashi et al., 2008). In addition, a previous study using micro-dialysis to investigate the release of orexin in the hypothalamus and basal forebrain also

reported that the orexin level was significantly higher in REM sleep than NREM sleep (Kiyashchenko et al., J. Neurosci., 2002), seemingly supporting this kind of activity pattern. Therefore, we used the words "elevated" and "modest". We feel sorry that such usage may lead to misunderstanding according to your opinion. Actually, we agreed that the activity of orexin neuronal entirety is low during REM sleep. Consistently, we also reported a generally low c-Fos expression level of orexin neuronal entirety after the REM sleep rebound. And the subgroup of SLD-projecting orexin neurons were activated against the generally silent background of the orexin neuronal entirety during REM sleep in this experiment.

Reviewer #2 (Remarks to the Author):

The authors addressed my points to my satisfaction.

Reviewer #3 (Remarks to the Author):

The authors have addressed my concerns, and have provided the substantial additional dataset. The paper has been substantially improved.

I do not have further concerns or comments.